# FERERO: A Flexible Framework for Preference-Guided Multi-Objective Learning

**Lisha Chen**[1],   **AFM Saif**[1],   **Yanning Shen**[2],   **Tianyi Chen**[1]
[1]Rensselaer Polytechnic Institute,   [2]University of California, Irvine

## Abstract

Finding specific preference-guided Pareto solutions that represent different trade-offs among multiple objectives is critical yet challenging in multi-objective problems. Existing methods are restrictive in preference definitions and/or their theoretical guarantees. In this work, we introduce a Flexible framEwork for pREfeRence-guided multi-Objective learning (**FERERO**) by casting it as a constrained vector optimization problem. Specifically, two types of preferences are incorporated into this formulation – the *relative preference* defined by the partial ordering induced by a polyhedral cone, and the *absolute preference* defined by constraints that are linear functions of the objectives. To solve this problem, convergent algorithms are developed with both single-loop and stochastic variants. Notably, this is the *first single-loop primal algorithm* for constrained optimization to our knowledge. The proposed algorithms adaptively adjust to both constraint and objective values, eliminating the need to solve different subproblems at different stages of constraint satisfaction. Experiments on multiple benchmarks demonstrate the proposed method is very competitive in finding preference-guided optimal solutions. Code is available at https://github.com/lisha-chen/FERERO/.

## 1 Introduction

Many machine learning tasks inherently involve multiple objectives, which can be different performance metrics such as accuracy, fairness, and privacy; or, the same metrics defined on different data [52, 42]. To tackle such multi-objective problems, it is common to learn a shared model that simultaneously performs well on all the objectives. Compared to learning one model for each objective, learning a shared model has the benefit of reducing both the model size and the inference time. This can be achieved through multi-objective optimization [52, 59, 35], which is to learn a model that minimizes the vector-valued objective. In practical applications, it is of interest to learn solutions with controlled trade-offs or preferences. To further illustrate, we give two examples below.

In fairness-aware machine learning, a trade-off exists between the fairness $f_{\text{fair}}(\theta)$ and accuracy $f_{\text{acc}}(\theta)$ [42, 37], see also Figure 1a. With $\theta$ denoting the model parameter, and $C$ denoting the partial order cone, to find the optimal models that consider different trade-offs, one can solve the following problem with different thresholds $\epsilon$ [9]

$$\text{maximize}_C \ \left(f_{\text{acc}}(\theta), f_{\text{fair}}(\theta)\right)^\top \ \text{s.t.} \ f_{\text{fair}}(\theta) \geq \epsilon. \tag{1.1}$$

Another example is in drug or molecule design, where the goal is to design drugs or molecules with multiple desired properties $f_1(\theta), f_2(\theta), \ldots, f_M(\theta)$. Aiming to align the values of the properties $F(\theta)$ with a predefined preference vector $v$ as in Figure 1b, one can solve the following problem [40, 1, 61]

$$\text{maximize}_C \ F(\theta) \coloneqq \left(f_1(\theta), \ldots, f_M(\theta)\right)^\top \ \text{s.t.} \ BF(\theta) = Bv, \ Bv = 0 \tag{1.2}$$

where $B \in \mathbb{R}^{(M-1) \times M}$ is full row rank.

38th Conference on Neural Information Processing Systems (NeurIPS 2024).

Table 1: Comparison to existing methods. "Flexibility" represents preference modeling, such as by using weights, preference vectors (rays), or constraints. "Exactness" represents the ability to align with a preference vector exactly. "Deter.", "Stoch." represent deterministic and stochastic, respectively. "✗" means not provided in the corresponding work, and "-" means not relevant.

| Method | Preference Flexibility | Exactness | Controlled ascent | Single loop | Convergence Deter. | Stoch. |
|---|---|---|---|---|---|---|
| Linear Scalarization | weight | - | ✗ | ✓ | $T^{-1}$ | $T^{-\frac{1}{2}}$ |
| (Smooth) Tchebycheff [32] | weight | - | ✗ | ✓ | non-asymptotic | ✗ |
| PMTL [33] | inequalities (absolute) | ✗ | ✗ | ✗ | asymptotic | ✗ |
| EPO [41] | $r^{-1}$ ray (ratio, absolute) | ✓ | ✓ | ✗ | asymptotic | ✗ |
| (X)WC-MGDA [44] | shifted ray (absolute) | ✓ | ✗ | ✗ | ✗ | ✗ |
| FERERO (ours) | relative & absolute | ✓ | ✓ | ✓ | $T^{-1}$ | $T^{-\frac{1}{2}}$ |

Then a natural question arises:

*Can we develop
a principled framework to capture flexible
preferences and admit provably convergent
deterministic and stochastic algorithms?*

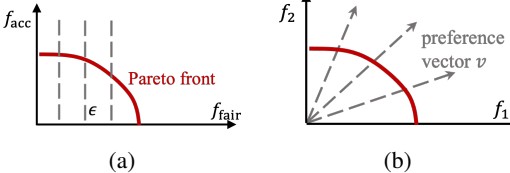

(a)          (b)

Figure 1: Illustration of preferences in different examples. The solid red curves represent the Pareto front, dashed lines represent preference constraints.

Our answer to this question is affirmative. Recognizing that all the aforementioned applications can be addressed within a unified framework, we formulate preference-guided multi-objective learning (PMOL) as a constrained vector optimization problem. Specifically, given a model $\theta \in \mathbb{R}^q$, and the objectives $f_m : \mathbb{R}^q \to \mathbb{R}$, $m = 1, \ldots, M$, we define the constrained vector optimization problem as

$$\min_{\theta \in \mathbb{R}^q} F(\theta) := \big(f_1(\theta), \ldots, f_M(\theta)\big)^\top, \quad \text{s.t.} \quad G(\theta) \leq 0, \ H(\theta) = 0 \qquad \text{(PMOL)}$$

where $G(\theta)$ and $H(\theta)$ are the vector-valued preference constraints such as the examples in (1.1) and (1.2). Here "≤" and "=" are element-wise relations on the vectors, with each row representing one constraint. In these examples, the preferences are directly defined in the objective space, as intersections of half-spaces defined by the hyperplanes; see Figure 1. Thus, $G(\cdot)$ and $H(\cdot)$ in (PMOL) can be expressed as linear functions of $F(\theta)$, given by

$$G(\theta) = B_g F(\theta) + b_g, \quad H(\theta) = B_h F(\theta) + b_h \qquad (1.3)$$

where $B_g \in \mathbb{R}^{M_g \times M}, B_h \in \mathbb{R}^{M_h \times M}$, and $b_g \in \mathbb{R}^{M_g}, b_h \in \mathbb{R}^{M_h}$. Different $B_g, B_h, b_g, b_h$ correspond to different preferences, and thus different trade-offs among the objectives.

A comparison of our methods to existing methods is summarized in Table 1. Specifically, our contributions are listed as follows:

**C1)** We cast the PMOL problem as a constrained vector optimization problem, and develop the FERERO framework to capture flexible preferences.

**C2)** Under the FERERO framework, we develop a meta primal algorithm with a unified subprogram adaptive to both objectives and constraints to meet flexible preferences, eliminating the need for multiple subprograms under different active constraints.

**C3)** Under the FERERO framework, we develop a practical single-loop algorithm with non-asymptotic convergence guarantees. To our best knowledge, this is the *first single-loop primal algorithm* in constrained vector optimization with convergence guarantees.

**C4)** We apply the proposed algorithms to various synthetic and real-world image and speech datasets to demonstrate its ability to find flexible preference-guided optimal models.

In our theoretical analysis, we address the following technical challenges.

**T1)** The commonly used constraint qualification assumptions do not generally hold for the PMOL problem. We overcome this challenge by leveraging the specific structure that the constraints are linear functions of $F$ to prove the calmness condition holds for PMOL. See more details in Lemma 2.

**T2)** The convergence of the single-loop algorithm is slower with the commonly-used merit functions. We provide a sharper analysis by introducing a different merit/Lyapunov function and exploiting the algorithm properties under additional assumptions. See Theorem 3.

**T3)** The convergence analysis often relies on assumptions on bounded functions or bounded constraints. We remove such assumptions by applying similar techniques in [7] with proper choice of Lyapunov functions, and exploiting algorithm properties. See Theorems 2 and 3.

## 2   Problem Setup and A Meta Algorithm

To characterize the optimality conditions of PMOL, we introduce the generalized notion of dominance and the related concept of optimality. We then present a meta-algorithm to solve PMOL.

### 2.1   Problem setup and preliminaries

We first introduce optimality definitions for PMOL that go beyond the standard definitions of Pareto optimality [15, 11, 36]. Given two vectors $v$ and $w$, we use $v < w$ and $v \le w$ to denote $v_i < w_i$ for all $i$, and $v_i \le w_i$ for all $i$, respectively. We use $v \lneq w$ to denote $v \le w$ and $v \ne w$, and define $>, \ge, \gneq$ analogously.

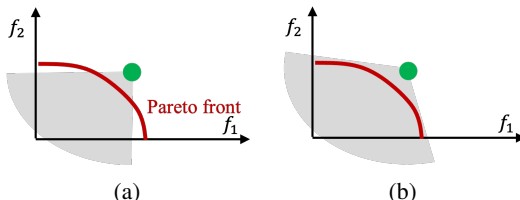

(a)                              (b)

Figure 2: Illustration of $C_A$-dominance. The solid red curves are the Pareto fronts, green dots are the reference points, gray shaded regions are the set of objectives dominating the reference points, under different $C_A$ in (a) and (b).

**Definition 1** ($C_A$-dominance [12, 27]). *Given $v, w \in \mathbb{R}^M$, $A \in \mathbb{R}^{M \times M}$, and $C_A \coloneqq \{y \in \mathbb{R}^M \mid Ay \ge 0\} \ne \emptyset$, we say $v$ strictly dominates $w$ based on $C_A$ if and only if $A(v - w) < 0$.*

The generalized dominance defines a partial order on $\mathbb{R}^M$, i.e., the relation between two vectors. Illustrations of different partial orders are given in Figure 2. Figure 2a shows the dominance relation under the widely used non-negative orthant cone with $C_A = \mathbb{R}_+^M$, corresponding to Pareto optimality. However, as illustrated by the figure, given the initial green reference point, a descent method such as MGDA [15] cannot find points on the Pareto front but outside of the gray shaded region. This poses a critical challenge for applications where specific preference-guided solutions on the Pareto front are needed. Nevertheless, this issue can be addressed by substituting $\mathbb{R}_+^M$ with a more general definition of $C_A$ as displayed in Figure 2b. Under this partial order, a general descent method is able to reach any points on the Pareto front starting from the green reference point.

Based on the partial order, one can then find the minimum or optimal elements in the vector-valued objective space, whose formal definition is provided below.

**Definition 2** ($C_A$-optimal). *A point $\theta \in \mathbb{R}^q$ is $C_A$-optimal if there is no $\theta' \ne \theta$ such that, $AF(\theta') \lneq AF(\theta)$. A point $\theta$ is weakly $C_A$-optimal if there is no $\theta' \ne \theta$ such that, $AF(\theta') < AF(\theta)$.*

Note that, $C_A$ is a polyhedral cone, or the intersection of half-spaces defined by the rows of the inequality $Ay \ge 0$. When $A = I_M$, an $M \times M$ identity matrix, $C_A = \mathbb{R}_+^M \coloneqq \{y \in \mathbb{R}^M \mid y_m \ge 0 \ \forall m \in [M]\}$, then Definition 1 reduces to the commonly used notion of dominance associated with Pareto optimality. The cone $C_A$ can be interpreted as a *relative preference* that defines the objectives' improvement directions, which generalizes the relative preference defined by $\mathbb{R}_+^M$. In contrast, the preference defined by constraints in (1.3) can be interpreted as an *absolute preference* that defines the feasible or preferred set of objective function values. In practice, $C_A$ can be chosen based on the requirements of specific applications. For example, when the controlled ascent of objectives is needed [41], we can choose $C_A$ such that the controlled ascent direction belongs to $-C_A$. We defer the detailed implementation to Section 3.2. The $C_A$-optimal set, denoted as $\mathcal{P}_A$, contains all the $C_A$-optimal models. When $A = I_M$, $\mathcal{P}_A$ is the Pareto optimal set $\mathcal{P}$. The Pareto front is the set of function values evaluated at Pareto optimal models, i.e., $\mathcal{F} = \{F(\theta) \mid \theta \in \mathcal{P}\}$.

We make the following standard assumptions throughout the paper [15, 25, 7].

**Assumption 1.** *1. (Non-negative objectives) $AF(\theta) \ge 0$, and $\mathbf{1}^\top AF(\theta) \ge c_{AF} > 0$ for all $\theta \in \mathbb{R}^q$.*
*2. (Differentiable objectives) $F$ is twice continuously differentiable.*
*3. (Ordering cone with non-empty interior) $C_A$ has a non-empty interior.*

### 2.2   Find the preference-guided direction

In this section, we proceed to discuss an adaptive method to solve (PMOL). At iteration $t$, the algorithm finds an update direction $d_t$ and performs the iterative update $\theta_{t+1} = \theta_t + \alpha_t d_t$ with a step size $\alpha_t$. Ideally, the update direction $d_t$ is chosen to improve the objective $F(\theta)$ and to satisfy

the preference constraints. It is desirable that when the constraints are not satisfied, $d_t$ decreases the violation of constraints and improves the objectives in the general partial ordering sense; when the constraints are satisfied, $d_t$ improves the objectives and ensures the constraints are satisfied. To achieve this, we find a direction $d^*(\theta)$ that solves following subprogram

$$\psi(\theta) \coloneqq \min_{(d,c)\in\mathbb{R}^q\times\mathbb{R}} c + \frac{1}{2}\|d\|^2 \quad \text{s.t.} \ \ A\nabla F(\theta)^\top d \leq \frac{c}{\mathbf{1}^\top AF(\theta)}AF(\theta) \tag{2.1}$$

$$\nabla G(\theta)^\top d + c_g G(\theta) \leq 0, \ \nabla H(\theta)^\top d + c_h H(\theta) = 0$$

where $\|\cdot\|$ denotes the $\ell_2$-norm, $c_g$ and $c_h$ are pre-defined positive constants. Larger $c_g$ and $c_h$ put more emphasis on constraint satisfaction than objective improvement. We call this subprogram *adaptive* since it deals with constraints in an adaptive way, which does not require the initial model to be feasible, nor $\theta_t$ to be feasible at each iteration. But rather, it finds an update direction that decreases the constraint violation. Because of this, it neither requires solving different subprograms at different stages nor requires different treatment of the active set of inequalities as in existing works [33, 41, 44].

We then show in Lemma 1 that the desired properties can be satisfied.

**Lemma 1.** *For the subprogram* (2.1)*, the following holds:*
*If $\theta$ is a local optimal solution with $AF(\theta) > 0$, then $d^*(\theta) = 0$, $\psi(\theta) = 0$. Otherwise, if $\theta$ is not a local optimal solution, then $d^*(\theta) \neq 0$, $\psi(\theta) < 0$, and when $\theta$ is feasible,*

$$2\psi(\theta) \leq -\|d^*(\theta)\|^2 < 0. \tag{2.2}$$

---

**Algorithm 1** A meta FERERO algorithm
1: Initialize $t = 0$, $\theta_0$, step size $\{\alpha_t\}$; define $A$.
2: **while** $\psi(\theta_t) \neq 0$ **do**
3:    Compute gradient $\nabla F(\theta_t)$;
4:    Compute $\lambda_t$ by (approximately) solving (2.3);
5:    Compute the update direction
6:    $d_t = -\nabla F(\theta_t)A_{ag}^\top \lambda_t$;
7:    Update $\theta_t$ by $\theta_{t+1} = \theta_t + \alpha_t d_t$;
8:    Set $t = t + 1$;
9: **end while**

---

*Let $\theta$ be a weak $C_A$-optimal solution, with $(AF(\theta))_m = 0$ for some $m \in [M]$. If there exists feasible and non-strictly improving directions at $\theta$ with $A\nabla F(\theta)^\top d \lneqq 0$, then $d^*(\theta) \neq 0$, $\psi(\theta) < 0$. Otherwise, $d^*(\theta) = 0$, $\psi(\theta) = 0$.*

By Lemma 1, $\|d^*(\theta)\| = 0$ is a stationary condition for PMOL. Recall the feasibility condition requires $[G(\theta)]_+ = 0$ and $|H(\theta)|_{ab} = 0$, where $[\cdot]_+$ and $|\cdot|_{ab}$ are entry-wise ReLU and absolute functions, respectively. And the complementary slackness condition requires $\lambda_g^{*\top}[-G(\theta)]_+ = 0$. Thus $\|d^*(\theta)\|^2 + \lambda_g^{*\top}[-G(\theta)]_+ + \|[G(\theta)]_+\|_1 + \|H(\theta)\|_1$ achieves zero if and only if the model $\theta$ satisfies the first-order KKT condition. Besides the properties in Lemma 1, it has an additional scale-invariant property that is deferred to Lemma 6 due to space limit.

By the Lagrangian of (2.1), the optimal update direction can be expressed in a simple form as a weighted combination of the gradients, i.e. $d^*(\theta) = -\nabla F(\theta)A_{ag}^\top \lambda^*$, with $A_{ag} \coloneqq [A; B_g; B_h]$, and

$$\lambda^* \in \arg\min_{\lambda\in\Omega_\lambda(\theta)} \varphi(\lambda;\theta) \coloneqq \frac{1}{2}\|\nabla F(\theta)A_{ag}^\top \lambda\|^2 - c_g\lambda_g^\top G(\theta) - c_h\lambda_h^\top H(\theta) \tag{2.3}$$

where $\lambda = [\lambda_f; \lambda_g; \lambda_h]$, $\Omega_\lambda(\theta)$ is the domain of the Lagrangian multipliers, given by [1]

$$\Omega_\lambda(\theta) \coloneqq \Omega_{\lambda_f}(\theta) \times \mathbb{R}_+^{M_g} \times \mathbb{R}^{M_h}, \ \text{with } \Omega_{\lambda_f}(\theta) \coloneqq \{\lambda \in \mathbb{R}_+^M \mid \lambda^\top AF(\theta) = \mathbf{1}_M^\top AF(\theta)\}. \tag{2.4}$$

Our goal is to design an algorithm that converges to a KKT solution based on (2.1). However, the KKT condition is not necessary unless certain constraint qualifications (CQs) hold. Prior works [20, 33] assume certain CQs hold, e.g., the Linear Independence Constraint Qualification (LICQ). However, the LICQ assumption (c.f., [20, Section 3.1, (A2)]) does not generally hold at a local optimal solution for problem (PMOL), c.f., Example 1 in Appendix D.3.2. Though some commonly used CQs do not hold generally, in our case, leveraging the specific structure that the constraints are linear functions of $F$, we can justify the calmness CQ in Definition 10 tailored for our problem in Lemma 2, thus the KKT condition is a necessary optimality condition. The proof is deferred to Appendix D.3.2.

---

[1]Note that, our formulation and analysis cover the constrained MOO problem with a simplified subprogram, where $\Omega_{\lambda_f} = \Delta^M$, which is detailed in Remark 4 in Appendix D.1.

**Lemma 2.** *Let $\bar{\theta} \in \mathbb{R}^q$ be a global solution to (PMOL). Define $\Sigma(p,q) \coloneqq \{y \in \mathbb{R}^M \mid B_g y + b_g \leq p, B_h y + b_h = q\}$. If $\Sigma(p,q)$ is a line, the PMOL calmness condition in Definition 10 is satisfied for (PMOL) at $\bar{\theta}$ if $A \in \mathbb{R}^{M \times M}$ is full rank, $H(\theta), G(\theta)$ defined by (1.3) satisfy $[B_h^\top, B_g^\top] \neq 0$, and $B_h, B_g$ are full row rank. Consequently, the KKT condition is a necessary optimality condition.*

Lemma 2 provides a sufficient condition for the KKT condition to be a necessary optimality condition without relying on unjustified assumptions. The requirement that the constraint set is a line in the objective space is common for applications such as alignment to a preference vector.

We then discuss a generic preference-guided multi-objective algorithm based on the subprogram.

### 2.3 A meta algorithm for preference-guided multi-objective learning

Given the model $\theta_t$ at iteration $t$, one can then solve (2.3) to obtain $\lambda_t$. The direction $d_t = -\nabla F(\theta_t) A_{ag}^\top \lambda_t$ is used to update the model $\theta_t$ by $\theta_{t+1} = \theta_t + \alpha_t d_t$ iteratively until convergence. The full procedure of this meta algorithm is summarized in Algorithm 1, where Step 4 is a generic step and can be customized in Section 3.

To establish the non-asymptotic convergence rate, we use the following standard smoothness assumption that has been commonly used in prior works for multi-objective learning [7, 36].

**Assumption 2** (Smooth objectives). *For all $m \in [M]$, $\nabla f_m(\theta)$ is $\ell_{f,1}$-Lipschitz continuous.*

We then state the convergence result for Algorithm 1 in Theorem 1.

**Theorem 1** (Convergence of the generic FERERO algorithm). *Suppose Assumptions 1, 2 hold. Let $\{\theta_t\}$ be the sequences produced by Algorithm 1, with $d_t$ being an $\epsilon$-optimal solution to the subprogram (2.1). If $\|\lambda^*(\theta_t)\|_1 \leq c_\lambda$, $\alpha_t \leq \min\{\frac{1}{c_\lambda \ell_{f,1}\|A_{ag}^\top\|_{\infty,1}}, c_g^{-1}, c_h^{-1}\}$, and $\alpha_t = \Theta(1)$, then*

$$\frac{1}{T}\sum_{t=0}^{T-1} \underbrace{\|\nabla F(\theta_t) A_{ag}^\top \lambda^*(\theta_t)\|^2}_{stationarity} + \underbrace{\lambda_g^*(\theta_t)^\top [-G(\theta_t)]_+}_{complementary\ slackness} + \underbrace{\|[G(\theta_t)]_+\|_1 + \|H(\theta_t)\|_1}_{feasibility} = \mathcal{O}(T^{-1} + \epsilon). \quad (2.5)$$

Theorem 1 guarantees the non-asymptotic convergence for the generic FERERO algorithm. In Algorithm 1, $\lambda_t$ can be solved through projected gradient descent or Frank Wolfe algorithm iteratively within an inner loop. In practice, we usually do not need to solve the subprogram exactly. Next, we discuss the efficient single-loop approximate algorithm based on Algorithm 1.

## 3 Efficient Single-loop Algorithms

In this section, we first discuss algorithm development with the approximate single-loop update and practical choice of preferences. We focus on (PMOL) with *equality constraints only*, i.e., $M_g = 0$. Building upon this, we then discuss the stochastic variants of the algorithms that can be applied to large-scale learning problems.

---
**Algorithm 2** FERERO-SA
---
1: Initialize $t = 0$, $\theta_0$, $\lambda_0$, step sizes $\{\alpha_t, \gamma_t\}$; define $A$, number of iterations $T$.
2: **for** $t = 0, \ldots, T-1$ **do**
3:     Compute gradient $\nabla F(\theta_t)$;
4:     Compute direction $d_t = -\nabla F(\theta_t) A_{ag}^\top \lambda_t$;
5:     Update $\theta_t$ by $\theta_{t+1} = \theta_t + \alpha_t d_t$;
6:     Update $\lambda_t$ by (3.1);
7: **end for**
---

### 3.1 Single-loop approximate algorithm

In practice, if one only requires the converging solutions generated by the algorithm to be feasible, but not all the iterates, then further approximations can be made to the subprogram (2.3). At iteration $t$, to obtain an approximate direction $d_t$, we adopt the following update

$$\lambda_{t+1} = \Pi_{\Omega_\lambda}\big(\lambda_t - \gamma_t \nabla_\lambda \varphi(\lambda_t; \theta_t)\big). \quad (3.1)$$

The single-loop algorithm with the approximate solution is summarized in Algorithm 2. We name it FERERO with Single-loop Approximate update (FERERO-SA) algorithm.

We make the following additional assumption of Lipschitz objectives to prove the convergence of Algorithm 2, which is standard in optimization literature.

**Assumption 3** (Lipschitz objectives). *For all $m \in [M]$, $f_m(\theta)$ is $\ell_f$-Lipschitz continuous.*

To prove the convergence of Algorithm 2, we can use the same merit function with $\ell_1$-norm of $H(\theta_t)$, which leads to a slow convergence rate of $\mathcal{O}(T^{-\frac{1}{6}})$. See Theorem 2 below and its proof in Appendix F.2, where the proof follows similar ideas of the proofs of Theorem 3 and Theorem 5 in [7].

---

**Theorem 2** (Convergence of the FERERO-SA algorithm). *Suppose Assumptions 1, 2, 3 hold, and $M_g = 0$. Let $\{\theta_t\}, \{\lambda_t\}$ be the sequences produced by Algorithm 2 with $A = I$ and $\Omega_{\lambda_f}(\theta) = \Delta^M$ (c.f. Remark 4). Assume $\lambda_t, \lambda^*(\theta_t)$, and $\lambda_\rho^*(\theta_t) := \arg\min_{\lambda \in \Omega_\lambda} \varphi(\lambda; \theta_t) + \frac{\rho}{2}\|\lambda\|^2$ are bounded. With properly chosen step sizes $\alpha = \Theta(T^{-\frac{5}{6}})$, $\gamma = \Theta(T^{-\frac{1}{6}})$, and hyperparameters, it holds that*

$$\frac{1}{T}\sum_{t=0}^{T-1} \|\nabla F(\theta_t) A_{ag}^\top \lambda_t\|^2 + \|H(\theta_t)\|_1 = \mathcal{O}\left(T^{-\frac{1}{6}}\right). \tag{3.2}$$

---

To obtain a sharper convergence rate, we consider a different merit function with $\ell_2$-norm of the constraint $H(\theta_t)$, and under additional assumptions listed below.

**Definition 3** (Proximal PL inequality). *Define $D_{\varphi,\gamma}(\lambda; \theta) := -\frac{2}{\gamma} \min_{\lambda' \in \Omega_\lambda} \left\{ \langle \nabla_\lambda \varphi(\lambda; \theta), \lambda' - \lambda \rangle + \frac{1}{2\gamma}\|\lambda' - \lambda\|^2 \right\}$. We say $\varphi(\lambda; \theta)$ satisfies the $\mu_\varphi$-proximal PL inequality on the point $(\lambda, \theta)$, if there exists some constant $\mu_\varphi > 0$ such that $D_{\varphi,\gamma}(\lambda; \theta) \geq \mu_\varphi\big(\varphi(\lambda; \theta) - \varphi(\lambda^*(\theta); \theta)\big)$.*

**Assumption 4.** *For $\theta \in \{\theta_t\}, \lambda \in \{\lambda_t\}$ on the trajectory of Algorithm 2, the following hold:*
*1. $\varphi(\cdot; \theta)$ is $\mu_\varphi$-proximal PL in Definition 3;*
*2. For all $m \in [M]$, $\nabla^2 f_m(\theta)$ is $\ell_{f,2}$-Lipschitz continuous.*

Assumption 4-1 essentially requires some regularity conditions of $\varphi(\cdot; \theta)$ on the trajectory of Algorithm 2. Leveraging the fact that $\varphi(\cdot; \theta)$ is convex, it has been discussed in e.g., [28, Appendix B] that if the smallest non-zero singular value of the Hessian is bounded away from zero, then Assumption 4-1 holds. This could be satisfied when the gradients $\nabla F(\theta_t) A_{ag}^\top$ have lower-bounded non-zero singular values on the trajectory. A more detailed analysis of the sufficient conditions for Assumption 4-1 to hold is left for furture work.

We then provide a sharper convergence analysis in Theorem 3. The detailed proof and choices of step sizes and hyperparameters are deferred to Appendix F.3.

---

**Theorem 3** (Sharper convergence of the FERERO-SA algorithm). *Suppose Assumptions 1, 2, 3, 4 hold, and $M_g = 0$. Let $\{\theta_t\}, \{\lambda_t\}$ be the sequences produced by Algorithm 2 with $A = I$ and $\Omega_{\lambda_f}(\theta) = \Delta^M$ (c.f. Remark 4). With properly chosen step sizes $\alpha_t = \Theta(1)$, $\gamma_t = \Theta(1)$, and hyperparameters, it holds that*

$$\frac{1}{T}\sum_{t=0}^{T-1} \|\nabla F(\theta_t) A_{ag}^\top \lambda_t\|^2 + \|H(\theta_t)\|^2 = \mathcal{O}\left(T^{-1}\right). \tag{3.3}$$

---

Theorem 3 states that $\{\theta_t\}$ produced by Algorithm 2 converges to a KKT solution of the PMOL problem in the general nonconvex case. Moreover, both $\|d_t\|^2$ and $\|H(\theta_t)\|^2$ converge to zero at a rate of $\mathcal{O}(T^{-1})$, implying the convergence of both the objective values and the preference constraints. Note that, the convergence in terms of $\|H(\theta_t)\|^2$ at a rate of $\mathcal{O}(T^{-1})$ is weaker compared to the one with $\|H(\theta_t)\|_1$ at the same rate for Algorithm 1. This is reasonable since Algorithm 2 only uses a one-step approximate update of $\lambda_t$ instead of exactly solving the subprogram.

**The stochastic variant.** We employ a stochastic variant of Algorithm 2 based on the double sampling techniques developed in the recent work [7]. The update is given by

$$\theta_{t+1} = \theta_t + \nabla F_{\xi_{t,1}}(\theta_t) A_{ag}^\top \lambda_t \tag{3.4a}$$

$$\lambda_{t+1} = \Pi_{\Omega_\lambda}\big(\lambda_t - \gamma_t \tilde{\nabla}_\lambda \varphi(\lambda_t; \theta_t)\big) \tag{3.4b}$$

$$\tilde{\nabla}_\lambda \varphi(\lambda_t; \theta_t) = A_{ag} \nabla F_{\xi_{t,2}}(\theta_t)^\top \nabla F_{\xi_{t,1}}(\theta_t) A_{ag}^\top \lambda_t - [0^\top, c_h H_{\xi_{t,1}}(\theta_t)^\top]^\top \tag{3.4c}$$

where $\tilde{\nabla}$ is the unbiased stochastic estimate of the gradient, and $\xi_{t,1}$ and $\xi_{t,2}$ are two independent stochastic samples obtained at iteration $t$.

The full description of the stochastic algorithm and its convergence guarantee are deferred to Appendix G. We provide a converegnce rate guarantee that matches the rate of SGD under additional assumptions on the bounded variance of the stochastic gradients.

## 3.2 Choice of relative preferences

As briefly discussed in Section 2.1, the ordering cone and the corresponding matrix $A$ can be specified according to practical needs. We first discuss how to obtain matrix $A$ for the relative preference given the set of improvement directions. Then we discuss how to choose the relative preference to allow controlled ascent update, which is useful for touring the Pareto front [41].

**Ordering cone generation.** In practice, to obtain the polyhedral cone that defines the partial order, one can usually first define the extreme rays of the polyhedral cone. We then show how to convert the extreme ray description of the cone to the half-space description given by matrix $A$, i.e., $C_A = \{y \in \mathbb{R}^M \mid Ay \geq 0\}$, by showing how to compute $A$ from the extreme rays.

Let $Y = [y_1 \cdots y_M] \in \mathbb{R}^{M \times M}$ be a matrix that contains all the extreme rays of $C_A$ as its column vectors, then $C_A = \{Y\lambda \mid \lambda \geq 0\}$. Let $a_m^\top \in \mathbb{R}^{1 \times M}$ denote the row vectors of $A$ for all $m \in [M]$. Then all $a_m$ can be found by $a$ that solves the following linear feasibility program

$$\underset{a \neq 0, \lambda \geq 0}{\text{find}} \quad \text{s.t.} \ \ Y\lambda = c, \ \ c^\top a = 0, \ \ Y^\top a \geq 0. \tag{3.5}$$

**Choice of $C_A$ for controlled ascent.** If $C_A$ is not pre-specified, and the decision maker wants to choose $C_A$ to allow controlled ascent, it can be achieved with the following procedure. Let $F_0 = F(\theta_0)$ be the objective of the initial iterate of the algorithm, and $F_{tg}$ be the target function value along the controlled ascent direction. To ensure $F_{tg} - F_0 \in -C_A$ for controlled ascent, we include $(F_0 - F_{tg})/\|F_0 - F_{tg}\|$ in the set of extreme rays, then take the extreme rays of the convex hull of the new set to form the columns of $Y$. Finally, we obtain $C_A$ by solving (3.5).

# 4 Related Works

To put our work in context, we review the most relevant literature in (preference-guided) multi-objective optimization, constrained optimization, with a focus on gradient-based approaches.

**Multi-objective optimization (MOO).** A straightforward approach of MOO is to use scalarization to transform MOO into a single-objective optimization problem [43]. Another popular approach focuses on finding update directions which avoid conflicts with the gradients of the objectives [52, 59, 35]. A foundational algorithm in this domain is the Multiple Gradient Descent Algorithm (MGDA) [15, 17, 11, 36], which dynamically weights gradients to find a steepest common descent direction for all objectives. Later on, variants of MGDA are developed, which are discussed in detail in Appendix B.1 and [7]. However, solutions based on MGDA usually cannot capture pre-defined user preferences that represent various trade-offs on the Pareto front. This motivates the development of *preference-guided multi-objective optimization* methods.

Preferences can be modeled through weights or thresholds assigned to different objectives [43]. For example, scalarization-based methods use the $\ell_p$-norm of the weighted vector-valued objective to convert the vector-valued objective into a scalar-valued objective, e.g., Linear scalarization (LS), Tchebycheff scalarization; see e.g., [32]. Then the problem can be solved by single-objective optimization on the scalar objective. The $\epsilon$-constraint methods enforce threshold constraints on different objectives, then solve the problem by constrained optimization; see e.g., [9]. More recently, preferences have been modeled by preference vectors defined in the objective space. Then the problem can be formulated as finding Pareto optimal solutions satisfying the constraints defined by the preference vectors [33], or optimizing the distance to the preference vectors [41, 44]. The key difference between FERERO and these works is that FERERO can capture more flexible preferences based on a general partial order, and general inequality/equality constraints. Moreover, we provide convergence rate guarantees for the proposed algorithms. A detailed comparison is summarized in Table 1 in Section 1 and Table 5 in Appendix B.2.

**Constrained optimization.** Constrained optimization methods include primal methods, penalty and barrier methods, and primal-dual methods [4, 39]. Our proposed method is related to the primal method that finds an update direction to ensure the models are feasible and improving along the optimization trajectory. To address the limitation that it usually requires a stage-one procedure to

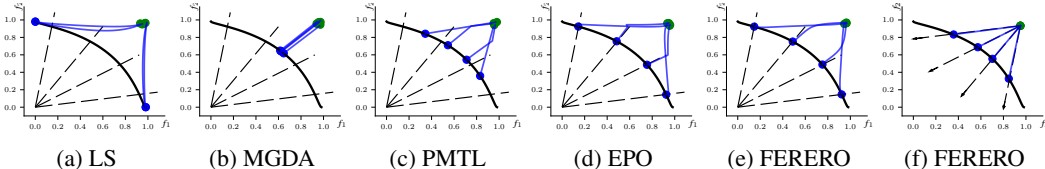

| (a) LS | (b) MGDA | (c) PMTL | (d) EPO | (e) FERERO | (f) FERERO |

Figure 3: Converging solutions (blue dots) and optimization trajectories (blue lines) on the objective space of different methods on synthetic objectives given in (5.1). Dashed arrows represent pre-specified preference vectors. The green dots represent initial objective values.

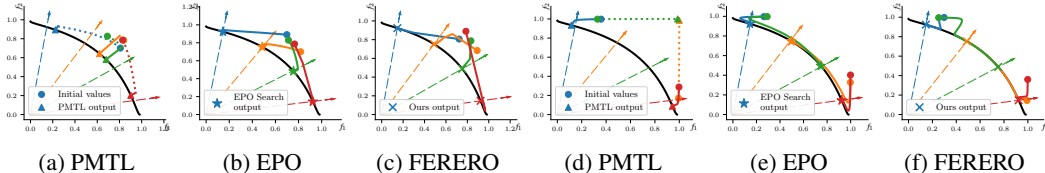

| (a) PMTL | (b) EPO | (c) FERERO | (d) PMTL | (e) EPO | (f) FERERO |

Figure 4: Outputs (colored markers) and optimization trajectories (colored lines) of different methods when initial objectives are near the Pareto front. Different colors represent different preferences.

ensure the initialization is feasible, we use an adaptive approach to ensure the constraint violation is decreasing and converging to zero. This idea can also be found in sequential quadratic programming (SQP). SQP has been widely applied to solve constrained single-objective optimization [21, 6]. Later on, it has also been applied to constrained MOO [16]. Compared to SQP, we use an identity matrix to approximate the Hessian of each objective, and we propose an adaptive variant that automatically adjust the descent amount of objectives. Furthermore, existing SQP algorithms typically require an inner loop to solve the optimal Lagrangian multiplier, resulting in double-loop algorithms. In contrast, we develop a single-loop algorithm which can be more efficient.

**Vector optimization.** Vector optimization [12, 27] generalizes multi-objective optimization by substituting the commonly used component-wise partial order with a more general partial order, such as a general convex-cone induced partial order used in this paper. In the unconstrained setting, the MGDA method is extended to a steepest cone descent method in the vector optimization setting in [25]. In the constrained setting, the first-order optimality conditions are studied in [23, 57]. Algorithms based on projected gradient [24, 18, 19] or conditional gradient [8] are developed to solve vector optimization with parameters in a constraint set, to name a few. Besides gradient-based vector optimization, another line of works focus on black-box vector optimization with discrete design space; see e.g. [3, 2]. To our best knowledge, we are the first to design gradient-based single-loop (stochastic) primal algorithms for constrained vector optimization with convergence rate guarantees.

## 5 Experiments

In this section, we conduct experiments to verify our theory and show the applicability of the algorithms to preference-guided multi-task learning, and multi-objective finetuning of large multi-lingual speech recognition models. We use Linear scalarization (LS), MGDA [52], PMTL [33], EPO [41], XWC-MGDA [44] as baselines for comparison.

**Metrics.** *Objective loss and accuracy.* We report the objective losses and accuracies in classification. *Relative loss profile.* We use the element-wise product of the preference vector and the objective values as a measure of the relative loss profile. *Hypervolume.* Let $F' \in \mathbb{R}^M$ denote a reference point, and $\mathcal{S}$ denote a set of objective function values of the obtained models. Hypervolume measures the size of the dominated space of $\mathcal{S}$ relative to $F'$, which can be computed by $H(\mathcal{S}) = \Lambda(\{q \in \mathbb{R}^M \mid \exists F \in \mathcal{S} : F \leq q \leq F'\})$, where $\Lambda(\cdot)$ denotes the Lebesgue measure. For a fair comparison, we use the Nadir point, i.e., the worst performance on single-task baselines, as the reference point $F'$.

**Additional details.** The implementation and additional experiments can be found in Appendix H.

### 5.1 Synthetic data

Following [33, 41, 44], the first objective we consider is

$$F(\theta) = \left(1 - e^{-\|\theta - \frac{1}{\sqrt{q}}\mathbf{1}\|_2^2}, \ 1 - e^{-\|\theta + \frac{1}{\sqrt{q}}\mathbf{1}\|_2^2}\right). \tag{5.1}$$

The objective has a nonconvex Pareto front (PF). See the results of different methods in Figure 3. With uniformly generated weights from a simplex, LS only finds extreme points on the PF with one objective minimized. MGDA can only find points close to the center of the PF. PMTL can find points

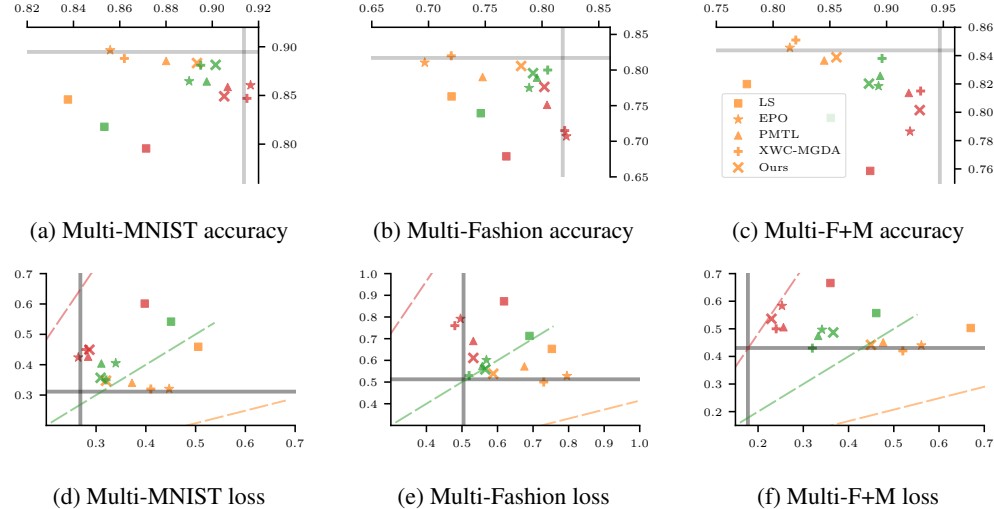

(a) Multi-MNIST accuracy      (b) Multi-Fashion accuracy      (c) Multi-F+M accuracy

(d) Multi-MNIST loss      (e) Multi-Fashion loss      (f) Multi-F+M loss

Figure 5: Training losses and accuracies of various methods with different preferences across three image datasets. The horizontal and vertical axes represent results for objective 1 and objective 2, respectively. Different colored dashed arrows indicate various preference vectors. Different markers denote the solutions obtained by different methods, with marker colors matching the preferences.

in the subregions but not aligned well with the exact preference vectors. Similar to EPO, in Figure 3e, our method finds points that align well with the exact preferences; and in Figure 3f, our method can handle different definitions of preferences.

We conduct another experiment in a more difficult setting where the initial objectives are close to the PF. In Figures 4a-4c, we consider a relatively easier case where the initial model is not too close to the Pareto optimal. For our method, by solving (3.5), $a_1 = [\frac{1}{\sqrt{5}}; \frac{2}{\sqrt{5}}]$, $a_2 = [\frac{2}{\sqrt{5}}; \frac{1}{\sqrt{5}}]$. The corresponding matrix $A$ is given by $A = [a_1, a_2]^\top$. In this setting, all methods converge to the PF, and our method takes the least number of iterations (PMTL takes 100, EPO Search takes 60, and our method takes only 10 iterations). PMTL does not align exactly with the preference vectors, while EPO and our method do. In Figures 4d-4f, PMTL and our method take 200 iterations, EPO Search takes 80 iterations. Results show that for the green and yellow preferences, PMTL moves further away from the PF in the first stage, and does not perform any update in the second stage. It converges to the PF only in 2 out of 4 cases. In contrast, with controlled ascent updates, EPO and our method can converge to the PF and trace the PF until the objectives align exactly with the preferences.

## 5.2 Real data

**Multi-patch image classification.** Following [33, 41, 44], we consider three datasets for image classification, including Multi-MNIST, Multi-Fashion, and Multi-Fashion+MNIST. The two tasks or objectives in all three datasets are to classify the top-left and the bottom-right images, respectively. For a fair comparison, we use LeNet as the backbone neural network. The training losses and accuracies of different methods given different preference vectors are plotted in Figure 5. Experiments for our method are repeated 5 times. Hypervolumes with means and standard deviations are reported in Table 2. The results for other methods in Table 2 are referenced from [44].

Table 2: Hypervolumes of different methods ($\times 10^{-2}$)

| Datasets | LS | PMTL [33] | EPO [41] | XWC-MGDA [44] | FERERO |
|---|---|---|---|---|---|
| Multi-MNIST loss | 1.68 | 1.41 | 1.35 | 1.42 | **1.97**±**0.21** |
| Multi-Fashion loss | 6.75 | 5.90 | 6.02 | 6.77 | **7.76**±**0.18** |
| Multi-F+M loss | 3.63 | 3.03 | 3.76 | **3.89** | 3.82±0.21 |
| Multi-MNIST accuracy | 0.19 | 0.15 | 0.15 | 0.16 | **0.24**±**0.04** |
| Multi-Fashion accuracy | 0.99 | 0.87 | 0.87 | 0.99 | **1.17**±**0.07** |
| Multi-F+M accuracy | 0.48 | 0.40 | 0.50 | 0.52 | **0.53**±**0.04** |

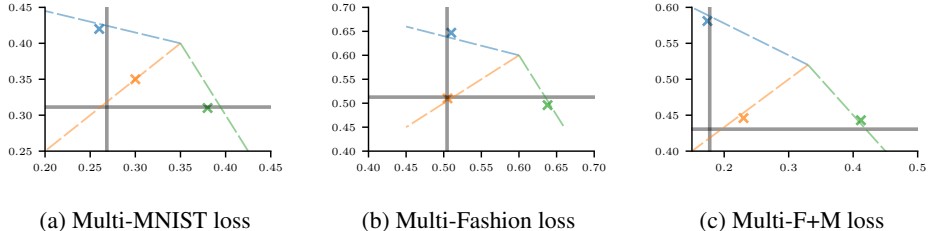

| (a) Multi-MNIST loss | (b) Multi-Fashion loss | (c) Multi-F+M loss |

Figure 6: Losses and preferences of FERERO when the initial objective is close to the Pareto front.

One limitation of EPO is that the preference is defined as a ray from the origin in the objective space, whose corresponding objectives can be unattainable, e.g., the yellow preferences in Figure 5. As a result, the losses of all methods are far away from the preference vectors. In this case, a more flexible choice of preferences is helpful to ensure preference satisfaction. To demonstrate this, we conduct experiments with more flexible preferences; see the results in Figure 6, where the obtained solutions align better with the preference lines compared to those in Figure 5. Moreover, it can perform controlled ascent updates during optimization, which cannot be achieved by PMTL or XWC-MGDA.

**Multi-lingual speech recognition.** We further apply the proposed method to the multi-objective finetuning of pre-trained multi-lingual speech models. We use the Librispeech (100 hours) [47], and AISHELL v1 [5] datasets for multi-lingual speech recognition. A conformer with 8 blocks is used as the model architecture. The total number of parameters is around 64.5M with 58.4M encoder layer parameters and the rest being the classification layer parameters.

Table 3: WERs (%) on Librispeech and AISHELL v1.

| Method | English | Chinese | Average |
|---|---|---|---|
| Komatsu et al. [29] | 7.11 | - | - |
| w/o CPC [51] | 11.8 | 10.2 | 11.0 |
| Init. (M2ASR) [51] | 7.3 | 6.2 | 6.7 |
| LS-FT | 6.8 | 5.9 | 6.4 |
| FERERO-FT | **5.4** | **4.9** | **5.1** |

We consider the objectives associated with the speech recognition Connectionist Temporal Classification (CTC) losses in Chinese and English, denoted as $f_t^{\text{ch}}$ and $f_t^{\text{en}}$, respectively. We also use the self-supervised Contrastive Predictive Coding (CPC) loss $f_p$ for representation learning; that is

$$\min_{\theta} \quad F(\theta) := \left( f_p(\theta), f_t^{\text{ch}}(\theta), f_t^{\text{en}}(\theta) \right)^{\top} \quad \text{s.t.} \quad f_p(\theta) \leq \epsilon_1, \ f_t^{\text{ch}}(\theta) - f_t^{\text{en}}(\theta) = \epsilon_2 \qquad (5.2)$$

where the first constraint ensures to learn a good representation with $\epsilon_1 = 1.2$, and the second constraint avoids one language loss dominates the other with $\epsilon_2 = 0.5$; see more details in Appendix H.1.

Results on the word error rate (WER) are reported in Table 3. The baselines include the state-of-the-art result from Komatsu et al. [29] without an additional large language model, our own implementation of training using only the sum of supervised CTC losses (w/o CPC), the initial pre-trained M2ASR model [51] (init.), linear scalarization of all three objectives for finetuning a pre-trained model with the CPC loss (LS-FT). Results show that considering CPC loss besides the supervised CTC loss improves the average WER by 4.2%, and this can be further improved by 0.3% by finetuning with linear scalarization. However, the LS-FT model has a much better performance in Chinese compared to English. With our proposed approach, the performance gap between different languages is reduced, and the average WER is further improved by 1.3%.

## 6 Conclusions

In this work, we frame preference-guided multi-objective learning as a constrained vector optimization problem. Specifically, we introduce constraints and partial order to capture the absolute and relative preferences. Under this framework, we develop algorithms to solve the constrained vector optimization problem. Our proposed algorithms use a unified formulation without solving different subprograms at different stages. And they enjoy the benefit of allowing controlled ascent and escaping weak optimal solutions. Theoretical guarantees on the non-asymptotic convergence of the deterministic algorithms and their stochastic variants are provided. Experiments on benchmark datasets demonstrate the broad applicability of the proposed algorithms.

## Broader Impacts and Limitations

This paper casts the preference-guided multi-objective learning as a constrained vector optimization problem and proposes an algorithm with single-loop and stochastic variants to solve the problem, which have non-asymptotic convergence guarantees. The proposed method is applied to image classification and speech recognition. The positive impact is that it is a principled method with efficient implementations that has broad applications across various domains. There is no negative social impact.

The proposed algorithm is able to model flexible preferences but at a cost of higher per-iteration complexity compared to scalarization methods. The theoretical guarantees make standard assumptions that the objectives are lower bounded, Lipschitz continuous and smooth. These are common assumptions in the optimization literature, and can be satisfied for neural networks with smooth activation functions.

## Acknowledgements

The work of L. Chen, AFM Saif, and T. Chen was supported by the National Science Foundation (NSF) projects 2401297, 2412486, the RPI-IBM Artificial Intelligence Research Collaboration (AIRC), the Cisco Research Award, and the IEEE Signal Processing Society scholarship. The work of Y. Shen was supported by NSF ECCS-2412484. We also thank Quan Xiao, Prof. Luis Nunes Vicente, Prof. Rongjie Lai for inspiring and helpful discussions, and the anonymous reviewers for their constructive feedback to improve our paper.

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

# Appendix for " FERERO: A Flexible Framework for Preference-Guided Multi-Objective Learning "

## Table of Contents

## A  Notations

A summary of notations used in this work is listed in Table 4 for ease of reference.

Recall that given vectors $v, w$, we use $v < w$ and $v \leq w$ to denote $v_i < w_i$ for all $i$, and $v_i \leq w_i$ for all $i$, respectively. We use $v \lneq w$ to denote $v \leq w$ and $v \neq w$, and define $>, \geq, \gneq$ analogously. In the proof, we use $\|\cdot\|$ to denote the $\ell_2$-norm, and $\|\cdot\|_1$ to denote the $\ell_1$-norm. We use $|\cdot|_{ab}$ to denote the operator that takes element-wise absolute value of a matrix. We use $\mathbf{1}$ and $0$ to denote the all-one and all-zero vectors, respectively. Their dimensions are specified only when they are not clear in the context. We use $[v, w]$ to represent column concatenation of matrices or vectors, and use $[v; w]$ to represent row concatenation of matrices or vectors.

## B  Related Works and Comparison

In this section, we provide a detailed review and comparison of additional related works in multi-task/objective learning, vector optimization, and Pareto front approximation.

Table 4: Notations and their descriptions.

| Notations | Descriptions |
|---|---|
| $\theta \in \mathbb{R}^q$ | Model parameter, or decision variable |
| $\xi$ | Stochastic samples during training |
| $f_{\xi,m}(\theta), f_m(\theta)$ | A scalar-valued objective function evaluated on data point $\xi$, with $f_{\xi,m} : \mathbb{R}^q \to \mathbb{R}$, or on dataset $D$, $f_m$, with $f_m := \frac{1}{|D|} \sum_{\xi \in D} f_{\xi,m}(\theta)$ |
| $\nabla f_m(\theta)$ | Gradient of $f_m(\theta)$, with $\nabla f_m : \mathbb{R}^q \to \mathbb{R}^q$ |
| $F_\xi(\theta), F(\theta)$ | A vector-valued objective function evaluated on data point $\xi$, with $F_\xi : \mathbb{R}^q \to \mathbb{R}^M$, or on dataset $D$, with $F := \frac{1}{|D|} \sum_{\xi \in D} F_\xi(\theta)$ |
| $\nabla F(\theta)$ | Gradient of $F(\theta)$, with $\nabla F : \mathbb{R}^q \to \mathbb{R}^{q \times M}$ |
| $\alpha$ | Step size to update model parameter $\theta$ |
| $\gamma$ | Step size to update multiplier $\lambda$ |

## B.1 Extended related works

In this section, we provide an extended discussion of the works that are closely related to ours.

**Variants and analysis of MGDA.** MGDA [15, 11] finds non-conflicting or the steepest common descent direction at each iteration, which we term as conflict-avoidant (CA) direction. Our work is related to MGDA in the unconstrained setting since when $C_A = \mathbb{R}_+^M$, $M_g = M_h = 0$, i.e., there are no constraints, and $\Omega_{\lambda_f}(\theta) = \Delta^M$ for the subprogram, our Algorithm 1 reduces to MGDA. Non-asymptotic convergence analysis for the deterministic MGDA was first provided in [17]. Convergence of the proximal algorithm was discussed in [54]. Later on, *stochastic* variants of MGDA were developed with convergence analysis [36, 60, 14, 7, 55, 13]. A critical challenge in developing convergent stochastic MGDA is that the CA directions can be biased even if they are calculated from unbiased stochastic gradients of the objectives. This issue can be mitigated using variance reduction techniques on the stochastic gradients. For example, one can use increased batch size [36], or momentum-based methods [60, 14, 13]. Alternatively, one can also use double (independent) sampling [7, 55]. Among these MGDA variants, [60, 14, 13, 7] also use *single-loop* updates, where, instead of exactly solving the weight to combine the objective gradients, the weight is approximately updated only once at each iteration. One benefit of such gradient-based single-loop update is that the approximation approach proposed in [34, Section 3.2] can be applied to largely improve the per-iteration complexity by eliminating the need to compute multiple gradients.

Convergence rate to Pareto stationarity of the above MGDA variants is discussed in existing literature. Specifically, the analysis in [36] focuses on the convex case, while the rest [60, 14, 7, 55, 13] focus on the nonconvex case. However, with merely convergence to the Pareto stationarity, the theoretical benefit of MGDA variants over linear scalarization is unclear. To address this, convergence of the stochastic approximate CA direction to the deterministic optimal CA direction besides convergence to the Pareto stationarity is first analyzed in [14], and later improved in [7] with relaxed assumptions and/or faster convergence rate. Some of the improved analysis techniques in [7] has been applied in [13] to further improve the convergence rate with a momentum-based algorithm, and in [55] with a double-loop algorithm. Moreover, it is discussed in [7] that the analysis technique is widely applicable to other algorithms, such as the SMG algorithm [36] in the nonconvex case for both convergence to Pareto stationarity and to the CA direction. In our proof of Theorem 2, the convergence of the single-loop algorithm, we use similar techniques as in [7], which are detailed in Appendix F.2.

**Pareto front approximation.** Pareto front approximation aims to find multiple different solutions whose objective values approximate the Pareto front. Scalarization-based methods can be used to approximate the Pareto front by enumerating different weights of the objectives. However, they cannot find solutions on the nonconvex part of the Pareto front [43]. Decomposition-based methods partition the objective space into different subsets with constraints that represent different trade-off preferences, and solve the resulting constrained multi-objective optimization problems with gradient-based or evolutionary algorithms [33, 22]. Probabilistic inference methods update a set of models following a distribution that converges to Pareto stationary [38, 49]. The expected update direction of the models typically follows the steepest common descent direction for all objectives. Pareto set learning methods use a neural network to learn a mapping from user preferences to corresponding

models. The learned neural network is able to generate different models with different input user preferences [45, 56, 30, 31]. Although we do not focus on Pareto front approximation in this work, our algorithm can be applied to generate different models based on different diverse preferences to approximate the Pareto front, as in [33].

### B.2 A detailed comparison with existing works

**Preferences as linear constraints of objectives.** Different constraints $S$ partition the objectives into sub-regions, as shown in Figure 1. Many preferences can be modeled by linear equality or inequality constraints [33, 41, 44]. For example, below we list different choices of $C$ for different methods in Figure 1.

(a) $B_g = [0, I_{2:M}]^\top \in \mathbb{R}^{M \times M}, b = -[0, \epsilon_2, \ldots, \epsilon_M]^\top$;

(b) $B_h \in \mathbb{R}^{(M-1) \times M}, b = 0$;

In Figure 1a, the preferences are based on the function values of $f_1$ controlled by different thresholds, corresponding to the inequality constraints defined by (a). In Figure 1b, the constraints are that the objectives $F(\theta)$ should lie on one of the preference vectors $v$, therefore should satisfy the equality constraint $B_h F(\theta) = 0$.

**Detailed comparison with the most relevant works.** Below we provide a fine-grained comparison with some existing works in Table 5, as an extension of Table 1.

In terms of preference modeling, the scalarization-based methods such as Linear Scalarization and Smooth Tchebycheff scalarization use weight of different objectives to model preferences. They are not flexible enough to capture preferences illustrated in Figure 1. PMTL uses a constrained multi-objective optimization formulation, with preferences modeled by inequalities. EPO models the preference by an $r^{-1}$ ray, same as the example given in Figure 1b. (X)WC-MGDA uses a shifted ray not necessarily from the origin to model the preferences. In all of these works, they only model the absolute preferences that define the preferred objective values. In contrast, we also consider the relative preference that define the relative improvement directions of objectives.

In addition to the comparison in Table 1, our framework enjoys additional benefits including the ability to escape weak optimal solutions and to maintain scale-invariance. These abilities are attributed to the subprogram that is adaptive to the objective values, as detailed in Lemma 6.

Table 5: Comparison to existing PMOL methods, extension of Table 1.

| Method | Handle nonconvex PF | General partial order | Single subprogram w/o computing active index | Scale invariance | Escape weak optimal | Provable CQ |
|---|---|---|---|---|---|---|
| Linear Scalarization | ✗ | ✗ | ✓ | ✗ | ✗ | - |
| (Smooth) Tchebycheff [32] | ✓ | ✗ | ✓ | ✗ | ✗ | - |
| PMTL [33] | ✓ | ✗ | ✗ | ✗ | ✗ | assume LICQ |
| EPO [41] | ✓ | ✗ | ✗ | ✗ | ✓ | ✗ |
| (X)WC-MGDA [44] | ✓ | ✗ | ✗ | ✗ | ✓ | ✗ |
| FERERO (ours) | ✓ | ✓ | ✓ | ✓ | ✓ | prove calmness |

Below, we further summarize the reasons behind the benefits of our proposed method. We use "$\rightarrow$" to indicate the reasons on the left and the corresponding benefits on the right.

Flexible preference
{
  relative preference (by general partial order) $\rightarrow$ allow controlled ascent
  absolute preference (by constraints) $\rightarrow$ {
    handle nonconvex Pareto Front
    equality constraints $\rightarrow$ align exactly to preference vector
    constraints are linear functions of objectives $\rightarrow$ provable CQ
  }
}

Adaptive subprogram
{
  adaptive to objectives $\rightarrow$ {
    scale invariance
    ability to escape weak optimality
  }
  adaptive to constraints $\rightarrow$ single subprogram w/o computing active indices $\rightarrow$ non-asymptotic convergence
}

# C    Preliminaries

In this section we introduce preliminaries on the general cone-induced partial ordering and the corresponding optimality conditions for completeness since we use these concepts in our proofs. Then we discuss the relation between the Pareto optimality and the optimality induced by a general polyhedral cone.

## C.1    General cone-induced partial ordering

In this section, we introduce basic definitions, lemmas, propositions, and theorems in vector optimization, including the cone-induced partial ordering, the minimum and weakly minimum associated with the partial ordering in real linear space, and necessary conditions for minimum. These concepts are defined in [27]. We restate them following our notations for completeness. We denote $Z$ as a real linear space, $C, S$ as subsets in $Z$, and $w, x, y, z$ as points or elements in $Z$, $0_Z$ as the zero vector in the space $Z$.

**Definition 4** (Cone). *Let $C$ be a nonempty subset of a real linear space $Z$.*
*The set $C$ is called a* cone, *if $y \in C, \lambda \geq 0 \implies \lambda y \in C$.*

**Lemma 3** (Convex cone). *A cone $C$ in a real linear space is convex if and only if $C + C \subset C$.*

**Definition 5** (Partially ordered linear space). *A real linear space equipped with a partial ordering is a partially ordered linear space.*

**Proposition 1.** *(a) If $\leq$ is a partial ordering on $Z$, then the set $C := \{z \in Z \mid 0_Z \leq z\}$ is a convex cone. If, in addition, $\leq$ is antisymmetric, then $C$ is pointed.*

*(b) If $C$ is a convex cone in $Z$, then the binary relation $\leq_C := \{(x, y) \in Z \times Z \mid y - x \in C\}$ is a partial ordering on $Z$. If, in addition, $C$ is pointed, then $\leq_C$ is antisymmetric.*

**Definition 6** (Ordering cone). *A convex cone characterizing a partial ordering in a real linear space is an ordering cone.*

**Definition 7** (Cone-induced partial ordering). *Let $C$ be a closed pointed convex cone of $\mathbb{R}^M$, with nonempty interior. The partial order in $\mathbb{R}^M$ induced by $C, \leq_C$ is defined by*

$$u \leq_C v, \ \ if \ v - u \in C. \tag{C.1}$$

*The relation induced by $\mathrm{int}(C)$ in $\mathbb{R}^M$, $<_C$ is defined by*

$$u <_C v, \ \ if \ v - u \in \mathrm{int}(C). \tag{C.2}$$

**Definition 8** ($C$-minimum and $C$-weakly minimum). *Let $S$ be a nonempty subset of a partially ordered linear space with an ordering cone $C$, then*
*(a) an element $z \in S$ is called a $C$-minimum of the set $S$, if $(\{z\} - C) \cap S \subset \{z\} + C$, in other words, there exists no other $z' \in S$ with $z' \leq_C z$ and $z' \neq z$;*
*(b) an element $z \in S$ is called a $C$-weakly minimum of the set $S$, if $(\{z\} - \mathrm{int}(C)) \cap S = \emptyset$, where $\mathrm{int}(C) \neq \emptyset$ is the algebraic interior of $C$, in other words, there exists no other $z' \in S$ with $z' <_C z$ and $z' \neq z$.*

**Definition 9** ($C$-stationary). *A point $\theta \in \mathbb{R}^q$ is $C$-stationary if there is no first-order common descent direction $d \in \mathbb{R}^q$ that $\nabla F(\theta)^\top d \in -\mathrm{int}(C)$, i.e., $\mathrm{range}(\nabla F(\theta)^\top) \cap (-\mathrm{int}(C)) = \emptyset$.*

## C.2    Necessary and sufficient conditions for $C$-optimality

Note that, when $C = \mathbb{R}^M_+ := \{z \in \mathbb{R}^M \mid z_m \geq 0 \text{ for all } m \in [M]\}$, $C$-minimum and $C$-weakly minimum in Definition 8 are Pareto minimum and weakly Pareto minimum, respectively. Recall that $F : \mathbb{R}^q \to \mathbb{R}^M$ is a continuously differentiable function. The problem we consider is to find the unconstrained $C$-minimizers of $F$, denoted as $\min_C F(\theta)$ with $\theta \in \mathbb{R}^q$. We then proceed to introduce the relation between $C$-stationarity and Pareto stationarity in this section.

**Proposition 2.** *Let $C$ be a closed convex pointed cone.*
*1) Suppose $C \subseteq \mathbb{R}^M_+$. If $\theta$ is Pareto stationary, $\theta$ is $C$-stationary. In other words, $C$-stationarity is a necessary condition for Pareto stationarity.*
*2) Suppose $\mathbb{R}^M_+ \subseteq C$. if $\theta$ is $C$-stationary, $\theta$ is Pareto stationary. In other words, $C$-stationarity is a sufficient condition for Pareto stationarity.*

*Proof of Proposition 2.* 1) By definition, if $\theta$ is Pareto stationary, then $\mathrm{range}(\nabla F(\theta)^\top) \cap (-\mathrm{int}(\mathbb{R}_+^M)) = \emptyset$. Since $C \subseteq \mathbb{R}_+^M$, then $-\mathrm{int}(C) \subseteq -\mathrm{int}(\mathbb{R}_+^M)$, and we have

$$\mathrm{range}(\nabla F(\theta)^\top) \cap (-\mathrm{int}(C)) \subseteq \mathrm{range}(\nabla F(\theta)^\top) \cap (-\mathrm{int}(\mathbb{R}_+^M)) = \emptyset. \tag{C.3}$$

Therefore, $\theta$ is $C$-stationary.

Following similar arguments, 2) can also be proved. $\qquad\square$

# D Proof of Auxiliary Lemmas

In this section, we provide proof of the main theoretical results in this paper.

## D.1 Lagrangian of the subprogram

*Proof of subprogram reformulation.* Define the Lagrangian function

$$
\begin{aligned}
L(c, d, \lambda_f, \lambda_g, \lambda_h) :=& c + \frac{1}{2}\|d\|^2 + \lambda_f^\top \left(A \nabla F(\theta)^\top d - c(\mathbf{1}^\top A F(\theta))^{-1} A F(\theta)\right) \\
&+ \lambda_g^\top \left(B_g \nabla F(\theta)^\top d + c_g G(\theta)\right) + \lambda_h^\top \left(B_h \nabla F(\theta)^\top d + c_h H(\theta)\right)
\end{aligned}
\tag{D.1}
$$

where $\lambda_f \in \mathbb{R}_+^M$, $\lambda_g \in \mathbb{R}_+^{M_g}$, $\lambda_h \in \mathbb{R}^{M_h}$. By the first-order optimality condition w.r.t. $d$ and $c$, we can obtain that

$$d^* + \nabla F(\theta)(A^\top \lambda_f^* + B_g^\top \lambda_g^* + B_h^\top \lambda_h^*) = 0; \tag{D.2}$$

$$\mathbf{1}^\top A F(\theta) - \lambda_f^{*\top} A F(\theta) = 0. \tag{D.3}$$

Combining the last equation with $\lambda_f \in \mathbb{R}_+^M$, we obtain $\lambda_f^* \in \Omega_{\lambda_f}(\theta)$. Plugging the above results into the Lagrangian function gives

$$
\begin{aligned}
[\lambda_f^*; \lambda_g^*; \lambda_h^*] \in \operatorname*{arg\,min}_{[\lambda_f; \lambda_g; \lambda_h] \in \Omega_\lambda(\theta)} & \frac{1}{2}\|\nabla F(\theta)(A^\top \lambda_f + B_g^\top \lambda_g + B_h^\top \lambda_h)\|^2 \\
& - c_g \lambda_g^\top G(\theta) - c_h \lambda_h^\top H(\theta)
\end{aligned}
\tag{D.4}
$$

which leads to the dual form in (2.3). Since (2.1) is a constrained convex optimization problem where the Slater's condition holds, therefore, the duality gap is zero. $\qquad\square$

**Remark 4.** *Note that we can also have a simplified subprogram with $A = I$, and without adaptation to the objective values, as defined below*

$$\psi(\theta) := \min_{(d,c) \in \mathbb{R}^q \times \mathbb{R}} c + \frac{1}{2}\|d\|^2 \quad \text{s.t.} \ \nabla F(\theta)^\top d \le c\mathbf{1} \tag{D.5}$$

$$\nabla G(\theta)^\top d + c_g G(\theta) \le 0, \ \nabla H(\theta)^\top d + c_h H(\theta) = 0.$$

*This formulation corresponds to the SQP method applied to the constrained MOO problem [16]. Then the corresponding Lagrangian function becomes*

$$
\begin{aligned}
L(c, d, \lambda_f, \lambda_g, \lambda_h) :=& c + \frac{1}{2}\|d\|^2 + \lambda_f^\top \left(\nabla F(\theta)^\top d - c\mathbf{1}\right) \\
&+ \lambda_g^\top \left(B_g \nabla F(\theta)^\top d + c_g G(\theta)\right) + \lambda_h^\top \left(B_h \nabla F(\theta)^\top d + c_h H(\theta)\right).
\end{aligned}
\tag{D.6}
$$

*By the first-order optimality condition w.r.t. $c$, (D.3) can be replaced by*

$$1 - \lambda_f^{*\top} \mathbf{1} = 0. \tag{D.7}$$

*And the rest results remain the same, i.e., (D.2) and (D.4) still hold, while $\Omega_{\lambda_f} = \Delta^M$.*

## D.2 First-order necessary optimality conditions

We then discuss the first-order necessary optimality conditions for problem (PMOL). We begin the discussion with the geometric notions of improving and feasible directions.

**Improving directions.** The improvement directions are defined as generalized common descent directions so that the iterates strictly improve or dominate the previous iterates based on $C_A$, i.e., $F(\theta_t) - F(\theta_{t+1}) \in \text{int}(C_A)$. Denote $d_t \in \mathbb{R}^q$ as an update direction at iteration $t$, and $\alpha_t > 0$ as the step size at the $t$-th iteration. The general update equation given update direction $d_t$ is $\theta_{t+1} = \theta_t + \alpha_t d_t$. Based on first-order Taylor expansion, the amount of improvement at iteration $t$ can be approximately expressed as $F(\theta_t) - F(\theta_{t+1}) \approx -\alpha_t \nabla F(\theta_t)^\top d_t \in \text{int}(C_A)$. We term such directions the general $C_A$-improving directions. The cone of $C_A$-improving directions at $x$ is

$$D_{C_A} = \{d \in \mathbb{R}^q \mid \nabla F(\theta)^\top d \in -\text{int}(C_A)\}. \tag{D.8}$$

When $A = I_M$, they are common descent directions.

**Feasible directions.** Similar to the concept in constrained single objective optimization, the feasible directions are those that ensure $F(\theta_t + \alpha_t d_t) \in S$. We rewrite problem (PMOL) with explicit $C_A$-induced partial ordering as

$$\min{}_{C_A} \ F(\theta) \ \text{s.t.} \ G(\theta) \leq 0, \ H(\theta) = 0. \tag{PMOL}$$

where $G : \mathbb{R}^q \to \mathbb{R}^{M_g}, H : \mathbb{R}^q \to \mathbb{R}^{M_h}$ are linear functions of $F$, and are differentiable. Let $I = \{i \mid G_i(\theta) = 0\}$ be the index set of the active inequality constraints in $G(\theta)$, and $G_I(\theta) = [\cdots, G_i(\theta), \cdots]^\top$ for $i \in I$. A subset of the feasible directions described by the gradients of the equality and active inequality constraints at $\theta$ is given by

$$D_g = \{d \in \mathbb{R}^q \mid \nabla G_I(\theta)^\top d < 0\}, \quad D_H = \{d \in \mathbb{R}^q \mid \nabla H(\theta)^\top d = 0\}. \tag{D.9}$$

A necessary optimality condition is that there exists no feasible and improving directions at $\theta$, i.e., $D_{C_A} \cap D_g \cap D_h = \emptyset$. An algebraic description of the necessary optimality conditions for (PMOL) is summarized below.

**Proposition 3** (First-order necessary optimality conditions for (PMOL)). *Let $C_A := \{y \in \mathbb{R}^M \mid Ay \geq 0\}$ that satisfies $\text{int}(C_A) \neq \emptyset$. If $\bar\theta$ solves (PMOL) locally, then there exists $\lambda_f \in \mathbb{R}_+^M$, $\lambda_g \in \mathbb{R}_+^{M_g}$, $[\lambda_f; \lambda_g] \neq 0$, and $\lambda_h \in \mathbb{R}^{M_h}$ that*

$$\nabla F(\bar\theta)A^\top \lambda_f + \nabla G(\bar\theta)\lambda_g + \nabla H(\bar\theta)\lambda_h = 0, \ \text{ and } \ \lambda_g^\top [-G(\bar\theta)]_+ = 0 \tag{D.10}$$

*Proof of Proposition 3.* The geometric description $D_{C_A} \cap D_g \cap D_h = \emptyset$ is equivalent to that the linear system below w.r.t. $d$ is inconsistent

$$\begin{bmatrix} A\nabla F(\bar\theta)^\top \\ \nabla G_I(\bar\theta)^\top \end{bmatrix} d < 0 \ \text{ and } \ \nabla H(\bar\theta)^\top d = 0. \tag{D.11}$$

By the Motzkin's transposition theorem, system (D.11) being inconsistent is equivalent to that the following linear system w.r.t. $p, \lambda_h$ has a solution with $p \gneq 0$

$$\begin{bmatrix} \nabla F(\bar\theta)A^\top & \nabla G_I(\bar\theta) \end{bmatrix} p + \nabla H(\bar\theta)\lambda_h = 0. \tag{D.12}$$

Letting $p = [\lambda_f; \lambda_{g,I}]$, where $\lambda_{g,I} = [\cdots; \lambda_{g,i}; \cdots], i \in I$, and $\lambda_{g,i'} = 0$, for all $i' \notin I$ completes the proof. □

**Remark 5.** *Notice that, Proposition 3 provides a Fritz John (FJ)-type first-order necessary optimality condition, which has been discussed in prior works such as [57, Theorem 1.2] with additional variational inequality constraints, and [23, Section 3, (2)-(5)] with inequality constraints only. We provide the derivation for our problem here for completeness. In the FJ-type necessary optimality condition, the multiplier $\lambda_f$ associated with the objective $F(\theta)$ can be zero if $|I| \geq 1$, which is undesirable. We need additional constraint qualifications to ensure the condition in (D.10) with $\lambda_f \neq 0$, i.e., the KKT condition, is also a necessary optimality condition. This is equivalent to $\mu_0 = 1$, and without considering the variational inequality constraints in [57, Theorem 1.2]. The constraint qualification is discussed in detail in Appendix D.3.2.*

### D.3 Properties of PMOL

In this section, we discuss the properties of PMOL and their proofs. These include the properties of the subprogram in Lemma 1, and the calmness CQ of PMOL in Lemma 2.

### D.3.1 Proof of Lemma 1: properties of the subprogram

**Lemma 6** (Additional properties of the subprogram). *For the subprogram* (2.3), *the following properties hold:*
*1. The solution $d^*(\theta)$ is unique.*
*2. If $\theta$ is a local weak optimal solution with $AF(\theta) > 0$, then $d^*(\theta) = 0$, $\psi(\theta) = 0$. Otherwise, if $\theta$ is not a local weak optimal solution, then $d^*(\theta) \neq 0$, $\psi(\theta) < 0$, and when $\theta$ is feasible,*

$$2\psi(\theta) \leq -\|d^*(\theta)\|^2 < 0. \tag{D.13}$$

*3. (Ability to escape weak optimal solutions). Let $\theta$ be a weak optimal solution, with $(AF(\theta))_m = 0$ for some $m \in [M]$. If there exists feasible and non-strictly improving directions at $\theta$ with $A\nabla F(\theta)^\top d \lesseqgtr 0$, then $d^*(\theta) \neq 0$, $\psi(\theta) < 0$. Otherwise, if there exists no feasible and non-strictly improving directions at $\theta$ with $A\nabla F(\theta)^\top d \lesseqgtr 0$, then $d^*(\theta) = 0$, $\psi(\theta) = 0$.*
*4. (Scale invariance) Suppose there are only equality constraints, i.e., $M_g = 0$, and $M_h = M - 1$, $B_h$ is full row rank and is selected such that $B_h(F(\theta_1) - F(\theta_2)) = 0$ with $F(\theta_1), F(\theta_2)$ being two different reference points in the objective space. For all $\theta \in \mathbb{R}^q$ that are feasible, i.e., $H(\theta) = 0$, when $A = I$, the normalized solution $d^*(\theta)/\|d^*(\theta)\|$ does not change when the objective $F(\theta)$ is scaled by an arbitrary positive diagonal matrix.*

*Proof of Lemma 6.* For **Property-1**, the uniqueness of $d^*(\theta)$ follows from the strict convexity of the objective function w.r.t. the direction $d$.

For **Property-2**, in the first case if $\theta$ is a local optimal solution, by definition, there exists no feasible and improving directions $d$ such that $A\nabla F(\theta)^\top d < 0$. Let $\Omega_d(\theta)$ be the set of $d \in \mathbb{R}^q$ that satisfy the constraints in (2.1), i.e.,

$$\Omega_d(\theta) := \{d \in \mathbb{R}^q \mid B_g \nabla F(\theta)^\top d + c_g G(\theta) \leq 0, B_h \nabla F(\theta)^\top d + c_h H(\theta) = 0\}. \tag{D.14}$$

Then, since $AF(\theta) > 0$, for all $d \in \Omega_d(\theta)$,

$$\max_{m \in [M]} (A\nabla F(\theta)^\top d)_m \geq 0 \tag{D.15}$$

$$\text{and} \quad \max_{m \in [M]} (A\nabla F(\theta)^\top d)_m / (AF(\theta))_m \geq 0. \tag{D.16}$$

And since $AF(\theta) > 0$, it holds that

$$\psi(\theta) := \min_{(d,c) \in \Omega_d(\theta) \times \mathbb{R}} c + \frac{1}{2}\|d\|^2$$

$$= \min_{d \in \Omega_d(\theta)} \max_{m \in [M]} (A\nabla F(\theta)^\top d)_m (\mathbf{1}^\top AF(\theta))/(AF(\theta))_m + \frac{1}{2}\|d\|^2 \geq 0 \tag{D.17}$$

with $\psi(\theta) = 0$ attainable by taking $d = 0 \in \Omega_d(\theta)$. The first case of Property-2 is proved.

In the second case, if $\theta$ is not a local weak optimal solution, then there exists $d \in \Omega_d(\theta)$ such that $A\nabla F(\theta)^\top d < 0$. Taking $\sigma = -\max_{m \in [M]}(A\nabla F(\theta)^\top d)_m(\mathbf{1}^\top AF(\theta))/((AF(\theta))_m\|d\|^2)$, and $d_\sigma = \sigma d$, then

$$\psi(\theta) := \min_{(d,c) \in \Omega_d(\theta) \times \mathbb{R}} c + \frac{1}{2}\|d\|^2$$

$$= \min_{d \in \Omega_d(\theta)} \max_{m \in [M]} (A\nabla F(\theta)^\top d)_m (\mathbf{1}^\top AF(\theta))/(AF(\theta))_m + \frac{1}{2}\|d\|^2$$

$$= \max_{m \in [M]} (A\nabla F(\theta)^\top d^*(\theta))_m (\mathbf{1}^\top AF(\theta))/(AF(\theta))_m + \frac{1}{2}\|d^*(\theta)\|^2$$

$$< \max_{m \in [M]} (A\nabla F(\theta)^\top d_\sigma)_m (\mathbf{1}^\top AF(\theta))/(AF(\theta))_m + \frac{1}{2}\|d_\sigma\|^2$$

$$= \sigma \max_{m \in [M]} (A\nabla F(\theta)^\top d)_m (\mathbf{1}^\top AF(\theta))/(AF(\theta))_m + \frac{1}{2}\sigma^2\|d\|^2 = -\frac{1}{2}\sigma^2\|d\|^2 < 0. \tag{D.18}$$

Thus $d^*(\theta) \neq 0$. Recall that

$$d^*(\theta) = -\nabla F(\theta)\left(A^\top \lambda_f^* + B_g^\top \lambda_g^* + B_h^\top \lambda_h^*\right) \tag{D.19}$$

where by the feasibility and optimality conditions,

$$\lambda_h^{*\top}\left(B_h\nabla F(\theta)^\top d^*(\theta)+c_hH(\theta)\right)=0, \tag{D.20a}$$

$$\lambda_g^{*\top}\left(B_g\nabla F(\theta)^\top d^*(\theta)+c_gG(\theta)\right)=0, \tag{D.20b}$$

$$\lambda_f^{*\top}\left(A\nabla F(\theta)^\top d^*(\theta)-c^*(\mathbf{1}_M^\top AF(\theta))^{-1}AF(\theta)\right)=0. \tag{D.20c}$$

Combining the above with (D.19), we have

$$
\begin{aligned}
\|d^*(\theta)\|^2 &= -d^*(\theta)^\top\nabla F(\theta)\left(A^\top\lambda_f^*+B_g^\top\lambda_g^*+B_h^\top\lambda_h^*\right)\\
&= -d^*(\theta)^\top\nabla F(\theta)A^\top\lambda_f^*+c_h\lambda_h^{*\top}H(\theta)+c_g\lambda_g^{*\top}G(\theta)\\
&\le -c^*(\theta)(\mathbf{1}^\top AF(\theta))^{-1}\lambda_f^{*\top}AF(\theta)=-c^*(\theta)
\end{aligned}
\tag{D.21}
$$

where the last inequality uses the fact that $\theta$ is feasible, and $G(\theta)\le 0$, $H(\theta)=0$.

Then it holds that

$$2\psi(\theta)=2c^*(\theta)+\|d^*(\theta)\|^2\le-\|d^*(\theta)\|^2<0. \tag{D.22}$$

Therefore, Property-2 holds.

For **Property-3**, let $I\subseteq[M]$ be the set such that $(AF(\theta))_m=0$ for all $m\in I$, then (2.1) is equivalent to

$$
\begin{aligned}
\psi(\theta)=\min_{(d,c)\in\mathbb{R}^q\times\mathbb{R}}\quad &c+\frac{1}{2}\|d\|^2 && \text{SP1w}\\
\text{s.t. }\ &(A\nabla F(\theta)^\top d)_m-c(\mathbf{1}^\top AF(\theta))^{-1}(AF(\theta))_m\le 0,\ \text{ for all }m\in[M]\setminus I\\
&(A\nabla F(\theta)^\top d)_m\le 0,\ \text{ for all }m\in I\\
&B_g\nabla F(\theta)^\top d+c_gG(\theta)\le 0\\
&B_h\nabla F(\theta)^\top d+c_hH(\theta)=0
\end{aligned}
$$

In the first case, if there exists feasible and non-strictly improving directions at $\theta$ with $A\nabla F(\theta)^\top d\lneq 0$, then such $d\ne 0$, $d\in\Omega_d$. Following similar arguments as (D.18) by taking $\sigma=-\max_{m\in[M]\setminus I}(A\nabla F(\theta)^\top d)_m(\mathbf{1}^\top AF(\theta))/((AF(\theta))_m\|d\|^2)$, and $d_\sigma=\sigma d$, then

$$
\begin{aligned}
\psi(\theta)&:=\min_{(d,c)\in\Omega_d(\theta)\times\mathbb{R}}c+\frac{1}{2}\|d\|^2\\
&=\min_{d\in\Omega_d(\theta)}\max_{m\in[M]\setminus I}(A\nabla F(\theta)^\top d)_m(\mathbf{1}^\top AF(\theta))/(AF(\theta))_m+\frac{1}{2}\|d\|^2\\
&<\max_{m\in[M]\setminus I}(A\nabla F(\theta)^\top d_\sigma)_m(\mathbf{1}^\top AF(\theta))/(AF(\theta))_m+\frac{1}{2}\|d_\sigma\|^2\\
&=\sigma\max_{m\in[M]\setminus I}(A\nabla F(\theta)^\top d)_m(\mathbf{1}^\top AF(\theta))/(AF(\theta))_m+\frac{1}{2}\sigma^2\|d\|^2=-\frac{1}{2}\sigma^2\|d\|^2<0.
\end{aligned}
\tag{D.23}
$$

And the corresponding $d^*(\theta)\ne 0$.

In the second case, if there exists no feasible and non-strictly improving directions at $\theta$, then for all $d\in\Omega_d(\theta)$,

$$\max_{m\in[M]\setminus I}(A\nabla F(\theta)^\top d)_m\ge 0 \tag{D.24}$$

$$\text{and}\quad\max_{m\in[M]\setminus I}(A\nabla F(\theta)^\top d)_m/(AF(\theta))_m\ge 0. \tag{D.25}$$

And since $(AF(\theta))_m>0$ for all $m\in[M]\setminus I$, it holds that

$$
\begin{aligned}
\psi(\theta)&:=\min_{(d,c)\in\Omega_d(\theta)\times\mathbb{R}}c+\frac{1}{2}\|d\|^2\\
&=\min_{d\in\Omega_d(\theta)}\max_{m\in[M]\setminus I}(A\nabla F(\theta)^\top d)_m(\mathbf{1}^\top AF(\theta))/(AF(\theta))_m+\frac{1}{2}\|d\|^2\ge 0
\end{aligned}
\tag{D.26}
$$

with $\psi(\theta) = 0$ if and only if $d = 0 \in \Omega_d(\theta)$.

Combining the above arguments, Property-3 is proved.

For **Property-4**, let $d^*(\theta)$ be the solution to the original problem (2.1) without inequality constraints. Using the fact that $H(\theta) = 0$, and letting $\lambda = A^\top \lambda_f + B_h^\top \lambda_h = \lambda_f + B_h^\top \lambda_h$, then the original dual problem can be written as

$$d^*(\theta) = -\nabla F(\theta)\lambda^*$$
$$\text{s.t. } \lambda^* \in \operatorname*{arg\,min}_{\lambda \in \Omega_{\tilde{\lambda}}(\theta)} \varphi(\lambda; \theta) := \frac{1}{2}\|\nabla F(\theta)\lambda\|^2 \tag{D.27}$$

where $\Omega_{\tilde{\lambda}}(\theta) = \left(\Omega_{\lambda_f}(\theta)\right) + B_h^\top\left(\mathbb{R}^{M_h}\right)$, and $\Omega_{\lambda_f}(\theta) = \{\lambda_f \in \mathbb{R}_+^M \mid {\lambda_f}^\top F(\theta) = \mathbf{1}^\top F(\theta)\}$.

Suppose the objective is scaled by a positive diagonal matrix $\Lambda \in \mathbb{R}^{M \times M}$, then the scaled subprogram has a dual given by

$$d^*(\theta) = -\nabla F(\theta)\Lambda\lambda^*$$
$$\text{s.t. } \lambda^* \in \operatorname*{arg\,min}_{\lambda \in \Omega_{\tilde{\lambda}}(\theta;\Lambda)} \varphi(\lambda; \theta) := \frac{1}{2}\|\nabla F(\theta)\Lambda\lambda\|^2 \tag{D.28}$$

where $\Omega_{\tilde{\lambda}}(\theta;\Lambda) = \left(\Omega_{\lambda_f}(\theta;\Lambda)\right) + {B_h'}^\top\left(\mathbb{R}^{M_h}\right)$, and $\Omega_{\lambda_f}(\theta;\Lambda) = \{\lambda_f \in \mathbb{R}_+^M \mid {\lambda_f}^\top \Lambda F(\theta) = \mathbf{1}^\top \Lambda F(\theta)\}$. Letting $\lambda' = \Lambda\lambda$, then

$$d^*(\theta) = -\nabla F(\theta)\lambda'^*$$
$$\text{s.t. } \lambda'^* \in \operatorname*{arg\,min}_{\lambda' \in \Omega_{\tilde{\lambda}'}(\theta;\Lambda)} \varphi(\lambda; \theta) := \frac{1}{2}\|\nabla F(\theta)\lambda'\|^2 \tag{D.29}$$

where $\Omega_{\tilde{\lambda}'}(\theta;\Lambda) = \Lambda\left(\Omega_{\lambda_f}(\theta;\Lambda)\right) + \Lambda {B_h'}^\top\left(\mathbb{R}^{M_h}\right)$. The set $\Lambda\left(\Omega_{\lambda_f}(\theta;\Lambda)\right)$ can be written as

$$\Lambda\left(\Omega_{\lambda_f}(\theta;\Lambda)\right) = \{\Lambda\lambda_f \mid \lambda_f \in \mathbb{R}_+^M, {\lambda_f}^\top \Lambda F(\theta) = \mathbf{1}^\top \Lambda F(\theta)\}$$
$$= \{\lambda_f' \in \mathbb{R}_+^M \mid F(\theta)^\top \lambda_f' = \mathbf{1}^\top \Lambda F(\theta)\}. \tag{D.30}$$

Notice that,

$$F(\theta)^\top \lambda_f' = \mathbf{1}^\top \Lambda F(\theta) = \mathbf{1}^\top F(\theta)c_s \tag{D.31}$$

where $c_s = \mathbf{1}^\top \Lambda F(\theta)/(\mathbf{1}^\top F(\theta))$. Therefore, $\Lambda\left(\Omega_{\lambda_f}(\theta;\Lambda)\right) = c_s\left(\Omega_{\lambda_f}(\theta)\right)$.

Also note that, $B_h \in \mathbb{R}^{(M-1)\times M}$ is full row rank, and is selected based on $F(\theta)$, which satisfies

$$B_h(F(\theta_1) - F(\theta_2)) = 0 \tag{D.32}$$

where $F(\theta_1), F(\theta_2)$ are two reference points which fully defines the kernel of $B_h$. Similarly, when $F(\theta)$ is scaled by $\Lambda$, the corresponding $B_h'$ satisfies

$$B_h'\Lambda(F(\theta_1) - F(\theta_2)) = 0. \tag{D.33}$$

This further implies

$${B_h'}^\top(\mathbb{R}^{M_h}) = \operatorname{range}(\Lambda {B_h'}^\top) = \ker(B_h'\Lambda)^\perp = \ker(B_h)^\perp = B_h(\mathbb{R}^{M_h}) = c_s B_h(\mathbb{R}^{M_h}). \tag{D.34}$$

Combining with $\Lambda\left(\Omega_{\lambda_f}(\theta;\Lambda)\right) = c_s\left(\Omega_{\lambda_f}(\theta)\right)$, it holds that

$$\Omega_{\tilde{\lambda}'}(\theta;\Lambda) = c_s\Omega_{\tilde{\lambda}}(\theta). \tag{D.35}$$

Therefore, the solution of $\tilde{\lambda}$ and $\lambda'$ is only subject to a scaling factor, which does not change the direction of $d^*(\theta)$. This proves Property-4, the scale invariance. $\qquad\square$

**Remark 7.** *Note that, Property 3, the ability to escape weak optimal solutions, and Property 4, the scale invariance, come from the subprogram design that is adaptive to the objectives. For the simplified subprogram that is not adaptive to the objectives, these two properties no longer hold, but Properties 1 and 2 still hold.*

### D.3.2 Proof of Lemma 2: calmness of PMOL

**Example 1.** *Let $F : \mathbb{R}^q \to \mathbb{R}^2$. Consider the problem below as a special case of (PMOL), given by*

$$\min_{\mathbb{R}^2_+} F(\theta) \text{ s.t. } f_2(\theta) = \min f_2(\theta). \tag{D.36}$$

*For $\bar{\theta} = \arg\min_{\theta \in \mathbb{R}^q} f_2(\theta)$, we have $\nabla f_2(\bar{\theta}) = 0$, and $\bar{\theta}$ satisfies (D.10) with $\lambda = [0,1]^\top \neq 0$ and $\lambda_h = 1$. However, $\nabla H(\bar{\theta}) = \nabla f_2(\bar{\theta}) = 0$ violates the LICQ, the Slater's CQ, and the MFCQ.*

Below we restate the definition of the Calmness condition for PMOL [57], which generalizes the calmness condition in single-objective optimization.

**Definition 10** (Calmness condition for PMOL [57, Restatement of Definition 4.5]). *Let $\bar{\theta}$ be a local solution to (PMOL). We say the PMOL problem satisfies the calmness condition at $\bar{\theta}$ provided that there exists $\epsilon > 0$ and a Lipschitz function $\phi : \mathbb{R}^{M_g + M_h} \to \mathbb{R}^M$ satisfying $\phi(0,0) = 0$ such that there exists no $(\theta, p, q) \in [(\bar{\theta}, 0, 0) + \epsilon \mathcal{B}]/\{(\bar{\theta}, 0, 0)\}$ satisfying*

$$G(\theta) + p \leq 0, \tag{D.37a}$$
$$H(\theta) + q = 0, \tag{D.37b}$$
$$F(\theta) - F(\bar{\theta}) + \phi(p, q) \in -\text{int}(C_A). \tag{D.37c}$$

Our proof relies on the following relative form of Hoffman error bound, which bounds the distance of a point to a nonempty solution set defined by constraints by a measure of the constraint violation of the point.

**Lemma 8** (Relative form of Hoffman error bound [48, Proposition 5]). *Given $B_h \in \mathbb{R}^{k_H \times M}, b_h \in \mathbb{R}^{k_H}, B_g \in \mathbb{R}^{k_G \times M}, b_g \in \mathbb{R}^{k_G}$, define $\Sigma(p, q) := \{y \in \mathbb{R}^M \mid B_g y + b_g \leq p, B_h y + b_h = q\}$, and $\text{dom} \Sigma := \{(p, q) \mid \Sigma(p, q) \neq \emptyset\}$. Let $\Omega_R \subseteq \mathbb{R}^M$ be a reference polyhedron (e.g., one defined by the intersection of half-spaces). Then for all $u \in \Omega_R$, and $(p, q) \in \text{dom} \Sigma$, there exists a relative Hoffman constant $c_{\text{hof}}$ depending only on $B_g, B_h, \Omega_R$ such that*

$$\text{dist}(u, \Sigma(p, q) \cap \Omega_R) \leq c_{\text{hof}}(B_g, B_h \mid \Omega_R) \left\| \begin{bmatrix} (B_g u + b_g - p)_+ \\ B_h u + b_h - q \end{bmatrix} \right\| \tag{D.38}$$

*where $(B_g u + b_g - p)_+ := \max\{0, B_g u + b_g - p\}$ which replaces each negative component of $B_g u + b_g - p$ by zero, and $\text{dist}(u, \Omega) := \inf_{u' \in \Omega} \|u - u'\|$.*

*Proof of Lemma 2.* We first construct $\phi(p, q) = \overline{c_{\text{hof}}} \|[p^\top, q^\top]^\top\| A^{-1} \mathbf{1}_M$, where $\overline{c_{\text{hof}}}$ is the Hoffman constant upper bound in Lemma 8. Then $\phi(0, 0) = 0$, and $\phi(p, q)$ is Lipschitz because

$$\|\phi(p, q) - \phi(p', q')\| \leq \overline{c_{\text{hof}}} M \|A^{-1}\| \left| \left\| \begin{bmatrix} p \\ q \end{bmatrix} \right\| - \left\| \begin{bmatrix} p' \\ q' \end{bmatrix} \right\| \right|$$
$$\leq \overline{c_{\text{hof}}} M \|A^{-1}\| \left\| \begin{bmatrix} p - p' \\ q - q' \end{bmatrix} \right\|. \tag{D.39}$$

Next we prove the PMOL calmness condition holds by contradiction. Suppose for every $\epsilon > 0$, there exists $(\hat{\theta}, p, q) \in [(\bar{\theta}, 0, 0) + \epsilon \mathcal{B}]/\{(\bar{\theta}, 0, 0)\}$ satisfying (D.37).

Define $\Omega_{F_1} := \{F(\theta) \in \Sigma(0, 0) \mid \theta \in \mathbb{R}^q\} \neq \emptyset$, there exists $\tilde{\theta} \in \mathbb{R}^q$ such that $F(\tilde{\theta}) \in \Omega_{F_1}$ and $\|F(\tilde{\theta})\| < \infty$. We then consider the following two cases:
*Case 1:* $F(\hat{\theta}) \in \Sigma(0, 0)$. In this case, $(\hat{\theta}, p, q) = (\hat{\theta}, 0, 0) \neq (\bar{\theta}, 0, 0)$, thus $\hat{\theta} \neq \bar{\theta}$. Take $\tilde{\theta} = \hat{\theta} \neq \bar{\theta}$.
*Case 2:* $F(\hat{\theta}) \notin \Sigma(0, 0)$. Take $\tilde{\theta}$ such that $F(\tilde{\theta}) \in \Omega_{F_1}$, then $F(\tilde{\theta}) \neq F(\hat{\theta})$.

In both cases, let $\Omega_R$ be the convex hull of $\{F(\hat{\theta}), F(\tilde{\theta})\}$, i.e., $\Omega_R = \text{conv}(\{F(\tilde{\theta}), F(\hat{\theta})\})$. Then $\Omega_R$ is a line segment (or reduces to a point in *case 1*), thus a polyhedron. Since $\Sigma(0, 0)$ is a line, $F(\tilde{\theta}) \in \Sigma(0, 0) \cap \Omega_R$, thus $\Sigma(0, 0) \cap \Omega_R = \Omega_R = \{F(\tilde{\theta})\}$ in *case 1*, and $\Sigma(0, 0) \cap \Omega_R = \{F(\tilde{\theta})\}$ in *case 2*. Therefore, in both cases,

$$\|F(\tilde{\theta}) - F(\hat{\theta})\| = \text{dist}(F(\hat{\theta}), \Sigma(0, 0) \cap \Omega_R) \tag{D.40}$$

where $\text{dist}(F, \Omega) := \inf_{F' \in \Omega} \|F - F'\|$.

We also have

$$\text{dist}(F(\hat{\theta}), \Sigma(0,0) \cap \Omega_R) \overset{(a)}{\leq} c_{\text{hof}}(\Omega_R) \left\| \begin{bmatrix} (B_g F(\hat{\theta}) + b_g)_+ \\ B_h F(\hat{\theta}) + b_h \end{bmatrix} \right\|$$

$$\overset{(b)}{\leq} \overline{c_{\text{hof}}} \left\| \begin{bmatrix} (-p)_+ \\ -q \end{bmatrix} \right\| \leq \overline{c_{\text{hof}}} \left\| \begin{bmatrix} p \\ q \end{bmatrix} \right\| \tag{D.41}$$

where $(a)$ follows from Lemma 8; $(b)$ follows from (D.37) that $0 \leq (B_g F(\hat{\theta}) + b_g)_+ \leq (-p)_+$, $B_h F(\hat{\theta}) + b_h = -q$, and that $c_{\text{hof}}(\Omega_R) \leq \overline{c_{\text{hof}}}$ for different bounded $\Omega_R$. Multiplying $\|A\|\mathbf{1}_M$ on both sides of the above inequality yields

$$\|A\|\text{dist}(F(\hat{\theta}), \Sigma(0,0) \cap \Omega_R)\mathbf{1}_M \leq A\phi(p,q). \tag{D.42}$$

It can then be derived that

$$AF(\tilde{\theta}) - AF(\hat{\theta}) \leq \|AF(\tilde{\theta}) - AF(\hat{\theta})\|\mathbf{1}_M \leq \|A\|\|F(\tilde{\theta}) - F(\hat{\theta})\|\mathbf{1}_M$$

$$\leq \|A\|\text{dist}(F(\hat{\theta}), \Sigma(0,0) \cap \Omega_R)\mathbf{1}_M \leq A\phi(p,q). \tag{D.43}$$

By rearranging the above inequality and applying (D.37c), we have that

$$AF(\tilde{\theta}) \leq AF(\hat{\theta}) + A\phi(p,q) < AF(\bar{\theta}) \tag{D.44}$$

which contradicts to that $\bar{\theta}$ is a global solution to (PMOL).

Therefore, the PMOL calmness condition in Definition 10 is satisfied. $\square$

# E   Proof of Theorem 1: convergence of Algorithm 1

Recall that, we let $\lambda = [\lambda_f; \lambda_g; \lambda_h] \in \mathbb{R}^{M+M_g+M_h}$, $A_{ag} = [A; B_g; B_h] \in \mathbb{R}^{(M+M_g+M_h)\times M}$, and use the following concise notation

$$d^*(\theta) = -\nabla F(\theta)A_{ag}^\top \lambda^*(\theta)$$

$$\text{s.t. } \lambda^*(\theta) \in \underset{\lambda \in \Omega_\lambda(\theta)}{\arg\min} \varphi(\lambda; \theta) := \frac{1}{2}\|\nabla F(\theta)A_{ag}^\top \lambda\|^2 - c_g \lambda_g^\top G(\theta) - c_h \lambda_h^\top H(\theta) \tag{E.1}$$

where $\Omega_\lambda(\theta) = \Omega_{\lambda_f}(\theta) \times \mathbb{R}_+^{M_g} \times \mathbb{R}^{M_h}$, and $\Omega_{\lambda_f}(\theta) = \{\lambda_f \in \mathbb{R}_+^M \mid {\lambda_f}^\top AF(\theta) = \mathbf{1}^\top AF(\theta)\}$.

In the following discussion in this section, we first present the supporting lemmas and their proofs, then provide the proof of Theorem 1.

## E.1   Auxiliary lemmas

Lemma 9 is a result from the smoothness of $F(\theta)$, and thus the smoothness of $G(\theta)$ and $H(\theta)$, whose smoothness constants depend on $B_g$ and $B_h$, respectively.

**Lemma 9.** *Suppose Assumptions 1, 2 hold. Then for all $\theta, \theta' \in \mathbb{R}^q$, and all $\lambda_f \in \mathbb{R}^M$, we have*

$$\lambda_f^\top AF(\theta_{t+1}) - \lambda_f^\top AF(\theta_t) \leq \alpha_t \lambda_f^\top A\nabla F(\theta_t)^\top d_t + \frac{\ell_{f,1}\|A^\top \lambda_f\|_1}{2}\alpha_t^2\|d_t\|^2 \tag{E.2}$$

$$G(\theta_{t+1}) - G(\theta_t) \leq \alpha_t \nabla G(\theta_t)^\top d_t + \frac{\ell_{f,1}}{2}\alpha_t^2\|B_g^\top\|_{\infty,1}\|d_t\|^2\mathbf{1} \tag{E.3}$$

$$H(\theta_{t+1}) - H(\theta_t) \leq \alpha_t \nabla H(\theta_t)^\top d_t + \frac{\ell_{f,1}}{2}\alpha_t^2\|B_h^\top\|_{\infty,1}\|d_t\|^2\mathbf{1}. \tag{E.4}$$

*Proof.* By Assumption 2, it holds that $\lambda_f^\top AF(\theta)$ is $\|A^\top \lambda_f\|_1 \ell_{f,1}$-smooth. By the definition of smoothness, we have

$$\lambda_f^\top AF(\theta_{t+1}) \leq \lambda_f^\top AF(\theta_t) + \alpha_t \lambda_f^\top A\nabla F(\theta_t)^\top d_t + \frac{\ell_{f,1}\|A^\top \lambda_f\|_1}{2}\alpha_t^2\|d_t\|^2. \tag{E.5}$$

Let $B_{g,m}$ and $B_{h,m}$ be the $m$-th row of $B_g$ and $B_h$, respectively, then by the $\ell_{f,1}$-smoothness of $F(\theta)$, $B_{g,m}F(\theta)$ is $\ell_{f,1}\|B_{g,m}\|_1$-smooth for all $m \in [M_g]$. Also because $\|B_{g,m}\|_1 \leq \|B_g^\top\|_{\infty,1}$ where $\|B_g^\top\|_{\infty,1} = \max_{m \in M_g} \|\|B_{g,m}\|_1\|$, $g_m(\theta)$ is $\ell_{f,1}\|B_g^\top\|_{\infty,1}$-smooth for all $m \in [M_g]$. By the definition of smoothness, it holds that

$$G(\theta_{t+1}) - G(\theta_t) \leq \alpha_t \nabla G(\theta_t)^\top d_t + \frac{\ell_{f,1}}{2}\alpha_t^2\|B_g^\top\|_{\infty,1}\|d_t\|^2\mathbf{1}. \tag{E.6}$$

Following similar arguments as the above for $G(\theta)$, (E.4) can be proved. $\qquad\square$

**Lemma 10.** *For the subprogram* (2.3) *or equivalently* (E.1), *it holds that for any* $\lambda \in \Omega_\lambda(\theta)$,

$$\langle \nabla F(\theta)A_{ag}^\top\lambda, \nabla F(\theta)A_{ag}^\top\lambda^*(\theta)\rangle - [0^\top, c_gG(\theta)^\top, c_hH(\theta)^\top](\lambda - \lambda^*(\theta)) \geq \|\nabla F(\theta)A_{ag}^\top\lambda^*(\theta)\|^2. \tag{E.7}$$

*Proof of Lemma 10.* Since $\varphi(\lambda; \theta)$ is a convex function w.r.t. $\lambda$, by the first order optimality condition, it holds that for all $\lambda \in \Omega_\lambda(\theta)$

$$\langle \nabla_\lambda\varphi(\lambda^*(\theta); \theta), \lambda - \lambda^*(\theta)\rangle \geq 0 \tag{E.8}$$

which can be further written as

$$\lambda^\top A_{ag}\nabla F(\theta)^\top\nabla F(\theta)A_{ag}^\top\lambda^*(\theta) - [0^\top, c_gG(\theta)^\top, c_hH(\theta)^\top](\lambda - \lambda^*(\theta)) \geq \|\nabla F(\theta)A_{ag}^\top\lambda^*(\theta)\|^2. \tag{E.9}$$

This completes the proof. $\qquad\square$

We next prove Lemma 11, which can be viewed as a descent lemma for $[G(\theta)]_+$ and $|H(\theta)|_{ab}$ based on the smoothness of $G(\theta)$ and $H(\theta)$, as well as proper hyperparameter choices. This is crucial for proving the convergence result in Theorem 1. One key technical challenge in proving the lemma is that even though $G(\theta)$ and $H(\theta)$ are smooth, $[G(\theta)]_+$ and $|H(\theta)|_{ab}$ are not. We address this challenge by exploiting the fact that $\nabla G(\theta_t)^\top d^*(\theta_t) \leq -c_gG(\theta_t)$ and $\nabla H(\theta_t)^\top d^*(\theta_t) = -c_gH(\theta_t)$, as well as choosing $\alpha_t$ properly depending on $c_g$ and $c_h$.

**Lemma 11.** *Let* $\epsilon \geq 0$ *be a constant. Define* $[y]_+ := \max\{y, 0\}$ *which replaces each negative component of* $y$ *by zero, and* $|y|_{ab}$ *replaces each component of* $y$ *by its absolute value. Let* $\{\theta_t\}$ *be the sequence produced by Algorithm 1 with the update* $\theta_{t+1} = \theta_t + \alpha_t d_t$, *where* $d_t$ *satisfies the constraints of the subprogram* (2.1) *up to an error of* $\epsilon$, *i.e.,*

$$[\nabla G(\theta_t)^\top d_t + c_gG(\theta_t)]_+ \leq \epsilon\mathbf{1}, \tag{E.10}$$

$$|\nabla H(\theta_t)^\top d_t + c_hH(\theta_t)|_{ab} \leq \epsilon\mathbf{1}. \tag{E.11}$$

*If* $\alpha_t \leq \min\{c_g^{-1}, c_h^{-1}\}$, *then it holds that*

$$[G(\theta_{t+1})]_+ - [G(\theta_t)]_+ \leq -\alpha_t c_g[G(\theta_t)]_+ + \frac{\ell_{f,1}}{2}\alpha_t^2\|B_g^\top\|_{\infty,1}\|d_t\|^2\mathbf{1} + \epsilon\mathbf{1} \tag{E.12}$$

$$|H(\theta_{t+1})|_{ab} - |H(\theta_t)|_{ab} \leq -\alpha_t c_h|H(\theta_t)|_{ab} + \frac{\ell_{f,1}}{2}\alpha_t^2\|B_h^\top\|_{\infty,1}\|d_t\|^2\mathbf{1} + \epsilon\mathbf{1}. \tag{E.13}$$

*Proof.* By the smoothness of $G(\theta)$ in Lemma 9 and $\nabla G(\theta)^\top d + c_gG(\theta) \leq [\nabla G(\theta)^\top d + c_gG(\theta)]_+ \leq \epsilon\mathbf{1}$, it holds that

$$G(\theta_{t+1}) - G(\theta_t) \leq \alpha_t \nabla G(\theta_t)^\top d_t + \frac{\ell_{f,1}}{2}\alpha_t^2\|B_g^\top\|_{\infty,1}\|d_t\|^2\mathbf{1} + \epsilon\mathbf{1}$$

$$\leq -\alpha_t c_gG(\theta_t) + \frac{\ell_{f,1}}{2}\alpha_t^2\|B_g^\top\|_{\infty,1}\|d_t\|^2\mathbf{1} + \epsilon\mathbf{1}. \tag{E.14}$$

For all $m \in [M_g]$, since $G(\theta_t) \leq [G(\theta_t)]_+$, it holds that

$$g_m(\theta_{t+1}) - [g_m(\theta_t)]_+ \leq g_m(\theta_t) - [g_m(\theta_t)]_+ - \alpha_t c_gg_m(\theta_t) + \frac{\ell_{f,1}}{2}\alpha_t^2\|B_g^\top\|_{\infty,1}\|d_t\|^2 + \epsilon \tag{E.15}$$

$$\leq -[-g_m(\theta_t)]_+ - \alpha_t c_gg_m(\theta_t) + \frac{\ell_{f,1}}{2}\alpha_t^2\|B_g^\top\|_{\infty,1}\|d_t\|^2 + \epsilon. \tag{E.16}$$

It can be further derived that

$$-[-g_m(\theta_t)]_+ - \alpha_t c_g g_m(\theta_t) = \begin{cases} -\alpha_t c_g g_m(\theta_t), g_m(\theta_t) \geq 0 \\ (1 - \alpha_t c_g) g_m(\theta_t), g_m(\theta_t) < 0 \end{cases}$$
$$\leq -\alpha_t c_g [g_m(\theta_t)]_+ \tag{E.17}$$

where the last inequality holds since $1 - \alpha_t c_g \geq 0$. Plugging this inequality back into (E.16), yields that when $g_m(\theta_{t+1}) \geq 0$,

$$[g_m(\theta_{t+1})]_+ - [g_m(\theta_t)]_+ \leq -\alpha_t c_g [g_m(\theta_t)]_+ + \frac{\ell_{f,1}}{2} \alpha_t^2 \|B_g^\top\|_{\infty,1} \|d_t\|^2 + \epsilon. \tag{E.18}$$

When $g_m(\theta_{t+1}) < 0$, we have

$$[g_m(\theta_{t+1})]_+ - [g_m(\theta_t)]_+ \leq -[g_m(\theta_t)]_+ \leq -\alpha_t c_g [g_m(\theta_t)]_+$$
$$\leq -\alpha_t c_g [g_m(\theta_t)]_+ + \frac{\ell_{f,1}}{2} \alpha_t^2 \|B_g^\top\|_{\infty,1} \|d_t\|^2 + \epsilon. \tag{E.19}$$

Combining (E.18) and (E.19) proves (E.12).

By the smoothness of $H(\theta)$ and $|\nabla H(\theta)^\top d + c_h H(\theta)|_{\mathrm{ab}} \leq \epsilon \mathbf{1}$, we have

$$|H(\theta_{t+1})|_{\mathrm{ab}} \leq |H(\theta_t) - \alpha_t c_h H(\theta_t)|_{\mathrm{ab}} + \frac{\ell_{f,1}}{2} \alpha_t^2 \|B_h^\top\|_{\infty,1} \|d_t\|^2 \mathbf{1} + \epsilon \mathbf{1}$$
$$= (1 - \alpha_t c_h)|H(\theta_t)|_{\mathrm{ab}} + \frac{\ell_{f,1}}{2} \alpha_t^2 \|B_h^\top\|_{\infty,1} \|d_t\|^2 \mathbf{1} + \epsilon \mathbf{1} \tag{E.20}$$

where the last equality holds because $1 - \alpha_t c_h \geq 0$, which proves (E.13). $\qquad\square$

## E.2 Proof of Theorem 1

In this section, we prove Theorem 1. Similar to the proof techniques used in [7], we use $\lambda_f^\top A F(\theta_t)$ with a fixed $\lambda_f \in \Omega_{\lambda_f}(\theta)$ as a part of the Lyapunov function, instead of using the dynamically changing $\lambda_{f,t}$. This eliminates the need to assume the objective values are bounded above in our theorem.

*Proof of Theorem 1.* To consider both objective function minimization and constraint satisfaction, we define a Lyapunov function below with a constant vector $\lambda = (\lambda_f, \lambda_g, \lambda_h) \in \Omega_\lambda(\theta)$, where $\lambda_f = \mathbf{1}$, $\lambda_g \in \mathbb{R}_+^{M_g}$, $\lambda_h \in \mathbb{R}^{M_h}$, and $\lambda_g > \lambda_g^*(\theta_t)$, $\lambda_h > \lambda_h^*(\theta_t)$ for all $t \in [T]$.

$$\mathbb{V}_t := \underbrace{\lambda_f^\top A F(\theta_t)}_{\mathbb{V}_{f,t}} + \underbrace{\lambda_g^\top [G(\theta_t)]_+}_{\mathbb{V}_{g,t}} + \underbrace{\lambda_h^\top |H(\theta_t)|_{\mathrm{ab}}}_{\mathbb{V}_{h,t}}. \tag{E.21}$$

Note that $\mathbb{V}_t \geq 0$ for all $t$ since $A F(\theta) \geq 0, \lambda_f \geq 0$.

For notation simplicity, we let $d_t^* = d^*(\theta_t)$. From Assumption 2, the smoothness of the objectives, and Lemma 9, based on the update $\theta_{t+1} = \theta_t + \alpha_t d_t$, it holds that

$$\mathbb{V}_{f,t+1} - \mathbb{V}_{f,t} \overset{(a)}{\leq} \alpha_t \lambda_f^\top A \nabla F(\theta_t)^\top d_t + \frac{\ell_{f,1}}{2} \alpha_t^2 \|A^\top\|_{\infty,1} \|d_t\|^2 \lambda_f^\top \mathbf{1}$$

$$\overset{(b)}{\leq} \alpha_t \lambda_f^\top A \nabla F(\theta_t)^\top d_t^* + \frac{\ell_{f,1}}{2} \alpha_t^2 \|A^\top\|_{\infty,1} \|d_t^*\|^2 \lambda_f^\top \mathbf{1} + \epsilon \mathbf{1}$$

$$\overset{(c)}{\leq} -\alpha_t \|d_t^*\|^2 + \alpha_t \big( c_g \lambda_g^*(\theta_t)^\top G(\theta_t) + c_h \lambda_h^*(\theta_t)^\top H(\theta_t) \big) + \frac{\ell_{f,1}}{2} \alpha_t^2 \|A^\top \mathbf{1}\|_1 \|d_t^*\|^2 + \epsilon \mathbf{1} \tag{E.22}$$

where $(a)$ follows Lemma 9; $(b)$ follows from that $d_t$ is an $\epsilon$-optimal solution to the subprogram; $(c)$ follows from Lemma 10 with $\lambda = [\lambda_f; 0; 0] \in \Omega_\lambda(\theta)$ therein.

From Lemma 11, for $\alpha_t \leq \min\{c_g^{-1}, c_h^{-1}\}$, it holds that

$$\mathbb{V}_{g,t+1} - \mathbb{V}_{g,t} \leq -\alpha_t c_g \lambda_g^\top [G(\theta_t)]_+ + \frac{\ell_{f,1}}{2} \alpha_t^2 \|B_g^\top\|_{\infty,1} \|d_t\|^2 \lambda_g^\top \mathbf{1} + \epsilon \lambda_g^\top \mathbf{1} \tag{E.23}$$

$$\mathbb{V}_{h,t+1} - \mathbb{V}_{h,t} \leq - \alpha_t c_h \lambda_h^\top |H(\theta_t)|_{\mathrm{ab}} + \frac{\ell_{f,1}}{2}\alpha_t^2 \|B_h^\top\|_{\infty,1}\|d_t\|^2 \lambda_h^\top \mathbf{1} + \epsilon \lambda_h^\top \mathbf{1}. \qquad (\text{E.24})$$

Combining the above inequalities for $\mathbb{V}_{f,t}, \mathbb{V}_{g,t}, \mathbb{V}_{h,t}$, we have

$$\begin{aligned}
\mathbb{V}_{t+1} - \mathbb{V}_t \leq & - \alpha_t\|d_t^*\|^2 + \alpha_t\big(c_g \lambda_g^*(\theta_t)^\top G(\theta_t) + c_h \lambda_h^*(\theta_t)^\top H(\theta_t)\big) \\
& + \frac{\ell_{f,1}}{2}\alpha_t^2 \|A_{ag}^\top \lambda\|_1 \|d_t^*\|^2 - \alpha_t c_g \lambda_g^\top [G(\theta_t)]_+ - \alpha_t c_h \lambda_h^\top |H(\theta_t)|_{\mathrm{ab}} + \epsilon \lambda^\top \mathbf{1} \\
\leq & - \alpha_t\|d_t^*\|^2 - \alpha_t c_g(\lambda_g - \lambda_g^*(\theta_t))^\top [G(\theta_t)]_+ - \alpha_t c_g \lambda_g^*(\theta_t)^\top [-G(\theta_t)]_+ \\
& - \alpha_t c_h(\lambda_h - \lambda_h^*(\theta_t))^\top |H(\theta_t)|_{\mathrm{ab}} + \frac{\ell_{f,1}}{2}\alpha_t^2\|A_{ag}^\top \lambda\|_1 \|d_t^*\|^2 + \epsilon \lambda^\top \mathbf{1} \qquad (\text{E.25})
\end{aligned}$$

where the last inequality holds because $\lambda_g^*(\theta_t)^\top G(\theta_t) = \lambda_g^*(\theta_t)^\top [G(\theta_t)]_+ - \lambda_g^*(\theta_t)^\top [-G(\theta_t)]_+$.
Taking telescoping sum of the above inequality from $t = 0, \ldots, T-1$ and rearranging, we have

$$\begin{aligned}
\sum_{t=0}^{T-1} & \alpha_t\Big(1 - \frac{1}{2}\|A_{ag}^\top \lambda\|_1 \ell_{f,1}\alpha_t\Big)\|d_t^*\|^2 + \alpha_t c_g(\lambda_g - \lambda_g^*(\theta_t))^\top [G(\theta_t)]_+ + \alpha_t c_g \lambda_g^*(\theta_t)^\top [-G(\theta_t)]_+ \\
& + \alpha_t c_h(\lambda_h - \lambda_h^*(\theta_t))^\top |H(\theta_t)|_{\mathrm{ab}} \leq \mathbb{V}_0 - \mathbb{V}_T + T\epsilon \lambda^\top \mathbf{1} \leq \mathbb{V}_0 + T\epsilon\|\lambda\|_1. \qquad (\text{E.26})
\end{aligned}$$

Recall that $\alpha_t \leq 1/(\ell_{f,1}\|A_{ag}^\top \lambda\|_1)$. Plugging this into the above inequality yields

$$\begin{aligned}
\sum_{t=0}^{T-1} & \frac{1}{2}\alpha_t\|d_t^*\|^2 + \alpha_t c_g(\lambda_g - \lambda_g^*(\theta_t))^\top [G(\theta_t)]_+ + \alpha_t c_g \lambda_g^*(\theta_t)^\top [-G(\theta_t)]_+ \\
& + \alpha_t c_h(\lambda_h - \lambda_h^*(\theta_t))^\top |H(\theta_t)|_{\mathrm{ab}} \leq \mathbb{V}_0 + T\epsilon\|\lambda\|_1. \qquad (\text{E.27})
\end{aligned}$$

Taking $\alpha_t = \Theta(1)$, then

$$\begin{aligned}
\frac{1}{T}\sum_{t=0}^{T-1} & \frac{1}{2}\|d^*(\theta_t)\|^2 + c_g(\lambda_g - \lambda_g^*(\theta_t))^\top [G(\theta_t)]_+ + c_g \lambda_g^*(\theta_t)^\top [-G(\theta_t)]_+ \\
& + c_h(\lambda_h - \lambda_h^*(\theta_t))^\top |H(\theta_t)|_{\mathrm{ab}} = \mathcal{O}\Big(\frac{1}{T} + \epsilon\Big). \qquad (\text{E.28})
\end{aligned}$$

The proof is complete. $\qquad\square$

Next we show that the subprogram converges with a projected gradient descent (PGD) algorithm on $\lambda$ with $K$ iterations.

**Lemma 12** (Convergence of the subprogram with projected gradient descent). *At the $t$-th iteration, given $\theta_t$, let $\{\lambda_{t,k}\}_k$ be the sequence generated by the projected gradient descent algorithm to solve the subprogram $\min_{\lambda \in \Omega_\lambda(\theta_t)} \varphi(\lambda; \theta_t)$, then*

$$\varphi(\lambda_{t,K}; \theta_t) - \min_{\lambda \in \Omega_\lambda(\theta_t)} \varphi(\lambda; \theta_t) \leq \frac{\|\lambda_{t,0} - \lambda^*(\theta_t)\|^2}{2\gamma K}. \qquad (\text{E.29})$$

*Proof.* The result follows from the convergence result of projected gradient descent for convex objective functions. Note that at each iteration $t$, given $\theta_t$, $\Omega_\lambda(\theta_t)$ is fixed. $\qquad\square$

**Lemma 13.** *Suppose Assumption 3 holds. Due to the $\ell_{\varphi_\lambda,1}$-smoothness and the convexity of the subprogram, it holds for all $\lambda \in \Omega_\lambda(\theta)$ that*

$$\|\nabla_\lambda \varphi(\lambda; \theta) - \nabla_\lambda \varphi(\lambda^*(\theta); \theta)\|^2 \leq 2\ell_{\varphi_\lambda,1}\big(\varphi(\lambda; \theta) - \varphi(\lambda^*(\theta); \theta)\big). \qquad (\text{E.30})$$

*Proof.* Since the objectives $f_m(\theta)$ are Lipschitz continuous for all $m \in [M]$, the subprogram objective $\varphi(\lambda; \theta)$ is $\ell_{\varphi_\lambda,1}$-smooth w.r.t. $\lambda$. By Proposition 1 (b) in [58], it holds that

$$\frac{1}{2\ell_{\varphi_\lambda,1}}\|\nabla_\lambda \varphi(\lambda; \theta) - \nabla_\lambda \varphi(\lambda^*(\theta); \theta)\|^2 + \langle \nabla_\lambda \varphi(\lambda^*(\theta); \theta), \lambda - \lambda^*(\theta)\rangle \leq \varphi(\lambda; \theta) - \varphi(\lambda^*(\theta); \theta).$$
$$(\text{E.31})$$

By the convexity of $\varphi(\lambda; \theta)$ w.r.t. $\lambda$, for all $\lambda \in \Omega_\lambda(\theta)$,

$$\langle \nabla_\lambda \varphi(\lambda^*(\theta); \theta), \lambda - \lambda^*(\theta) \rangle \geq 0. \tag{E.32}$$

Combining the above two inequalities proves the result. $\qquad\square$

**Corollary 14** (Convergence of Algorithm 1 with $K$-iteration PGD for the subprogram). *Suppose Assumptions 1, 2 hold. Let $\{\theta_t\}$ be the sequence produced by Algorithm 1 with the update $\theta_{t+1} = \theta_t + \alpha_t d_t$, where $d_t$ is the $\epsilon$-optimal solution to the subprogram (2.1) obtained by $K$-iteration PGD for the subprogram on $\lambda$. Define $\lambda := (\lambda_f, \lambda_g, \lambda_h) \in \Omega_\lambda(\theta)$ with $\lambda_g \geq \lambda_g^*(\theta) + 1$, $\lambda_h \geq \lambda_h^*(\theta) + 1$ for all $\theta \in \mathbb{R}^q$. If the step size $\alpha_t \leq 1/(\ell_{f,1} \|A_{ag}^\top \lambda\|_1)$ and $\alpha_t = \Theta(1)$, then*

$$\sum_{t=0}^{T-1} \frac{1}{2} \|d_t^*\|^2 + c_g(\lambda_g - \lambda_g^*(\theta_t))^\top [G(\theta_t)]_+ + c_g \lambda_g^*(\theta_t)^\top [-G(\theta_t)]_+$$
$$+ c_h(\lambda_h - \lambda_h^*(\theta_t))^\top |H(\theta_t)|_{ab} = \mathcal{O}(1). \tag{E.33}$$

*Proof.* For $t = 0, \ldots, T-1$, we take $K = T^2$, applying Lemma 12, we have

$$\varphi(\lambda_{t,K}; \theta_t) - \min_{\lambda \in \Omega_\lambda(\theta_t)} \varphi(\lambda; \theta_t) \leq \frac{\|\lambda_{t,0} - \lambda^*(\theta_t)\|^2}{2\gamma T^2}. \tag{E.34}$$

From Lemma 13, the above inequality implies

$$\|\nabla \varphi(\lambda_t; \theta_t) - \nabla \varphi(\lambda^*(\theta_t); \theta_t)\|^2 \leq 2\ell_{\varphi_\lambda, 1}\big(\varphi(\lambda_t; \theta_t) - \varphi(\lambda^*(\theta_t); \theta_t)\big) \leq \frac{\ell_{\varphi_\lambda, 1} \|\lambda_{t-1} - \lambda^*(\theta_t)\|^2}{\gamma T^2}. \tag{E.35}$$

Plugging in the gradient $\nabla \varphi(\lambda_t; \theta_t)$, we have

$$\|A \nabla F(\theta_t)^\top (d_t - d_t^*)\|^2 + \|\nabla G(\theta_t)^\top (d_t - d_t^*)\|^2 + \|\nabla G(\theta_t)^\top (d_t - d_t^*)\|^2$$
$$\leq \frac{\ell_{\varphi_\lambda, 1} \|\lambda_{t-1} - \lambda^*(\theta_t)\|^2}{\gamma T^2} \leq \frac{4 \ell_{\varphi_\lambda, 1} c_\lambda^2}{\gamma T^2}. \tag{E.36}$$

Let $\epsilon = \frac{4 \ell_{\varphi_\lambda, 1} c_\lambda^2}{\gamma T^2}$, from Theorem 1, it holds that

$$\mathbb{V}_{t+1} - \mathbb{V}_t \leq -\alpha_t \|d_t^*\|^2 + \alpha_t c_g \big(\lambda_g^*(\theta_t) - \lambda_g\big)^\top [G(\theta_t)]_+ + \alpha_t c_g \lambda_g^*(\theta_t)^\top [-G(\theta_t)]_+$$
$$+ \alpha_t c_h \big(\lambda_h^*(\theta_t) - \lambda_h\big)^\top |H(\theta_t)|_{ab} + \epsilon^{\frac{1}{2}} + \frac{1}{2} \gamma \alpha_t \|\nabla_\lambda \varphi(\lambda_t; \theta_t)\|^2 + \frac{\ell_{f,1}}{2} \alpha_t^2 \|A_{ag}^\top \lambda\|_1 \|d_t^*\|^2. \tag{E.37}$$

Taking telescoping sum of the above inequality from $t = 0, \ldots, T-1$, rearranging, and letting $\alpha_t \leq 1/(\|\lambda\|_1 \ell_{f,1} \|A_{ag}^\top\|_{\infty,1})$, we have

$$\sum_{t=T}^{T-1} \frac{1}{2} \alpha_t \|d_t^*\|^2 + \alpha_t c_g(\lambda_g - \lambda_g^*(\theta_t))^\top [G(\theta_t)]_+ + \alpha_t c_g \lambda_g^*(\theta_t)^\top [-G(\theta_t)]_+$$
$$+ \alpha_t c_h(\lambda_h - \lambda_h^*(\theta_t))^\top |H(\theta_t)|_{ab} \leq \mathbb{V}_T + T \epsilon^{\frac{1}{2}}. \tag{E.38}$$

Letting $\alpha_t = \Theta(1), \gamma = \Theta(1)$ yields

$$\sum_{t=T}^{T-1} \frac{1}{2} \|d_t^*\|^2 + c_g(\lambda_g - \lambda_g^*(\theta_t))^\top [G(\theta_t)]_+ + c_g \lambda_g^*(\theta_t)^\top [-G(\theta_t)]_+$$
$$+ c_h(\lambda_h - \lambda_h^*(\theta_t))^\top |H(\theta_t)|_{ab} = \mathcal{O}(1). \tag{E.39}$$

The proof is complete. $\qquad\square$

# F   Proof of Theorems 2 and 3: convergence of Algorithm 2

In this section, we prove the convergence of Algorithm 2 with single-loop updates. We focus on the problem with equality constraints only, i.e., $M_g = 0$. Furthermore, we consider the simplified subprogram without adaptivity to the objectives, thus $\Omega_{\lambda_f}(\theta) = \Delta^M$.

We provide two theoretical results in Theorems 2 and 3, respectively. Specifically, Theorem 2 uses the same merit function as Theorem 1, but provides a slower convergence rate. Theorem 3 uses a different merit function, and provides a faster convergence rate than Theorem 1 under additional assumptions.

## F.1   Auxiliary lemmas

**Lemma 15** (Smoothness of $\varphi$ w.r.t. $\lambda$). *Suppose Assumptions 1 and 3 hold. $\varphi(\lambda; \theta)$ is $\ell_{\varphi_\lambda,1}$-smooth w.r.t. $\lambda$, with $\ell_{\varphi_\lambda,1} = M\|A_{ag}\|^2 \ell_f^2$.*

*Proof.* The Hessian of $\varphi(\lambda; \theta)$ w.r.t. $\lambda$ can be computed by
$$\nabla_\lambda^2 \varphi(\lambda; \theta) = A_{ag} \nabla F(\theta)^\top \nabla F(\theta) A_{ag}^\top.$$
By Assumption 3, the Lipschitz continuity of $F$, it holds that
$$\|\nabla_\lambda^2 \varphi(\lambda; \theta)\| \leq \|A_{ag} \nabla F(\theta)^\top \nabla F(\theta) A_{ag}^\top\| \leq \|\nabla F(\theta) A_{ag}^\top\|^2 \leq M\|A_{ag}\|^2 \ell_f^2.$$
The result is proved. $\qquad\square$

**Lemma 16** ($\|\nabla_{\lambda_f} \varphi(\lambda_t; \theta_t)\|$ is bounded by $\|d_t\|$). *Suppose Assumptions 1 and 3 hold. For $\{\theta_t\}$ produced by Algorithm 2, we have*
$$\|\nabla_{\lambda_f} \varphi(\lambda_t; \theta_t)\| \leq \|A^\top\|_{\infty,1} \ell_f \|d_t\|. \tag{F.1}$$

*Proof.* The gradient of $\varphi(\lambda_t; \theta_t)$ w.r.t. $\lambda_f$ can be computed by
$$\nabla_{\lambda_f} \varphi(\lambda_t; \theta_t) = A\nabla F(\theta_t)^\top \nabla F(\theta_t) A_{ag}^\top \lambda_t = -A\nabla F(\theta_t)^\top d_t. \tag{F.2}$$
By Assumption 3, it holds that
$$\|\nabla_{\lambda_f} \varphi(\lambda_t; \theta_t)\| \leq \|A^\top\|_{\infty,1} \ell_f \|d_t\|. \tag{F.3}$$
The proof is complete. $\qquad\square$

**Lemma 17.** *Let $\lambda_t = [\lambda_{f,t}; \lambda_{h,t}]$. Consider the sequence $\{\lambda_t\}_{t=1}^T$ generated by the update (3.1). Then for all $\lambda \in \Omega_\lambda(\theta_t)$ with $\lambda = (\lambda_f, \lambda_h)$, it holds that*
$$2\gamma_t \langle \lambda_{f,t} - \lambda_f, \nabla_{\lambda_f} \varphi(\lambda_t; \theta_t)\rangle \leq \|\lambda_{f,t} - \lambda_f\|^2 - \|\lambda_{f,t+1} - \lambda_f\|^2 + \gamma_t^2 \|\nabla_{\lambda_f} \varphi(\lambda_t; \theta_t)\|^2;$$
$$2\gamma_t \langle \lambda_{h,t} - \lambda_h, \nabla_{\lambda_h} \varphi(\lambda_t; \theta_t)\rangle = \|\lambda_{h,t} - \lambda_h\|^2 - \|\lambda_{h,t+1} - \lambda_h\|^2 + \gamma_t^2 \|\nabla_{\lambda_h} \varphi(\lambda_t; \theta_t)\|^2. \tag{F.4}$$

*Proof.* By the update of $\lambda_{f,t}$, and the non-expansiveness of projection, for all $\lambda_f \in \Delta^M$, we have
$$\|\lambda_{f,t+1} - \lambda_f\|^2 \leq \|\lambda_{f,t} - \gamma_t \nabla_{\lambda_f} \varphi(\lambda_t; \theta_t) - \lambda_f\|^2$$
$$= \|\lambda_{f,t} - \lambda_f\|^2 - 2\gamma_t \langle \lambda_{f,t} - \lambda_f, \nabla_{\lambda_f} \varphi(\lambda_t; \theta_t)\rangle + \gamma_t^2 \|\nabla_{\lambda_f} \varphi(\lambda_t; \theta_t)\|^2. \tag{F.5}$$
Rearranging the above inequality proves the first inequality.

By the update of $\lambda_{h,t}$, for all constant $\lambda_h \in \mathbb{R}^{M_h}$, we have
$$\|\lambda_{h,t+1} - \lambda_h\|^2 = \|(\lambda_{h,t} - \gamma_t \nabla_{\lambda_h} \varphi(\lambda_t; \theta_t)) - \lambda_h\|^2$$
$$= \|\lambda_{h,t} - \lambda_h\|^2 + \gamma_t^2 \|\nabla_{\lambda_h} \varphi(\lambda_t; \theta_t)\|^2 - 2\gamma_t \langle \lambda_{h,t} - \lambda_h, \nabla_{\lambda_h} \varphi(\lambda_t; \theta_t)\rangle. \tag{F.6}$$
Rearranging the above inequality proves the second inequality. $\qquad\square$

**Corollary 18.** *Let $\lambda_t = [\lambda_{f,t}; \lambda_{h,t}]$. Consider the sequence $\{\lambda_t\}_{t=1}^T$ generated by the update (3.1). Then for all $\lambda \in \Omega_\lambda$ with $\lambda = (\lambda_f, \lambda_h)$, it holds that*
$$2\gamma_t \big(\varphi(\lambda_t; \theta_t) - \varphi(\lambda; \theta_t)\big) \leq \|\lambda_t - \lambda\|^2 - \|\lambda_{t+1} - \lambda\|^2 + \gamma_t^2 \|\nabla_\lambda \varphi(\lambda_t; \theta_t)\|^2. \tag{F.7}$$

*Proof of Corollary 18.* The result follows from combining the two inequalities in Lemma 17, and applying the convexity property of $\varphi$ w.r.t. $\lambda$. $\qquad\square$

## F.2 Analysis with the same merit function: proof of Theorem 2

In this section, we provide analysis with the same merit function as Theorem 1. The proof follows similar ideas of the proofs of Theorem 3 (for convergence of the subprogram with the approximate single-loop update) and Theorem 5 (for convergence of the main program) in [7]. We follow the proofs in [7], as they provide, to the best of our knowledge, the fastest convergence rate guarantees for single-loop MOO algorithms under minimal assumptions.

Similar to [7], we first define the following auxiliary functions to assist our analysis. Note that the functions are only used for analysis but not for the algorithm update.

$$\varphi_\rho(\lambda; \theta) := \varphi(\lambda; \theta) + \frac{\rho}{2} \|\lambda\|^2, \quad \lambda_\rho^*(\theta) := \arg\min_{\lambda \in \Omega_\lambda} \varphi_\rho(\lambda; \theta). \tag{F.8}$$

We then present the following Lemmas that are useful for the proof of convergence of Algorithm 2.

**Lemma 19.** *Suppose Assumption 3 holds, and $\lambda^*(\theta)$ and $\lambda_\rho^*(\theta)$ are bounded for $\theta \in \{\theta_t\}_{t=0}^{T-1}$ produced by Algorithm 2, i.e., $\|\lambda^*(\theta)\| \le c_{\overline{\lambda}}$, $\|\lambda_\rho^*(\theta)\| \le c_{\overline{\lambda}}$. Then on the trajectory of Algorithm 2, with $\theta \in \{\theta_t\}_{t=0}^{T-1}$, we have*

$$\varphi(\lambda_\rho^*(\theta); \theta) - \varphi(\lambda^*(\theta); \theta) \le \frac{\rho}{2} c_{\overline{\lambda}}. \tag{F.9}$$

*Proof of Lemma 19.* The proof follows the proof of [7, Lemma 13]. $\square$

**Corollary 20.** *Suppose Assumption 3 holds, and $\lambda^*(\theta)$ and $\lambda_\rho^*(\theta)$ are bounded for $\theta \in \{\theta_t\}_{t=0}^{T-1}$ produced by Algorithm 2, i.e., $\|\lambda^*(\theta)\| \le c_{\overline{\lambda}}$, $\|\lambda_\rho^*(\theta)\| \le c_{\overline{\lambda}}$. Then on the trajectory of Algorithm 2, with $\theta \in \{\theta_t\}_{t=0}^{T-1}$, we have*

$$\|\nabla_{\lambda_h} \varphi(\lambda; \theta)\|^2 \le 2\ell_{\varphi_\lambda, 1} \left( \varphi(\lambda; \theta) - \varphi(\lambda_\rho^*(\theta); \theta) \right) + \ell_{\varphi_\lambda, 1} \rho c_{\overline{\lambda}}. \tag{F.10}$$

*Proof of Corollary 20.* By applying Lemma 13, and that $\nabla_{\lambda_h} \varphi(\lambda^*(\theta); \theta) = 0$, we have

$$\|\nabla_{\lambda_h} \varphi(\lambda; \theta)\|^2 = \|\nabla_{\lambda_h} \varphi(\lambda; \theta) - \nabla_{\lambda_h} \varphi(\lambda^*(\theta); \theta)\|^2 \le \|\nabla_\lambda \varphi(\lambda; \theta) - \nabla_\lambda \varphi(\lambda^*(\theta); \theta)\|^2$$
$$\overset{\text{Lemma 13}}{\le} 2\ell_{\varphi_\lambda, 1} \left( \varphi(\lambda; \theta) - \min_{\lambda \in \Omega_\lambda(\theta)} \varphi(\lambda; \theta) \right). \tag{F.11}$$

Applying Lemma 19, we can further derive

$$\varphi(\lambda; \theta) - \min_{\lambda \in \Omega_\lambda(\theta)} \varphi(\lambda; \theta) = \varphi(\lambda; \theta) - \varphi(\lambda^*(\theta); \theta) + \varphi(\lambda_\rho^*(\theta); \theta) - \varphi(\lambda_\rho^*(\theta); \theta)$$
$$\overset{\text{Lemma 19}}{\le} \varphi(\lambda; \theta) - \varphi(\lambda_\rho^*(\theta); \theta) + \frac{\rho}{2} c_{\overline{\lambda}}. \tag{F.12}$$

Combining (F.11) and (F.12) yields the result. $\square$

**Lemma 21** (Continuity of $\lambda_\rho^*(\theta)$). *For $\lambda_\rho^*(\theta)$ defined in (F.8), and $\Omega_\lambda(\theta) = \Omega_\lambda$, the following holds*

$$\|\lambda_\rho^*(\theta) - \lambda_\rho^*(\theta')\| \le \rho^{-1} \|\nabla_\lambda^2 \varphi(\lambda_\rho^*(\theta); \theta) - \nabla_\lambda^2 \varphi(\lambda_\rho^*(\theta'); \theta')\|$$
$$\le 2\rho^{-1} \ell_{f,1} \ell_f \|A_{ag}^\top\|_{\infty,1}^2 \|\theta - \theta'\|. \tag{F.13}$$

*Proof of Lemma 21.* The proof follows the proof of [7, Lemma 12]. $\square$

**Lemma 22.** *Suppose Assumptions 1, 2, 3 hold. Let $\{\theta_t\}, \{\lambda_t\}$ be the sequences produced by Algorithm 2 with step sizes $\alpha_t = \alpha > 0$, $\gamma_t = \gamma > 0$. Assume $\|\lambda^*(\theta_t)\|, \|\lambda_\rho^*(\theta_t)\|, \|\lambda_t\| \le c_{\overline{\lambda}}$. Then for any $\rho > 0$, it holds that*

$$\frac{1}{T} \sum_{t=0}^{T-1} \varphi(\lambda_t; \theta_t) - \varphi(\lambda_\rho^*(\theta_t); \theta_t) \le \frac{2c_{\overline{\lambda}}^2}{\gamma T} (1 + 2\rho^{-1} \alpha T \ell_{f,1} \ell_f^2 \|A_{ag}^\top\|_{\infty,1}^3) + \frac{\gamma}{2T} \sum_{t=0}^{T-1} \|\nabla_\lambda \varphi(\lambda_t; \theta_t)\|^2. \tag{F.14}$$

*Proof of Lemma 22.* The proof follows the proof techniques of [7, Lemma 15].

First, applying Corollary 18 and $\gamma_t = \gamma$ yields

$$2\gamma\big(\varphi(\lambda_t;\theta_t) - \varphi(\lambda_\rho^*(\theta_t);\theta_t)\big) \leq \|\lambda_t - \lambda_\rho^*(\theta_t)\|^2 - \|\lambda_{t+1} - \lambda_\rho^*(\theta_t)\|^2 + \gamma^2\|\nabla_\lambda\varphi(\lambda_t;\theta_t)\|^2. \tag{F.15}$$

Taking telescoping sum of the above inequality and rearranging, we have

$$\frac{1}{T}\sum_{t=0}^{T-1}\varphi(\lambda_t;\theta_t) - \varphi(\lambda_\rho^*(\theta_t);\theta_t) \leq \frac{1}{2\gamma T}\Big(\underbrace{\sum_{t=0}^{T-1}\|\lambda_t - \lambda_\rho^*(\theta_t)\|^2 - \|\lambda_{t+1} - \lambda_\rho^*(\theta_t)\|^2}_{J_1}\Big)$$

$$+ \frac{\gamma}{2T}\sum_{t=0}^{T-1}\|\nabla_\lambda\varphi(\lambda_t;\theta_t)\|^2 \tag{F.16}$$

where $J_1$ can be further bounded by

$$J_1 \leq \|\lambda_0 - \lambda_\rho^*(\theta_0)\|^2 - \|\lambda_T - \lambda_\rho^*(\theta_{T-1})\|^2 + \sum_{t=0}^{T-2}\|2\lambda_{t+1} - \lambda_\rho^*(\theta_{t+1}) - \lambda_\rho^*(\theta_t)\|\|\lambda_\rho^*(\theta_{t+1}) - \lambda_\rho^*(\theta_t)\|$$

$$\leq 4c_{\overline{\lambda}}^2 + 4c_{\overline{\lambda}}\sum_{t=0}^{T-2}\|\lambda_\rho^*(\theta_{t+1}) - \lambda_\rho^*(\theta_t)\| \leq 4c_{\overline{\lambda}}^2 + 8c_{\overline{\lambda}}\sum_{t=0}^{T-2}\rho^{-1}\alpha_t\ell_{f,1}\ell_f\|A_{ag}^\top\|_{\infty,1}^2\|d_t\|$$

where the last inequality follows from Lemma 21 and the update of $\theta_t$.

Finally, taking $\alpha_t = \alpha$, plugging the above bound for $J_1$ back into (F.16), and bounding $\|d_t\|$ by Assumption 3 and that $\|\lambda_t\| \leq c_{\overline{\lambda}}$ prove the result. $\qquad\square$

*Proof of Theorem 2.* We consider the following Lyapunov function with a constant vector $\lambda = [\lambda_f;\lambda_h] \in \Omega_\lambda$, where $\lambda_f \in \Delta^M$, $\lambda_h \in \mathbb{R}^{M_h}$.

$$\mathbb{V}_t := \underbrace{\lambda_f^\top AF(\theta_t)}_{\mathbb{V}_{f,t}} + \underbrace{\underbrace{\frac{\alpha_0}{2\gamma_0}\|\lambda_{f,t} - \lambda_f\|^2}_{\mathbb{V}_{\lambda_f,t}} + \underbrace{\frac{\alpha_0}{2\gamma_0}\|\lambda_{h,t} - \lambda_h\|^2}_{\mathbb{V}_{\lambda_h,t}}}_{\mathbb{V}_{\lambda,t}} + \underbrace{\underbrace{\lambda_h^\top H(\theta_t)}_{\mathbb{V}_{h,1,t}} + \underbrace{c_{V_h}\|H(\theta_t)\|_1}_{\mathbb{V}_{h,3,t}}}_{\mathbb{V}_{h,t}}. \tag{F.17}$$

Recall that $\lambda_t = [\lambda_{f,t};\lambda_{h,t}]$, and the algorithm takes the update $\theta_{t+1} = \theta_t + \alpha_t d_t$ with $d_t = \nabla F(\theta_t)A_{ag}^\top\lambda_t$. From Assumption 2, the smoothness of the objectives, and Lemma 9, the function $\lambda_f^\top AF(\theta)$ is smooth, thus

$$\mathbb{V}_{f,t+1} - \mathbb{V}_{f,t} \leq \langle\nabla F(\theta_t)A^\top\lambda_f, \theta_{t+1} - \theta_t\rangle + \frac{\ell_{f,1}}{2}\|A^\top\lambda_f\|_1\|\theta_{t+1} - \theta_t\|^2$$

$$= \alpha_t\langle\nabla F(\theta_t)A^\top\lambda_f, d_t\rangle + \frac{\ell_{f,1}}{2}\alpha_t^2\|A^\top\lambda_f\|_1\|d_t\|^2. \tag{F.18}$$

By Lemma 17, taking $\gamma_t > 0$ and rearranging, we have

$$\langle\nabla F(\theta_t)A^\top\lambda_f, d_t\rangle \leq \frac{1}{2\gamma_t}\big(\|\lambda_{f,t} - \lambda_f\|^2 - \|\lambda_{f,t+1} - \lambda_f\|^2\big)$$

$$+ \frac{1}{2}\gamma_t\|\nabla_{\lambda_f}\varphi(\lambda_t;\theta_t)\|^2 - \langle\lambda_{f,t}, \nabla_{\lambda_f}\varphi(\lambda_t;\theta_t)\rangle. \tag{F.19}$$

Combining (F.18) and (F.19), and choosing $\frac{\alpha_t}{\gamma_t} = \frac{\alpha_0}{\gamma_0}$ for all $t \in [T]$, we have

$$\mathbb{V}_{f,t+1} - \mathbb{V}_{f,t} + \mathbb{V}_{\lambda_f,t+1} - \mathbb{V}_{\lambda_f,t}$$

$$\leq \frac{\ell_{f,1}}{2}\alpha_t^2\|A^\top\lambda_f\|_1\|d_t\|^2 + \frac{1}{2}\alpha_t\gamma_t\|\nabla_{\lambda_f}\varphi(\lambda_t;\theta_t)\|^2 - \alpha_t\langle\lambda_{f,t}, \nabla_{\lambda_f}\varphi(\lambda_t;\theta_t)\rangle. \tag{F.20}$$

By the smoothness of $\lambda_h^\top H(\theta)$, and $\nabla_{\lambda_h}\varphi(\lambda_t;\theta_t) = -\nabla H(\theta_t)^\top d_t - c_h H(\theta_t)$, it holds that

$$\mathbb{V}_{h,1,t+1} - \mathbb{V}_{h,1,t} \leq \alpha_t\lambda_h^\top\nabla H(\theta_t)^\top d_t + \frac{\ell_{f,1}}{2}\alpha_t^2\|B_h^\top\lambda_h\|_1\|d_t\|^2$$

$$= -\alpha_t c_h \lambda_h^\top H(\theta_t) + \frac{\ell_{f,1}}{2}\alpha_t^2 \|B_h^\top \lambda_h\|_1 \|d_t\|^2$$
$$- \alpha_t \langle \lambda_h, \nabla_{\lambda_h}\varphi(\lambda_t;\theta_t)\rangle. \tag{F.21}$$

Bounding the last term in the above inequality by Lemma 17, and taking $\gamma_t > 0$, we have

$$\mathbb{V}_{h,1,t+1} - \mathbb{V}_{h,1,t} \leq -\alpha_t c_h \lambda_h^\top H(\theta_t) + \frac{\ell_{f,1}}{2}\alpha_t^2 \|B_h^\top \lambda_h\|_1 \|d_t\|^2 + \frac{1}{2}\alpha_t\gamma_t \|\nabla_{\lambda_h}\varphi(\lambda_t;\theta_t)\|^2$$
$$- \alpha_t \langle \lambda_{h,t}, \nabla_{\lambda_h}\varphi(\lambda_t;\theta_t)\rangle + \frac{\alpha_t}{2\gamma_t}\big(\|\lambda_{h,t} - \lambda_h\|^2 - \|\lambda_{h,t+1} - \lambda_h\|^2\big). \tag{F.22}$$

Adding up (F.20) and (F.22) yields

$$\mathbb{V}_{f,t+1} - \mathbb{V}_{f,t} + \mathbb{V}_{\lambda,t+1} - \mathbb{V}_{\lambda,t} + \mathbb{V}_{h,1,t+1} - \mathbb{V}_{h,1,t}$$
$$\leq -\alpha_t \langle \lambda_t, \nabla_\lambda \varphi(\lambda_t;\theta_t)\rangle + \frac{1}{2}\gamma_t\alpha_t\|\nabla_\lambda\varphi(\lambda_t;\theta_t)\|^2 - \alpha_t c_h \lambda_h^\top H(\theta_t) + \frac{\ell_{f,1}}{2}\alpha_t^2 \|A_{ag}^\top \lambda\|_1 \|d_t\|^2$$
$$\leq -\alpha_t \|d_t\|^2 + \alpha_t c_h (\lambda_{h,t} - \lambda_h)^\top H(\theta_t) + \frac{1}{2}\gamma_t\alpha_t\|\nabla_\lambda\varphi(\lambda_t;\theta_t)\|^2 + \frac{\ell_{f,1}}{2}\alpha_t^2 \|A_{ag}^\top \lambda\|_1 \|d_t\|^2 \tag{F.23}$$

where the last inequality uses the fact that $\langle \lambda_t, \nabla_\lambda\varphi(\lambda_t;\theta_t)\rangle = \|d_t\|^2 - c_h \lambda_{h,t}^\top H(\theta_t)$.

Using the fact that $\nabla_{\lambda_h}\varphi(\lambda_t;\theta_t) = -\nabla H(\theta_t)^\top d_t - c_h H(\theta_t)$, and with similar arguments as (E.20) in Lemma 11, we can further derive that

$$|H(\theta_{t+1})|_{\mathrm{ab}} \leq |H(\theta_t) - \alpha_t c_h H(\theta_t) - \alpha_t \nabla_{\lambda_h}\varphi(\lambda_t;\theta_t)|_{\mathrm{ab}} + \frac{\ell_{f,1}}{2}\alpha_t^2 \|B_h^\top\|_{\infty,1}\|d_t\|^2 \mathbf{1}$$
$$\leq (1 - \alpha_t c_h)|H(\theta_t)|_{\mathrm{ab}} + \frac{\ell_{f,1}}{2}\alpha_t^2 \|B_h^\top\|_{\infty,1}\|d_t\|^2 \mathbf{1} + \alpha_t |\nabla_{\lambda_h}\varphi(\lambda_t;\theta_t)|_{\mathrm{ab}}. \tag{F.24}$$

Therefore,

$$\mathbb{V}_{h,2,t+1} - \mathbb{V}_{h,3,t} \leq -\alpha_t c_h c_{V_h}\|H(\theta_t)\|_1 + \frac{\ell_{f,1}}{2}c_{V_h}M_h\alpha_t^2 \|B_h^\top\|_{\infty,1}\|d_t\|^2 + \alpha_t c_{V_h}\|\nabla_{\lambda_h}\varphi(\lambda_t;\theta_t)\|_1. \tag{F.25}$$

Combining (F.23) and (F.25), and by choosing step sizes $\alpha_t$, $\gamma_t$, parameter $c_{V_h}$ such that

$$\frac{\ell_{f,1}}{2}c_{V_h}M_h\alpha_t^2 \|B_h^\top\|_{\infty,1} + \frac{\ell_{f,1}}{2}\alpha_t^2 \|A_{ag}^\top\lambda\|_1 \leq \frac{1}{2}, \tag{F.26}$$

we have

$$\mathbb{V}_{t+1} - \mathbb{V}_t \leq -\frac{1}{2}\alpha_t \|d_t\|^2 - \alpha_t c_h (c_{V_h} - \|\lambda_h - \lambda_{h,t}\|_1)\|H(\theta_t)\|_1 + \frac{1}{2}\gamma_t\alpha_t\ell_\varphi^2 + \alpha_t c_{V_h}\|\nabla_{\lambda_h}\varphi(\lambda_t;\theta_t)\|_1. \tag{F.27}$$

Taking telescoping sum of the above inequality over $t = 0,\ldots,T-1$, and applying that $\|\nabla_{\lambda_h}\varphi(\lambda_t;\theta_t)\|_1 \leq \sqrt{M_h}\|\nabla_{\lambda_h}\varphi(\lambda_t;\theta_t)\|$, we have

$$\sum_{t=0}^{T-1}\mathbb{V}_{t+1} - \mathbb{V}_t \leq \sum_{t=0}^{T-1} -\frac{1}{2}\alpha_t \|d_t\|^2 - \alpha_t c_h (c_{V_h} - \|\lambda_h - \lambda_{h,t}\|_1)\|H(\theta_t)\|_1 + \frac{1}{2}\gamma_t\alpha_t\ell_\varphi^2$$
$$+ \alpha_t c_{V_h}\sqrt{M_h}\|\nabla_{\lambda_h}\varphi(\lambda_t;\theta_t)\| \tag{F.28}$$

where $\sum_{t=0}^{T-1}\|\nabla_{\lambda_h}\varphi(\lambda_t;\theta_t)\|$ can be further bounded by applying Lemma 22 and Corollary 20 along with Jensen's inequality as follows

$$\Big(\frac{1}{T}\sum_{t=0}^{T-1}\|\nabla_{\lambda_h}\varphi(\lambda_t;\theta_t)\|\Big)^2 \leq \frac{1}{T}\sum_{t=0}^{T-1}\|\nabla_{\lambda_h}\varphi(\lambda_t;\theta_t)\|^2$$
$$\overset{\text{Corollary 20}}{\leq} \frac{1}{T}\sum_{t=0}^{T-1}2\ell_{\varphi_\lambda,1}\big(\varphi(\lambda_t;\theta_t) - \varphi(\lambda_\rho^*(\theta_t);\theta_t)\big) + \ell_{\varphi_\lambda,1}\rho c_{\overline{\lambda}}$$

$$\overset{\text{Lemma 22}}{\leq} 4\ell_{\varphi_\lambda,1}c_\lambda^2 \frac{1}{\gamma T}\left(1+2\rho^{-1}\alpha T\ell_{f,1}\ell_f^2\|A_{ag}^\top\|_{\infty,1}^3\right) + \frac{\gamma}{2T}\sum_{t=0}^{T-1}\|\nabla_\lambda\varphi(\lambda_t;\theta_t)\|^2 + \rho\ell_{\varphi_\lambda,1}c_{\overline{\lambda}} \tag{F.29}$$

where $\|\nabla_\lambda\varphi(\lambda_t;\theta_t)\|^2 = \|\nabla_{\lambda_h}\varphi(\lambda_t;\theta_t)\|^2 + \|\nabla_{\lambda_f}\varphi(\lambda_t;\theta_t)\|^2 \lesssim \|\nabla_{\lambda_h}\varphi(\lambda_t;\theta_t)\|^2 + \|d_t\|^2$. Plugging the above inequality back into (F.28), choosing $\rho = \Theta\left(\left(\frac{\alpha}{\gamma}\right)^{\frac{1}{2}}\right)$, and rearranging yield

$$\frac{1}{T}\sum_{t=0}^{T-1}\|d_t\|^2 + \|H(\theta_t)\|_1 = \mathcal{O}\left(\frac{1}{\alpha T} + \frac{1}{(\gamma T)^{\frac{1}{2}}} + \left(\frac{\alpha}{\gamma}\right)^{\frac{1}{4}} + \gamma\right). \tag{F.30}$$

Choosing $\alpha = \Theta(T^{-\frac{5}{6}})$, $\gamma = \Theta(T^{-\frac{1}{6}})$ proves the result. $\qquad\square$

## F.3 Sharper analysis with a different merit function: proof of Theorem 3

In this section, we provide an analysis of convergence of Algorithm 2 with a different merit function and faster convergence rate. We first present the auxiliary lemmas and then prove Theorem 3.

**Lemma 23.** *Suppose Assumptions 1 and 3 hold. For $\{\theta_t\}, \{\lambda_t\}$ produced by Algorithm 2 with $M_g = 0$ and $\Omega_{\lambda_f}(\theta_t) = \Delta^M$, and for all $t = 0,\ldots,T$, $\|d_t\|$ can be bounded by*

$$\|d_t\| \leq \ell_{f,1}\|A_{ag}^\top\|_1(1+\|\lambda_{h,t}\|_1), \tag{F.31}$$

*and $\|\nabla_\lambda\varphi(\lambda_t;\theta_t)\|$ can be bounded by*

$$\|\nabla_\lambda\varphi(\lambda_t;\theta_t)\|^2 \leq 2\|A_{ag}\|^2 M\ell_f^2\|d_t\|^2 + 2c_h^2\|H(\theta_t)\|^2. \tag{F.32}$$

*Proof.* Since $d_t = \nabla F(\theta_t)A_{ag}^\top\lambda_t$, we have

$$\|d_t\| = \|\nabla F(\theta_t)A_{ag}^\top\lambda_t\| \leq \ell_{f,1}\|A_{ag}^\top\|_1\|\lambda_t\|_1 \leq \ell_{f,1}\|A_{ag}^\top\|_1(1+\|\lambda_{h,t}\|_1) \tag{F.33}$$

which proves (F.31).

Furthermore, invoking that $\nabla_\lambda\varphi(\lambda_t;\theta_t) = A_{ag}\nabla F(\theta_t)^\top d_t - c_h H(\theta_t)$, we have

$$\|\nabla_\lambda\varphi(\lambda_t;\theta_t)\|^2 = \|A_{ag}\nabla F(\theta_t)^\top d_t - c_h H(\theta_t)\|^2$$
$$\leq 2\|A_{ag}\nabla F(\theta_t)^\top d_t\|^2 + 2\|c_h H(\theta_t)\|^2 \leq 2\|A_{ag}\|^2 M\ell_f^2\|d_t\|^2 + 2c_h^2\|H(\theta_t)\|^2 \tag{F.34}$$

which proves (F.32). $\qquad\square$

**Lemma 24.** *Suppose Assumptions 1, 2, and 3 hold, and $M_g = 0$. For $\{\theta_t\}, \{\lambda_t\}$ produced by Algorithm 2, further assume $\{\lambda_{h,t}\}$ are bounded on the trajectory, i.e., $\|\lambda_{h,t}\|_1 \leq c_{\lambda_h}$. For all $t = 0,\ldots,T$, choose $\alpha_t$ such that $\alpha_t \leq \frac{c_{\alpha,h}}{\ell_{f,1}\|A_{ag}^\top\|_1(1+c_{\lambda_h})}$, and $\alpha_t\|H(\theta_t)\| \leq c_{\alpha,h}$ for any $0 < c_{\alpha,h} < \infty$, then it holds that*

$$\|H(\theta_{t+1})\|^2 - \|H(\theta_t)\|^2 \leq \alpha_t 2H(\theta_t)^\top\nabla H(\theta_t)^\top d_t + \frac{1}{2}\alpha_t^2\ell_{H^2,1,t}\|d_t\|^2$$

*with $\ell_{H^2,1,t} = 2M\|B_h\|^2\ell_f^2 + 2(\alpha_t^{-1}+\ell_H)c_{\alpha,h}\sqrt{M}\ell_{f,1}\|B_h\|$, and $\ell_H = \|B_h\|\sqrt{M}\ell_f$.* (F.35)

*Proof.* By choosing $\alpha_t \leq \frac{c_{\alpha,h}}{\ell_{f,1}\|A_{ag}^\top\|_1(1+\|\lambda_{h,t}\|_1)}$, and invoking (F.31) in Lemma 23, we have

$$\alpha_t\|d_t\| \leq c_{\alpha,h}. \tag{F.36}$$

By the mean-value theorem, for all $t = 0,\ldots,T$, there exists $\tilde{\theta}_t$ such that

$$\|H(\theta_{t+1})\|^2 - \|H(\theta_t)\|^2 \leq \alpha_t 2H(\theta_t)^\top\nabla H(\theta_t)^\top d_t + \frac{1}{2}\alpha_t^2\|\nabla^2(H(\tilde{\theta}_t)^\top H(\tilde{\theta}_t))\|\|d_t\|^2. \tag{F.37}$$

The term $\|\nabla^2(H(\tilde{\theta}_t)^\top H(\tilde{\theta}_t))\|$ can be upper bounded by

$$\|\nabla^2(H(\tilde{\theta}_t)^\top H(\tilde{\theta}_t))\| \leq 2\|\nabla H(\tilde{\theta}_t)\nabla H(\tilde{\theta}_t)^\top\| + 2\|\nabla^2 H(\tilde{\theta}_t)\|\|H(\tilde{\theta}_t)\|$$

$$\leq 2M\|B_h\|^2 \ell_f^2 + 2\|H(\tilde{\theta}_t)\|\sqrt{M}\ell_{f,1}\|B_h\|. \tag{F.38}$$

Since $H(\tilde{\theta}_t)$ is $\ell_H$-Lipschitz continuous with $\ell_H = \|B_h\|\sqrt{M}\ell_f$, and $\tilde{\theta}_t$ lies on the line segment of $\theta_t$ and $\theta_{t+1}$ with $\|\theta_{t+1} - \theta_t\| = \alpha_t\|d_t\|$, therefore,

$$\|H(\tilde{\theta}_t)\| \leq \|H(\theta_t)\| + \alpha_t \ell_H \|d_t\| \leq \|H(\theta_t)\| + \ell_H c_{\alpha,h}. \tag{F.39}$$

Plugging the above inequality into (F.37) yields

$$\|H(\theta_{t+1})\|^2 - \|H(\theta_t)\|^2 \leq \alpha_t 2H(\theta_t)^\top \nabla H(\theta_t)^\top d_t + \frac{1}{2}\alpha_t^2 \|\nabla^2 H(\tilde{\theta}_t)^\top H(\tilde{\theta}_t)\|\|d_t\|^2$$

$$\leq \alpha_t 2H(\theta_t)^\top \nabla H(\theta_t)^\top d_t + \frac{1}{2}\alpha_t^2 \Big(2M\|B_h\|^2\ell_f^2 + 2(\alpha_t^{-1} + \ell_H)c_{\alpha,h}\sqrt{M}\ell_{f,1}\|B_h\|\Big)\|d_t\|^2. \tag{F.40}$$

The proof is complete. $\qquad\square$

**Lemma 25** (Smoothness of $\varphi$ w.r.t. $\theta$). *Under Assumptions 2, 3, 4-2, $\varphi(\lambda;\theta)$ is $\ell_{\varphi_\theta,1}$-smooth w.r.t. $\theta$ for all $\lambda, \theta$ on the trajectory of Algorithm 2. with $\ell_{\varphi_\theta,1} = M(\ell_{f,1}^2 + \ell_{f,2}\ell_f)(\|A\| + \|B_h\|c_{\lambda_h})^2 + c_h c_{\lambda_h}\|B_h\|\ell_{f,1}$.*

*Proof of Lemma 25.* By the definition of $\varphi$, its gradient w.r.t $\theta$ can be computed by

$$\nabla_\theta\varphi(\lambda;\theta) = \big(\nabla F(\theta)A_{ag}^\top\lambda\big)^\top \nabla^2 F(\theta)\big(A_{ag}^\top\lambda\big) - c_h\lambda_h^\top \nabla H(\theta). \tag{F.41}$$

For brevity, let $v = A_{ag}^\top\lambda$. Then for any $\theta, \theta' \in \{\theta_t\}$ on the trajectory of Algorithm 2, $\|\nabla_\theta\varphi(\lambda;\theta) - \nabla_\theta\varphi(\lambda;\theta')\|$ can be further bounded by

$$\|\nabla_\theta\varphi(\lambda;\theta) - \nabla_\theta\varphi(\lambda;\theta')\| \leq \|\nabla F(\theta) - \nabla F(\theta')\|\|v\|\|\nabla^2 F(\theta)v\|$$

$$+ \|\nabla F(\theta')v\|\|\nabla^2 F(\theta) - \nabla^2 F(\theta')\|\|v\| + c_h\|\lambda_h\|\|\nabla H(\theta) - \nabla H(\theta')\|$$

$$\leq \big(M\ell_{f,1}^2\|v\|^2 + M\ell_f\ell_{f,2}\|v\|^2 + c_h\|\lambda_h\|\|B_h\|\ell_{f,1}\big)\|\theta - \theta'\| \tag{F.42}$$

where the last inequality follows from Assumptions 2 and 3. Using the fact that $\lambda_f \in \Delta^M$, and $\|\lambda_{h,t}\| \leq c_{\lambda_h}$ on the trajectory of Algorithm 2, for all $\lambda \in \{\lambda_t\}$ on the trajectory of Algorithm 2, $\|v\| = \|A_{ag}^\top\lambda\|$ can be further bounded by

$$\|A_{ag}^\top\lambda\| \leq \|A^\top\lambda_f\| + \|B_h^\top\lambda_h\| \leq \|A\| + \|B_h\|c_{\lambda_h}. \tag{F.43}$$

Plugging the above inequality back into (F.42) completes the proof. $\qquad\square$

**Lemma 26** ([53, Lemma 4]). *Let $\Omega \subseteq \mathbb{R}^M$ be a closed convex set, and let $\Pi_\Omega$ denote Euclidean projection to $\Omega$. Given any $\lambda \in \Omega, d \in \mathbb{R}^M$ and $\gamma > 0$, it holds that*

$$\Pi_\Omega(\lambda - \gamma d) = \underset{\lambda' \in \Omega}{\arg\min}\, \langle d, \lambda'\rangle + \frac{1}{2\gamma}\|\lambda - \lambda'\|^2. \tag{F.44}$$

**Lemma 27** (Proximal PL inequality implies proximal error bound and quadratic growth). *Suppose Assumptions 1, 3, and 4-1 hold. Then for $\lambda, \theta$ on the trajectory of Algorithm 2, $\varphi(\lambda;\theta) + g(\lambda)$, with $g(\lambda)$ being an indicator function defined on the set $\Omega_\lambda$, satisfies the $\frac{1}{\bar{\mu}_\varphi}$-proximal error bound (EB) and the $\frac{1}{\mu'_\varphi}$-quadratic growth (QG) w.r.t. $\lambda$ for some $\bar{\mu}_\varphi, \mu'_\varphi > 0$ depending on $\mu_\varphi$, as defined below*

$$\frac{1}{\bar{\mu}_\varphi}\mathrm{dist}(\lambda, S_\varphi(\theta)) \leq \frac{1}{\gamma}\|\lambda - \Pi_{\Omega_\lambda}(\lambda - \gamma\nabla_\lambda\varphi(\lambda;\theta))\| \quad \text{(proximal EB)} \tag{F.45}$$

$$\frac{1}{\mu'_\varphi}\mathrm{dist}^2(\lambda, S_\varphi(\theta)) \leq \varphi(\lambda;\theta) - \varphi(\lambda^*(\theta);\theta) \quad \text{(QG)} \tag{F.46}$$

*where $S_\varphi(\theta) := \{\lambda \in \Omega_\lambda \mid \varphi(\lambda;\theta) = \varphi(\lambda^*(\theta);\theta)\}$.*

*Proof of Lemma 27.* By Lemma 15, $\varphi(\lambda;\theta)$ is smooth w.r.t. $\lambda$. Furthermore, by [28, Appendix G], and combined with Assumption 4-1, the proximal PL inequality, it implies that $\varphi(\lambda;\theta) + g(\lambda)$ satisfies the proximal error bound. From [10, Corollary 3.6], the proximal error bound further implies the quadratic growth, which proves the result. $\qquad\square$

**Lemma 28** (Lipschitz continuity of $\lambda^*(\theta)$, [53, Lemma 5]). *Suppose Assumption 3 holds. If given $\lambda \in \Omega_\lambda$, and $\theta' \in \mathbb{R}^q$, $\varphi(\lambda; \theta')$ satisfies the $\frac{1}{\bar{\mu}_\varphi}$-proximal error bound w.r.t. $\lambda$. Then given $\theta \in \mathbb{R}^q$, for any $\lambda^*(\theta) \in \arg\min_{\lambda \in \Omega_\lambda} \varphi(\lambda; \theta)$, there exists $\lambda^*(\theta') \in \arg\min_{\lambda \in \Omega_\lambda} \varphi(\lambda; \theta')$ such that*

$$\|\lambda^*(\theta) - \lambda^*(\theta')\| \le \ell_{\lambda^*}\|\theta - \theta'\|$$

*with $\ell_{\lambda^*} = \ell_{\varphi_\lambda, 1}\bar{\mu}_\varphi$, and $\ell_{\varphi_\lambda, 1}$ defined in Lemma 15.*

**Lemma 29** (Danskin-type Lemma for proximal PL functions [53, Proposition 6]). *Suppose Assumptions 1, 2, 3, 4 hold, then $\varphi(\lambda^*(\theta); \theta)$ is differentiable with the gradient computed by*

$$\nabla\varphi(\lambda^*(\theta); \theta) = \nabla_\theta\varphi(\lambda; \theta), \quad \forall \lambda \in \arg\min_{\lambda \in \Omega_\lambda} \varphi(\lambda; \theta). \tag{F.47}$$

*Moreover, $\varphi(\lambda^*(\theta); \theta)$ is $\ell_{\varphi^*, 1}$-smooth with $\ell_{\varphi^*, 1} := \ell_{\varphi, 1}(1 + \ell_{\lambda^*})$.*

Below, Lemma 30 establishes the approximate descent or contraction of the subprogram after taking one-step update on $\lambda_t$. This is crucial for a sharper analysis of convergence of Algorithm 2.

**Lemma 30** (Error of subprogram). *Suppose Assumptions 1, 2, 3, 4 hold, $M_g = 0$, and $\Omega_{\lambda_f}(\theta) = \Delta^M$. Let $\{\theta_t\}, \{\lambda_t\}$ be the sequences produced by Algorithm 2 with step size $\gamma_t \le \ell_{\varphi_\lambda, 1}^{-1}$. Then for any $c_{\varphi, d} > 0$, the following hold*

$$\varphi(\lambda_{t+1}; \theta_t) - \varphi(\lambda^*(\theta_t); \theta_t) \le (1 - \gamma_t\mu_\varphi)\big(\varphi(\lambda_t; \theta_t) - \varphi(\lambda^*(\theta_t); \theta_t)\big) \tag{F.48a}$$

$$\varphi(\lambda_{t+1}; \theta_{t+1}) - \varphi(\lambda^*(\theta_{t+1}); \theta_{t+1}) \le \Big(1 + \alpha_t c_{\varphi, d}\ell_{\varphi_\theta, 1}^2\mu_\varphi'\Big)\Big(\varphi(\lambda_{t+1}; \theta_t) - \varphi(\lambda^*(\theta_t); \theta_t)\Big)$$

$$+ \Big(\frac{\alpha_t}{2c_{\varphi, d}} + \frac{\ell_{\varphi_\theta, 1} + \ell_{\varphi^*, 1}}{2}\alpha_t^2\Big)\|d_t\|^2. \tag{F.48b}$$

*Proof of Lemma 30.* We first prove (F.48a). Recall the definition of $D_{\varphi, \gamma}(\lambda; \theta)$ in Definition 3. By the $\ell_{\varphi_\lambda, 1}$-smoothness of $\varphi$ w.r.t. $\lambda$ and the update on $\lambda_t$, and that $\gamma_t \le \ell_{\varphi_\lambda, 1}^{-1}$, we have

$$\varphi(\lambda_{t+1}; \theta_t) \le \varphi(\lambda_t; \theta_t) + \langle \nabla_\lambda\varphi(\lambda_t; \theta_t), \lambda_{t+1} - \lambda_t \rangle + \frac{1}{2\gamma_t}\|\lambda_{t+1} - \lambda_t\|^2$$

$$\le \varphi(\lambda_t; \theta_t) - \frac{\gamma_t}{2}D_{\varphi, \gamma_t}(\lambda_t; \theta_t) \quad \text{by Lemma 26 and Definition 3}$$

$$\le \varphi(\lambda_t; \theta_t) - \gamma_t\mu_\varphi\Big(\varphi(\lambda_t; \theta_t) - \varphi(\lambda^*(\theta_t); \theta_t)\Big) \quad \text{by Assumption 4-1 and Definition 3} \tag{F.49}$$

where the last inequality follows from the proximal PL inequality. Subtracting both sides of the above inequality by $\varphi(\lambda^*(\theta_t); \theta_t)$ proves (F.48a).

Next we prove (F.48b). We decompose the error on the left hand side of (F.48b) by

$$\varphi(\lambda_{t+1}; \theta_{t+1}) - \varphi(\lambda^*(\theta_{t+1}); \theta_{t+1})$$

$$= \underbrace{\varphi(\lambda_{t+1}; \theta_{t+1}) - \varphi(\lambda^*(\theta_{t+1}); \theta_{t+1}) - (\varphi(\lambda_{t+1}; \theta_t) - \varphi(\lambda^*(\theta_t); \theta_t))}_{J_1} + \varphi(\lambda_{t+1}; \theta_t) - \varphi(\lambda^*(\theta_t); \theta_t)$$

$$\tag{F.50}$$

where we use the $(\ell_{\varphi_\theta, 1} + \ell_{\varphi^*, 1})$-smoothness of $\varphi(\lambda; \theta) - \varphi(\lambda^*(\theta); \theta)$ w.r.t. $\theta$ to further bound $J_1$ by

$$J_1 \le \langle \nabla_\theta\varphi(\lambda_{t+1}; \theta_t) - \nabla\varphi(\lambda^*(\theta_t); \theta_t), \theta_{t+1} - \theta_t \rangle + \frac{\ell_{\varphi_\theta, 1} + \ell_{\varphi^*, 1}}{2}\|\theta_{t+1} - \theta_t\|^2$$

$$\le -\alpha_t\langle \nabla_\theta\varphi(\lambda_{t+1}; \theta_t) - \nabla\varphi(\lambda^*(\theta_t); \theta_t), d_t \rangle + \frac{\ell_{\varphi_\theta, 1} + \ell_{\varphi^*, 1}}{2}\alpha_t^2\|d_t\|^2$$

$$\le \alpha_t\|\nabla_\theta\varphi(\lambda_{t+1}; \theta_t) - \nabla\varphi(\lambda^*(\theta_t); \theta_t)\|\|d_t\| + \frac{\ell_{\varphi_\theta, 1} + \ell_{\varphi^*, 1}}{2}\alpha_t^2\|d_t\|^2. \tag{F.51}$$

Then we can further derive that

$$J_1 \overset{(a)}{\le} \frac{\alpha_t c_{\varphi, d}}{2}\|\nabla_\theta\varphi(\lambda_{t+1}; \theta_t) - \nabla\varphi(\lambda^*(\theta_t); \theta_t)\|^2 + \frac{\alpha_t}{2c_{\varphi, d}}\|d_t\|^2 + \frac{\ell_{\varphi_\theta, 1} + \ell_{\varphi^*, 1}}{2}\alpha_t^2\|d_t\|^2$$

$$\stackrel{(b)}{\leq} \alpha_t c_{\varphi,d} \ell_{\varphi_\theta,1}^2 \mu_\varphi' \big( \varphi(\lambda_{t+1}; \theta_t) - \varphi(\lambda^*(\theta_t); \theta_t) \big) + \frac{\alpha_t}{2c_{\varphi,d}} \|d_t\|^2 + \frac{\ell_{\varphi_\theta,1} + \ell_{\varphi^*,1}}{2} \alpha_t^2 \|d_t\|^2$$

(F.52)

where $(a)$ is from Cauchy-Swartz inequality, and $(b)$ holds because

$$\|\nabla_\theta \varphi(\lambda_{t+1}; \theta_t) - \nabla \varphi(\lambda^*(\theta_t); \theta_t)\|^2$$
$$\leq \ell_{\varphi_\theta,1}^2 \big( \text{dist}(\lambda_{t+1}, \lambda^*(\theta_t)) \big)^2 \qquad \text{by Lemma 29 and } \ell_{\varphi_\theta,1}\text{-Lipschitz continuity of } \nabla_\theta \varphi(\lambda; \theta)$$
$$\leq \ell_{\varphi_\theta,1}^2 \mu_\varphi' \big( \varphi(\lambda_{t+1}; \theta_t) - \varphi(\lambda^*(\theta_t); \theta_t) \big)$$

(F.53)

where the last inequality follows from Lemma 27, the $\frac{1}{\mu_\varphi'}$-quadratic growth of $\varphi(\cdot; \theta)$.

Finally, plugging (F.52) back into (F.50) completes the proof of (F.48b). $\qquad \square$

**Corollary 31.** *Suppose Assumptions 1, 2, 3, 4 hold, and $M_g = 0$. Let $\{\theta_t\}, \{\lambda_t\}$ be the sequences produced by Algorithm 2 with step size $\gamma_t \leq \ell_{\varphi_\lambda,1}^{-1}$. Then for any $c_{\varphi,d} > 0$, it holds that*

$$\big( \varphi(\lambda_{t+1}; \theta_{t+1}) - \varphi(\lambda^*(\theta_{t+1}); \theta_{t+1}) \big) - \big( \varphi(\lambda_t; \theta_t) - \varphi(\lambda^*(\theta_t); \theta_t) \big)$$
$$\leq \left( \left( 1 + \alpha_t c_{\varphi,d} \ell_{\varphi_\theta,1}^2 \mu_\varphi' \right) (1 - \gamma_t \mu_\varphi) - 1 \right) \big( \varphi(\lambda_t; \theta_t) - \varphi(\lambda^*(\theta_t); \theta_t) \big) + \left( \frac{\alpha_t}{2c_{\varphi,d}} + \frac{\ell_{\varphi_\theta,1} + \ell_{\varphi^*,1}}{2} \alpha_t^2 \right) \|d_t\|^2.$$

(F.54)

*Proof of Corollary 31.* The proof directly follows by plugging (F.48a) into (F.48b). $\qquad \square$

**Lemma 32.** *If by choosing $\alpha_t = \min \big\{ \frac{c_{\alpha,h}}{\|H(\theta_t)\|}, c_\alpha \big\}$ with $0 < c_{\alpha,h}, c_\alpha < \infty$, and from the algorithm update and properties we can derive $\alpha_t \|H(\theta_t)\|^2 = \mathcal{O}(1)$ is bounded, then $\|H(\theta_t)\|$ is bounded.*

*Proof of Lemma 32.* We prove by contradiction. Suppose $\|H(\theta_t)\| = \omega(1)$ is not bounded, then

$$\alpha_t = \min \left\{ \frac{c_{\alpha,h}}{\|H(\theta_t)\|}, c_\alpha \right\} = \frac{c_{\alpha,h}}{\|H(\theta_t)\|}.$$

(F.55)

Furthermore,

$$\alpha_t \|H(\theta_t)\|^2 = c_{\alpha,h} \|H(\theta_t)\| = \mathcal{O}(1)$$

(F.56)

which implies $\|H(\theta_t)\| = \mathcal{O}(1)$ and contradicts with $\|H(\theta_t)\| = \omega(1)$. Therefore, we have proved $\|H(\theta_t)\|$ is bounded. $\qquad \square$

**Remark 33.** *Lemma 32 uses the algorithm properties to prove that $\|H(\theta_t)\|$ is bounded, instead of directly assuming $\|H(\theta_t)\|$ is bounded. This will be used in the proof of Theorem 3 to show that $\|H(\theta_t)\|$ is bounded on the trajectory of Algorithm 2.*

Next we proceed to prove Theorem 3, the sharper convergence of Algorithm 2.

*Proof of Theorem 3.* We consider the following Lyapunov function with a constant vector $\lambda_f \in \Delta^M$.

$$\mathbb{V}_t := \underbrace{\lambda_f^\top A F(\theta_t)}_{\mathbb{V}_{f,t}} + \underbrace{\frac{\alpha_0}{2\gamma_0} \|\lambda_{f,t} - \lambda_f\|^2}_{\mathbb{V}_{\lambda_f,t}} + \underbrace{\underbrace{\lambda_{h,t}^\top H(\theta_t)}_{\mathbb{V}_{h,0,t}} + \underbrace{\frac{1}{2} \|H(\theta_t)\|^2}_{\mathbb{V}_{h,3,t}}}_{\mathbb{V}_{h,t}} + \underbrace{\varphi(\lambda_t; \theta_t) - \varphi(\lambda^*(\theta_t); \theta_t)}_{\mathbb{V}_{\varphi,t}}.$$

(F.57)

Following the same arguments from (F.18)-(F.20), and by choosing $\frac{\alpha_t}{\gamma_t} = \frac{\alpha_0}{\gamma_0} = \frac{1}{c_{\gamma,\alpha}}$ for all $t \in [T]$, we have

$$\mathbb{V}_{f,t+1} - \mathbb{V}_{f,t} + \mathbb{V}_{\lambda_f,t+1} - \mathbb{V}_{\lambda_f,t}$$
$$\leq \frac{\ell_{f,1}}{2} \alpha_t^2 \|A^\top \lambda_f\|_1 \|d_t\|^2 + \frac{1}{2} \alpha_t \gamma_t \|\nabla_{\lambda_f} \varphi(\lambda_t; \theta_t)\|^2 - \alpha_t \langle \lambda_{f,t}, \nabla_{\lambda_f} \varphi(\lambda_t; \theta_t) \rangle.$$

(F.58)

Similarly, we can derive that

$$\mathbb{V}_{h,0,t+1} - \mathbb{V}_{h,0,t} \leq -\alpha_t c_h \lambda_{h,t}^\top H(\theta_t) + \frac{\ell_{f,1}}{2}\alpha_t^2 \|B_h^\top \lambda_{h,t}\|_1 \|d_t\|^2 - \alpha_t \langle \lambda_{h,t}, \nabla_{\lambda_h}\varphi(\lambda_t;\theta_t)\rangle$$
$$+ (\lambda_{h,t+1} - \lambda_{h,t})^\top H(\theta_t). \tag{F.59}$$

Combining (F.58) and (F.59) yields

$$\mathbb{V}_{f,t+1} - \mathbb{V}_{f,t} + \mathbb{V}_{\lambda_f,t+1} - \mathbb{V}_{\lambda_f,t} + \mathbb{V}_{h,0,t+1} - \mathbb{V}_{h,0,t} \leq -\alpha_t \|d_t\|^2$$
$$+ \frac{1}{2}\gamma_t\alpha_t \|\nabla_{\lambda_f}\varphi(\lambda_t;\theta_t)\|^2 + \frac{\ell_{f,1}}{2}\alpha_t^2 (\|A^\top \lambda_f\|_1 + \|B_h^\top \lambda_{h,t}\|_1)\|d_t\|^2 - \gamma_t \nabla_{\lambda_h}\varphi(\lambda_t;\theta_t)^\top H(\theta_t). \tag{F.60}$$

Next we proceed to bound $\mathbb{V}_{h,3,t+1} - \mathbb{V}_{h,3,t}$. By Lemma 24, it holds that

$$\mathbb{V}_{h,3,t+1} - \mathbb{V}_{h,3,t} \leq \alpha_t H(\theta_t)^\top \nabla H(\theta_t)^\top d_t + \frac{1}{4}\alpha_t^2 \ell_{H^2,1,t}\|d_t\|^2 \tag{F.61}$$

where $\ell_{H^2,1,t} = 2M\ell_f^2 + 2(\alpha_t^{-1}c_{\alpha,h} + \ell_H c_d)\sqrt{M}\ell_{f,1}$. Because $\nabla_{\lambda_h}\varphi(\lambda_t;\theta_t) = -\nabla H(\theta_t)^\top d_t - c_h H(\theta_t)$, the term $H(\theta_t)^\top \nabla H(\theta_t)^\top d_t$ can be further written as

$$H(\theta_t)^\top \nabla H(\theta_t)^\top d_t = -H(\theta_t)^\top \big(\nabla_{\lambda_h}\varphi(\lambda_t;\theta_t) + c_h H(\theta_t)\big)$$
$$= -c_h \|H(\theta_t)\|^2 - H(\theta_t)^\top \nabla_{\lambda_h}\varphi(\lambda_t;\theta_t). \tag{F.62}$$

Plugging (F.62) into (F.61) yields

$$\frac{1}{2}\|H(\theta_{t+1})\|^2 - \frac{1}{2}\|H(\theta_t)\|^2 \leq -\alpha_t c_h \|H(\theta_t)\|^2 - \alpha_t H(\theta_t)^\top \nabla_{\lambda_h}\varphi(\lambda_t;\theta_t) + \frac{1}{4}\alpha_t^2 \ell_{H^2,1,t}\|d_t\|^2. \tag{F.63}$$

Letting $\ell_{FH,1} = \ell_{f,1}(\|A^\top\|_1 + \|B_h^\top\|_1 c_{\lambda_h})$, and adding up (F.60) and (F.63), we have

$$\mathbb{V}_{f,t+1} - \mathbb{V}_{f,t} + \mathbb{V}_{\lambda_f,t+1} - \mathbb{V}_{\lambda_f,t} + \mathbb{V}_{h,t+1} - \mathbb{V}_{h,t} \leq -\alpha_t\|d_t\|^2 - \alpha_t c_h\|H(\theta_t)\|^2$$
$$- (\alpha_t + \gamma_t)\nabla_{\lambda_h}\varphi(\lambda_t;\theta_t)^\top H(\theta_t) + \frac{1}{2}\gamma_t\alpha_t\|\nabla_{\lambda_f}\varphi(\lambda_t;\theta_t)\|^2 + \frac{1}{4}\alpha_t^2(2\ell_{FH,1} + \ell_{H^2,1,t})\|d_t\|^2 \tag{F.64}$$

where $\|\nabla_{\lambda_f}\varphi(\lambda_t;\theta_t)\|^2 \leq \|\nabla_\lambda\varphi(\lambda_t;\theta_t)\|^2$ is further bounded by Lemma 23, (F.32) as

$$\|\nabla_\lambda\varphi(\lambda_t;\theta_t)\|^2 \leq 2\|A_{ag}\|^2 M\ell_f^2\|d_t\|^2 + 2c_h^2\|H(\theta_t)\|^2. \tag{F.65}$$

Plugging (F.65) back into (F.64) yields

$$\mathbb{V}_{f,t+1} - \mathbb{V}_{f,t} + \mathbb{V}_{\lambda_f,t+1} - \mathbb{V}_{\lambda_f,t} + \mathbb{V}_{h,t+1} - \mathbb{V}_{h,t} \leq -\alpha_t\|d_t\|^2 - \alpha_t c_h\|H(\theta_t)\|^2$$
$$- (\alpha_t + \gamma_t)\nabla_{\lambda_h}\varphi(\lambda_t;\theta_t)^\top H(\theta_t) + \underbrace{\frac{1}{4}\alpha_t^2(2\ell_{FH,1} + \ell_{H^2,1,t})\|d_t\|^2}_{J_1}$$
$$+ \underbrace{\gamma_t\alpha_t\|A_{ag}\|^2 M\ell_f^2\|d_t\|^2}_{J_2} + \underbrace{\gamma_t\alpha_t c_h^2\|H(\theta_t)\|^2}_{J_3} \tag{F.66}$$

where by choosing the step sizes $\alpha_t \leq \frac{1}{2\ell_{FH,1} + \ell_{H^2,1,t}}$, $\gamma_t \leq \min\left\{\frac{1}{4\|A_{ag}\|^2 M\ell_f^2}, \frac{1}{2c_h}\right\}$, it holds that

$$J_1 \leq \frac{1}{4}\alpha_t\|d_t\|^2, \quad J_2 \leq \frac{1}{4}\alpha_t\|d_t\|^2, \quad J_3 \leq \frac{1}{2}\alpha_t c_h\|H(\theta_t)\|^2. \tag{F.67}$$

Plugging (F.67) into (F.66), and rearranging, we have

$$\mathbb{V}_{f,t+1} - \mathbb{V}_{f,t} + \mathbb{V}_{\lambda_f,t+1} - \mathbb{V}_{\lambda_f,t} + \mathbb{V}_{h,t+1} - \mathbb{V}_{h,t}$$
$$\leq -\frac{1}{2}\alpha_t\|d_t\|^2 - \frac{1}{2}\alpha_t c_h\|H(\theta_t)\|^2 - (\alpha_t + \gamma_t)\nabla_{\lambda_h}\varphi(\lambda_t;\theta_t)^\top H(\theta_t)$$
$$\leq -\frac{1}{2}\alpha_t\|d_t\|^2 - \frac{1}{4}\alpha_t c_h\|H(\theta_t)\|^2 + \frac{(1+c_{\gamma,\alpha})^2}{c_h}\alpha_t\|\nabla_{\lambda_h}\varphi(\lambda_t;\theta_t)\|^2 \tag{F.68}$$

where the last inequality follows from Cauchy-Schwarz inequality and that $\gamma_t = c_{\gamma,\alpha}\alpha_t$.

By applying Corollary 31 with $\gamma_t \leq \ell_{\varphi_\lambda,1}^{-1}$, $\alpha_t \leq \frac{1}{(\ell_{\varphi_\theta,1}+\ell_{\varphi^*,1})c_{\varphi,d}}$ and $c_{\gamma,\alpha} \geq \frac{2c_{\varphi,d}\ell_{\varphi_\theta,1}^2\mu'_\varphi}{\mu_\varphi}$, we further have that

$$\mathbb{V}_{\varphi,t+1} - \mathbb{V}_{\varphi,t} \leq -\frac{1}{2}\mu_\varphi\gamma_t\big(\varphi(\lambda_t;\theta_t) - \varphi(\lambda^*(\theta_t);\theta_t)\big) + \frac{\alpha_t}{c_{\varphi,d}}\|d_t\|^2. \tag{F.69}$$

Then note that from (F.11) we have $\|\nabla_{\lambda_h}\varphi(\lambda_t;\theta_t)\|^2 \leq 2\ell_{\varphi_\lambda,1}\big(\varphi(\lambda_t;\theta_t) - \varphi(\lambda^*(\theta_t);\theta_t)\big)$. Adding up (F.69) and (F.68) with properly chosen hyperparameters $c_h \geq (1+c_{\gamma,\alpha})^2$, $c_{\gamma,\alpha} \geq \frac{4\ell_{\varphi_\lambda,1}}{\mu_\varphi}$, and $c_{\varphi,d} = 4$ yields

$$\mathbb{V}_{t+1} - \mathbb{V}_t \leq -\frac{1}{4}\alpha_t\|d_t\|^2 - \frac{1}{4}\alpha_t c_h\|H(\theta_t)\|^2.$$

Taking telescoping sum of the above inequality over $t = 0, \ldots, T-1$ yields

$$\sum_{t=0}^{T-1}\alpha_t\Big(\|d_t\|^2 + c_h\|H(\theta_t)\|^2\Big) \leq 4(\mathbb{V}_0 - \mathbb{V}_T)$$

$$\leq 4\mathbb{V}_{f,0} + 4\big(\lambda_{h,0}^\top H(\theta_0) - \lambda_{h,T}^\top H(\theta_T)\big) + 2\|H(\theta_0)\|^2 + 4\mathbb{V}_{\varphi,0} \leq 2c_0 + 8c_{\lambda_h}c_H \tag{F.70}$$

where the second last inequality follows from $\mathbb{V}_{f,t} \geq 0$, choosing $\lambda_f = \lambda_{f,0}$, and $\mathbb{V}_{\varphi,t} \geq 0$, the last inequality follows from choosing $\theta_0, \lambda_0$ such that $\lambda_f^\top AF(\theta_0), H(\theta_0), \varphi(\lambda_0;\theta_0) - \varphi(\lambda^*(\theta_0);\theta_0)$ are bounded, thus $2\mathbb{V}_{f,0} + \|H(\theta_0)\|^2 + 2\mathbb{V}_{\varphi,0} \leq c_0 < \infty$, $\lambda_{h,t}$ are bounded on the trajectory, and $\|H(\theta_0)\|_1, \|H(\theta_T)\|_1 \leq c_H$, thus $4\big(\lambda_{h,0}^\top H(\theta_0) - \lambda_{h,T}^\top H(\theta_T)\big) \leq 8c_{\lambda_h}c_H$.

We then summarize the best possible choices for $\alpha_t, \gamma_t$. Recall that we require $\alpha_t \leq \frac{1}{2\ell_{FH,1}+\ell_{H^2,1,t}}$. Rearranging this inequality with $\ell_{H^2,1,t} = 2M\|B_h\|^2\ell_f^2 + 2(\alpha_t^{-1} + \ell_H)c_{\alpha,h}\sqrt{M}\ell_{f,1}\|B_h\|$, and choosing $c_{\alpha,h} = \frac{1}{4\sqrt{M}\ell_{f,1}\|B_h\|}$ yield

$$\alpha_t(2\ell_{FH,1} + \ell_{H^2,1,t}) = 2\alpha_t\ell_{FH,1} + 2\alpha_t M\|B_h\|^2\ell_f^2 + \frac{1}{2}(1 + \alpha_t\ell_H) \leq 1. \tag{F.71}$$

Then we can choose the following to ensure the above inequality holds

$$\alpha_t \leq \frac{1}{4\big(\ell_{FH,1} + M\|B_h\|^2\ell_f^2\big) + \ell_H}. \tag{F.72}$$

To summarize, we can choose the following hyperparameters and step sizes

$$c_{\gamma,\alpha} \geq \max\Big\{\frac{8\ell_{\varphi_\theta,1}^2\mu'_\varphi}{\mu_\varphi}, \frac{4\ell_{\varphi_\lambda,1}}{\mu_\varphi}\Big\}, \quad c_h = (1+c_{\gamma,\alpha})^2, \quad c_{\alpha,h} = \frac{1}{4\sqrt{M}\ell_{f,1}\|B_h\|} \tag{F.73a}$$

$$\gamma_t = c_{\gamma,\alpha}\alpha_t, \quad \text{and} \quad \alpha_t = \min\Big\{\frac{c_{\alpha,h}}{\max\{\|H(\theta_t)\|, \ell_{f,1}\|A_{ag}^\top\|_1(1+c_{\lambda_h})\}}, \frac{1}{4(\ell_{\varphi_\theta,1}+\ell_{\varphi^*,1})},$$

$$\frac{1}{c_{\gamma,\alpha}\ell_{\varphi_\lambda,1}}, \frac{1}{4c_{\gamma,\alpha}\|A_{ag}\|^2 M\ell_f^2}, \frac{1}{2c_{\gamma,\alpha}c_h}, \frac{1}{4\big(\ell_{FH,1} + M\|B_h\|^2\ell_f^2\big) + \ell_H}\Big\}, \tag{F.73b}$$

where $\ell_{FH,1} = \ell_{f,1}(\|A^\top\|_1 + \|B_h^\top\|_1 c_{\lambda_h})$, and $\ell_H = \|B_h\|\sqrt{M}\ell_f$. Then it holds that

$$\sum_{t=0}^{T-1}\alpha_t\Big(\|d_t\|^2 + c_h\|H(\theta_t)\|^2\Big) = \mathcal{O}(1). \tag{F.74}$$

Therefore, $\alpha_t c_h\|H(\theta_t)\|^2$ are bounded for all $t = 0, \ldots, T$. Combining with Lemma 32, we have $\|H(\theta_t)\|$ are bounded for all $t = 0, \ldots, T$, thus we can choose $\alpha_t = \Omega(1)$, i.e., $\alpha_t$ is lower bounded by a constant.

Collecting the results above, we have proved that we can choose $\alpha_t = \Theta(1)$, $\gamma_t = \Theta(1)$ such that

$$\frac{1}{T}\sum_{t=0}^{T-1}\Big(\|d_t\|^2 + \|H(\theta_t)\|^2\Big) = \mathcal{O}\Big(\frac{1}{T}\Big). \tag{F.75}$$

The proof is complete. $\qquad\square$

# G  Stochastic Algorithms

In this section, we discuss the single-loop stochastic algorithm and its convergence guarantees. Note that, the extension of the analysis of the double-loop algorithm, i.e., Algorithm 1 and the extension of the single-loop algorithm analysis in Theorem 2 to their stochastic variants with double sampling as used in [7], are rather straightforward, thus we ommit the discussion in this paper, and only focus on the *single-loop stochastic* algorithm with equality constraints only, i.e., $M_g = 0$, and with a sharper analysis as an extension of Theorem 3.

Let $\xi$ and $\xi'$ be i.i.d. random variables. The stochastic constrained vector optimization problem is defined as

$$\min_{\theta \in \mathbb{R}^q} F(\theta) := \mathbb{E}[F_\xi(\theta)], \quad \text{s.t.} \quad H(\theta) := \mathbb{E}[H_{\xi'}(\theta)] = 0, \quad \text{with} \quad H_{\xi'}(\theta) = B_h F_{\xi'}(\theta) + b_h. \quad \text{(G.1)}$$

## G.1  Algorithm summary

The stochastic algorithm is summarized in Algorithm 3. Note that, instead of computing $\nabla F_{\xi_{t,1}}(\theta_t), \nabla F_{\xi_{t,2}}(\theta_t)$, which requires $2M$ gradient computation at each iteration, we compute $\nabla F_{\xi_{t,1}}(\theta_t), \nabla\big(F_{\xi_{t,2}}(\theta_t)A_{ag}^\top\lambda_t\big)$, which requires $M + 1$ gradient computation per iteration. This saves nearly half of the per-iteration complexity compared to the most relevant existing stochastic algorithm for multi-objective optimization [7]. Furthermore, with the gradient-based single-loop update for $\lambda_t$, the approximation approach proposed in [34, Section 3.2] can be further applied to largely reduce the per-iteration complexity, which we leave for future work.

---

**Algorithm 3** Stochastic FERERO-SA

---

1: Initialize $t = 0$, $\theta_0$, $\lambda_0$, step sizes $\alpha_t$, $\gamma_t$;
2: **for** $t = 0, \ldots, T-1$ **do**
3:      Compute the stochastic gradients $\nabla F_{\xi_{t,2}}(\theta_t), \nabla F_{\xi_{t,1}}(\theta_t)A_{ag}^\top\lambda_t$;
4:      Compute the stochastic estimate of the constraint $H_{\xi_{t,1}}(\theta_t)$;
5:      Compute an update direction $d_t = \nabla F_{\xi_{t,1}}(\theta_t)A_{ag}^\top\lambda_t$;
6:      Choose the step size $\alpha_t$ by a predefined schedule;
7:      Update $\theta_t$ by $\theta_{t+1} = \theta_t + \alpha_t d_t$;
8:      Update $\lambda_t$ by (3.4);
9: **end for**

---

## G.2  Proof of Theorem 4: convergence of Algorithm 3

We first introduce the supporting lemmas, and then present the main proofs. Denote $\mathcal{F}_t$ as the $\sigma$-algebra generated by $\nabla F_{\xi_0}(\theta_0), \nabla F_{\xi_1}(\theta_1), \ldots, \nabla F_{\xi_t}(\theta_t)$, where $\xi_t = \{\xi_{t,1}, \xi_{t,2}\}$. For brevity, we let $\mathbb{E}_t[\cdot] := \mathbb{E}[\cdot \mid \mathcal{F}_{t-1}]$. Also recall that $\tilde{\nabla}$ is the unbiased stochastic estimate of the gradient.

We make the following additional assumptions for proof of convergence.

**Assumption 5.** *For $\{\theta_t, \lambda_t\}_{t=0}^{T-1}$ on the trajectory of Algorithm 3, it holds that*
*1. The variance of $\nabla F_{\xi_t}(\theta_t)$ is bounded by $\sigma^2$.*
*2. The variance of $\tilde{\nabla}_\lambda \varphi(\lambda_t; \theta_t)$ is bounded by $\gamma_t \sigma^2$.*
*3. The function $\|H(\theta_t)\|$ is bounded by $c_H$.*

Note that the bounded variance assumption is common in optimization literature. However, for sharp analysis here, we additionally require $\tilde{\nabla}_\lambda \varphi(\lambda_t; \theta_t)$ has reduced variance in the order of $\mathcal{O}(\gamma_t)$, which can be achieved using a large batch size. Note that, even without assuming reduced variance, i.e., Assumption 5-2, the stochastic algorithm still converges, which can be proved by extending Theorem 2 to the stochastic case using the same techniques in [7] for MoDo, a double-sampling-based single-loop stochastic variant of MGDA. However, the convergence rate will be slower than $\mathcal{O}(T^{-\frac{1}{2}})$. Here we use this additional assumption to achieve a faster convergence rate.

The following Lemma 34 extends Lemma 17 to the stochastic case.

**Lemma 34.** *Let $\lambda_t = [\lambda_{f,t}; \lambda_{h,t}]$. Consider the stochastic sequence $\{\lambda_t\}_{t=0}^{T}$ produced by Algorithm 3. Then for all $\lambda = [\lambda_f; \lambda_h] \in \Omega_\lambda$, it holds that*

$$2\gamma_t \mathbb{E}_t[\langle \lambda_{f,t} - \lambda_f, \nabla_{\lambda_f}\varphi(\lambda_t;\theta_t)\rangle] \leq \mathbb{E}_t[\|\lambda_{f,t} - \lambda_f\|^2 - \|\lambda_{f,t+1} - \lambda_f\|^2 + \gamma_t^2\|\tilde\nabla_{\lambda_f}\varphi(\lambda_t;\theta_t)\|^2];$$
$$2\gamma_t \mathbb{E}_t[\langle \lambda_{h,t} - \lambda_h, \nabla_{\lambda_h}\varphi(\lambda_t;\theta_t)\rangle] \leq \mathbb{E}_t[\|\lambda_{h,t} - \lambda_h\|^2 - \|\lambda_{h,t+1} - \lambda_h\|^2 + \gamma_t^2\|\tilde\nabla_{\lambda_h}\varphi(\lambda_t;\theta_t)\|^2].$$
$$\text{(G.2)}$$

*Proof.* By the update of $\lambda$, it holds that

$$\|\lambda_{f,t+1} - \lambda_f\|^2 \leq \|\lambda_{f,t} - \gamma_t\tilde\nabla_{\lambda_f}\varphi(\lambda_t;\theta_t) - \lambda_f\|^2$$
$$= \|\lambda_{f,t} - \lambda_f\|^2 - 2\gamma_t\langle \lambda_{f,t} - \lambda_f, \tilde\nabla_{\lambda_f}\varphi(\lambda_t;\theta_t)\rangle + \gamma_t^2\|\tilde\nabla_{\lambda_f}\varphi(\lambda_t;\theta_t)\|^2. \qquad \text{(G.3)}$$

Taking expectation over the stochastic samples and rearranging the above inequality, we have

$$2\gamma_t\mathbb{E}_t[\langle \lambda_{f,t} - \lambda_f, \nabla_{\lambda_f}\varphi(\lambda_t;\theta_t)\rangle] = 2\gamma_t\mathbb{E}_t[\langle \lambda_{f,t} - \lambda_f, \nabla_{\lambda_f}\varphi(\lambda_t;\theta_t)\rangle]$$
$$\leq \mathbb{E}_t[\|\lambda_{f,t} - \lambda_f\|^2 - \|\lambda_{f,t+1} - \lambda_f\|^2 + \gamma_t^2\|\tilde\nabla_{\lambda_f}\varphi(\lambda_t;\theta_t)\|^2]. \qquad \text{(G.4)}$$

Following similar arguments, it holds that

$$2\gamma_t\mathbb{E}_t[\langle \lambda_{h,t} - \lambda_h, \nabla_{\lambda_h}\varphi(\lambda_t;\theta_t)\rangle]$$
$$\leq \mathbb{E}_t[\|\lambda_{h,t} - \lambda_h\|^2 - \|\lambda_{h,t+1} - \lambda_h\|^2 + \gamma_t^2\|\tilde\nabla_{\lambda_h}\varphi(\lambda_t;\theta_t)\|^2]. \qquad \text{(G.5)}$$

The proof is complete. $\qquad\qquad\square$

**Lemma 35** (Restatement of [50, Lemma 2]). *Let $\bar\varphi(x) = \varphi(x) + h(x)$, where $\varphi : \mathbb{R}^q \to \mathbb{R}$ is $L$-smooth, and $h : \mathbb{R}^q \to \mathbb{R}$ is nonsmooth but convex and relatively simple. Define $y = \mathrm{prox}_{\gamma h}(x - \gamma d')$ for some $d' \in \mathbb{R}^q$. Then for $y$, the following inequality holds for all $z \in \mathbb{R}^q$:*

$$\bar\varphi(y) \leq \bar\varphi(z) + \langle y - z, \nabla\varphi(x) - d'\rangle$$
$$+ \left(\frac{L}{2} - \frac{1}{2\gamma}\right)\|y - x\|^2 + \left(\frac{L}{2} + \frac{1}{2\gamma}\right)\|z - x\|^2 - \frac{1}{2\gamma}\|y - z\|^2. \qquad \text{(G.6)}$$

The following Lemma 36 extends Lemma 30 to the stochastic case.

**Lemma 36** (Error of subprogram in the stochastic setting). *Suppose Assumptions 1, 2, 3, 4, 5 hold, and $M_g = 0$. Let $\{\theta_t\}, \{\lambda_t\}$ be the sequences produced by Algorithm 3 with step size $\gamma_t \leq \ell_{\varphi_\lambda,1}^{-1}$. Then for any $c_{\varphi,d} > 0$, the following hold*

$$\mathbb{E}[\varphi(\lambda_{t+1};\theta_t) - \varphi(\lambda^*(\theta_t);\theta_t)] \leq (1 - \gamma_t\mu_\varphi)\mathbb{E}[\varphi(\lambda_t;\theta_t) - \varphi(\lambda^*(\theta_t);\theta_t)] + \gamma_t^2\sigma^2 \qquad \text{(G.7a)}$$

$$\mathbb{E}[\varphi(\lambda_{t+1};\theta_{t+1}) - \varphi(\lambda^*(\theta_{t+1});\theta_{t+1})] \leq \left(1 + \alpha_t c_{\varphi,d}\ell_{\varphi_\theta,1}^2\mu_\varphi'\right)\mathbb{E}\Big[\varphi(\lambda_{t+1};\theta_t) - \varphi(\lambda^*(\theta_t);\theta_t)\Big]$$
$$+ \left(\frac{\alpha_t}{2c_{\varphi,d}} + \frac{\ell_{\varphi_\theta,1} + \ell_{\varphi^*,1}}{2}\alpha_t^2\right)\mathbb{E}[\|\nabla F(\theta_t)A_{ag}^\top\lambda_t\|^2] + \frac{\ell_{\varphi_\theta,1} + \ell_{\varphi^*,1}}{2}\alpha_t^2\sigma^2. \qquad \text{(G.7b)}$$

*Proof of Lemma 36.* The proof follows most of that of Lemma 30. We highlight the difference.

First we define $\lambda'_{t+1} = \Pi_{\Omega_\lambda}(\lambda_t - \nabla_\lambda\varphi(\lambda_t;\theta_t))$ as an auxiliary variable. By the $\ell_{\varphi_\lambda,1}$-smoothness of $\varphi(\cdot;\theta)$, we have

$$\mathbb{E}[\varphi(\lambda'_{t+1};\theta_t)] \leq \mathbb{E}[\varphi(\lambda_t;\theta_t) + \left(\frac{\ell_{\varphi_\lambda,1}}{2} - \frac{1}{\gamma_t}\right)\|\lambda'_{t+1} - \lambda_t\|^2]. \qquad \text{(G.8)}$$

Applying Lemma 35 with $y = \lambda_{t+1}, z = \lambda'_{t+1}, x = \lambda_t$, and that $\lambda_t, \lambda_{t+1}, \lambda'_{t+1} \in \Omega_\lambda$ yields

$$\mathbb{E}[\varphi(\lambda_{t+1};\theta_t)] \leq \mathbb{E}\Big[\varphi(\lambda'_{t+1};\theta_t) + \langle \lambda_{t+1} - \lambda'_{t+1}, \nabla_\lambda\varphi(\lambda_t;\theta_t) - \tilde\nabla_\lambda\varphi(\lambda_t;\theta_t)\rangle$$
$$+ \left(\frac{\ell_{\varphi_\lambda,1}}{2} - \frac{1}{2\gamma_t}\right)\|\lambda_{t+1} - \lambda_t\|^2 + \left(\frac{\ell_{\varphi_\lambda,1}}{2} + \frac{1}{2\gamma_t}\right)\|\lambda'_{t+1} - \lambda_t\|^2 - \frac{1}{2\gamma_t}\|\lambda_{t+1} - \lambda'_{t+1}\|^2\Big]. \qquad \text{(G.9)}$$

Furthermore, following similar arguments as (F.49), by Assumption 4-1, and taking total expectation, we have

$$\mathbb{E}[\varphi(\lambda'_{t+1};\theta_t)] \leq \mathbb{E}\Big[\varphi(\lambda_t;\theta_t) - \gamma_t\mu_\varphi\Big(\varphi(\lambda_t;\theta_t) - \varphi(\lambda^*(\theta_t);\theta_t)\Big)\Big]. \tag{G.10}$$

Adding up $\frac{2}{3}\times$ (G.8), $1\times$ (G.9), and $\frac{1}{3}\times$ (G.10) yields

$$\mathbb{E}[\varphi(\lambda_{t+1};\theta_t)] \leq \mathbb{E}\Big[\varphi(\lambda_t;\theta_t) + (\frac{5\ell_{\varphi_\lambda,1}}{6} - \frac{1}{6\gamma_t})\|\lambda'_{t+1} - \lambda_t\|^2 + (\frac{\ell_{\varphi_\lambda,1}}{2} - \frac{1}{2\gamma_t})\|\lambda_{t+1} - \lambda_t\|^2$$
$$- \frac{\mu_\varphi\gamma_t}{3}\big(\varphi(\lambda_t;\theta_t) - \varphi(\lambda^*(\theta_t);\theta_t)\big) - \frac{1}{2\gamma_t}\|\lambda_{t+1} - \lambda'_{t+1}\|^2$$
$$+ \langle\lambda_{t+1} - \lambda'_{t+1}, \nabla_\lambda\varphi(\lambda_t;\theta_t) - \tilde{\nabla}_\lambda\varphi(\lambda_t;\theta_t)\rangle\Big]. \tag{G.11}$$

Choosing $\gamma_t \geq \frac{1}{\ell_{\varphi_\lambda,1}} \geq \frac{1}{5\ell_{\varphi_\lambda,1}}$ and applying Cauchy-Schwarz and Young's inequality, we have

$$\mathbb{E}[\varphi(\lambda_{t+1};\theta_t)] \leq \mathbb{E}\Big[\varphi(\lambda_t;\theta_t) - \frac{\mu_\varphi\gamma_t}{3}\big(\varphi(\lambda_t;\theta_t) - \varphi(\lambda^*(\theta_t);\theta_t)\big)$$
$$+ \frac{\gamma_t}{2}\|\nabla_\lambda\varphi(\lambda_t;\theta_t) - \tilde{\nabla}_\lambda\varphi(\lambda_t;\theta_t)\|^2\Big]. \tag{G.12}$$

The first inequality is proved. We then prove the second inequality. Note that (F.50) still holds here. Following similar arguments in (F.51), $\mathbb{E}[J_1]$ in (F.50) can be further bounded by

$$\mathbb{E}[J_1] \leq \mathbb{E}\Big[-\alpha_t\langle\nabla_\theta\varphi(\lambda_{t+1};\theta_t) - \nabla\varphi(\lambda^*(\theta_t);\theta_t), d_t\rangle + \frac{\ell_{\varphi_\theta,1} + \ell_{\varphi^*,1}}{2}\alpha_t^2\|d_t\|^2\Big]$$
$$\leq \mathbb{E}\Big[-\alpha_t\langle\nabla_\theta\varphi(\lambda_{t+1};\theta_t) - \nabla\varphi(\lambda^*(\theta_t);\theta_t), \nabla F(\theta_t)A_{ag}^\top\lambda_t\rangle + \frac{\ell_{\varphi_\theta,1} + \ell_{\varphi^*,1}}{2}\alpha_t^2\|d_t\|^2\Big]$$
$$\leq \mathbb{E}\Big[\alpha_t\|\nabla_\theta\varphi(\lambda_{t+1};\theta_t) - \nabla\varphi(\lambda^*(\theta_t);\theta_t)\|\|\nabla F(\theta_t)A_{ag}^\top\lambda_t\| + \frac{\ell_{\varphi_\theta,1} + \ell_{\varphi^*,1}}{2}\alpha_t^2\|d_t\|^2\Big]. \tag{G.13}$$

Then following similar arguments in (F.52) and (F.53), we have

$$\mathbb{E}[J_1] \leq \mathbb{E}\Big[\alpha_t c_{\varphi,d}\ell_{\varphi_\theta,1}^2\mu'_\varphi\big(\varphi(\lambda_{t+1};\theta_t) - \varphi(\lambda^*(\theta_t);\theta_t)\big)$$
$$+ \frac{\alpha_t}{2c_{\varphi,d}}\|\nabla F(\theta_t)A_{ag}^\top\lambda_t\|^2 + \frac{\ell_{\varphi_\theta,1} + \ell_{\varphi^*,1}}{2}\alpha_t^2\|d_t\|^2\Big]. \tag{G.14}$$

Plugging (G.14) back into (F.50) with total expectation completes the proof of the second inequality. □

Next we proceed to state and prove Theorem 4, which generalizes Theorem 3 to its stochastic variants, with a matching convergence rate to the unconstrained stochastic MOO algorithms and stochastic gradient descent. This allows us to apply the algorithm to large-scale machine learning problems, which we detail in Section 5. Its proof also extends that of Theorem 3. We ommit the similar derivations and only highlight the difference.

**Theorem 4** (Convergence of the single-loop stochastic FERERO algorithm). *Suppose Assumptions 1, 2, 3, 4, 5 hold, and $M_g = 0$. Let $\{\theta_t\}$, $\{\lambda_t\}$ be the sequences produced by Algorithm 3 with $A = I$ and $\Omega_{\lambda_f}(\theta) = \Delta^M$ (c.f. Remark 4). With properly chosen step sizes $\alpha_t = \alpha = \Theta(T^{-\frac{1}{2}})$, $\gamma_t = \gamma = \Theta(T^{-\frac{1}{2}})$, it holds that*

$$\frac{1}{T}\sum_{t=0}^{T-1}\mathbb{E}\Big[\|\nabla F(\theta_t)A_{ag}^\top\lambda_t\|^2 + \|H(\theta_t)\|^2\Big] = \mathcal{O}\Big(T^{-\frac{1}{2}}\Big). \tag{G.15}$$

*Proof of Theorem 4.* Reuse the Lyapunov functions defined in (F.57). Let $\lambda_t = [\lambda_{f,t}; \lambda_{h,t}]$. The algorithm takes the update $\theta_{t+1} = \theta_t + \alpha_t d_t$ with $d_t = \nabla F_{\xi_{t,1}}(\theta_t)A_{ag}^\top\lambda_t$. From Lemma 9, the function $\lambda_f^\top AF(\theta)$ is $\ell_{f,1}\|A^\top\|_1$-smooth. Then following similar arguments from (F.18)-(F.20),

choosing $\frac{\alpha_t}{\gamma_t} = \frac{\alpha_0}{\gamma_0} = \frac{1}{c_{\gamma,\alpha}}$ for all $t \in [T]$, and taking total expectation, we have the stochastic version of (F.58) below

$$\mathbb{E}[\mathbb{V}_{f,t+1} - \mathbb{V}_{f,t} + \mathbb{V}_{\lambda_f,t+1} - \mathbb{V}_{\lambda_f,t}]$$

$$\leq \frac{\ell_{f,1}\|A^\top\|_1}{2}\alpha_t^2\mathbb{E}[\|d_t\|^2] + \frac{1}{2}\alpha_t\gamma_t\mathbb{E}[\|\tilde{\nabla}_{\lambda_f}\varphi(\lambda_t;\theta_t)\|^2] - \alpha_t\mathbb{E}[\langle\lambda_{f,t},\nabla_{\lambda_f}\varphi(\lambda_t;\theta_t)\rangle]. \quad \text{(G.16)}$$

The stochastic version of (F.59) is

$$\mathbb{E}[\mathbb{V}_{h,0,t+1} - \mathbb{V}_{h,0,t}] \leq -\alpha_t c_h\mathbb{E}[\lambda_{h,t}^\top H(\theta_t)] + \frac{\ell_{f,1}}{2}\alpha_t^2\|B_h^\top\|_1 c_{\lambda_h}\mathbb{E}[\|d_t\|^2]$$

$$-\alpha_t\mathbb{E}[\langle\lambda_{h,t},\nabla_{\lambda_h}\varphi(\lambda_t;\theta_t)\rangle] - \gamma_t\mathbb{E}[\tilde{\nabla}_{\lambda_h}\varphi(\lambda_t;\theta_t)^\top H(\theta_t)]. \quad \text{(G.17)}$$

By Lemma 24, and that $\nabla_{\lambda_h}\varphi(\lambda_t;\theta_t) = -\nabla H(\theta_t)^\top\nabla F(\theta_t)A_{ag}^\top\lambda_t - c_h H(\theta_t)$, the stochastic version of (F.63) is

$$\mathbb{E}[\mathbb{V}_{h,3,t+1} - \mathbb{V}_{h,3,t}] \leq \alpha_t\mathbb{E}[H(\theta_t)^\top\nabla H(\theta_t)^\top\nabla F_{\xi_{t,1}}(\theta_t)A_{ag}^\top\lambda_t] + \frac{1}{4}\alpha_t^2\mathbb{E}[\ell_{H^2,1,t}\|d_t\|^2]$$

$$\leq -\alpha_t c_h\mathbb{E}[\|H(\theta_t)\|^2] - \alpha_t\mathbb{E}[H(\theta_t)^\top\nabla_{\lambda_h}\varphi(\lambda_t;\theta_t)] + \frac{1}{4}\alpha_t^2\mathbb{E}[\ell_{H^2,1,t}\|d_t\|^2] \quad \text{(G.18)}$$

where $\ell_{H^2,1,t} = \ell_{H^2,1} = 2M\|B_h\|^2\ell_f^2 + 2(c_H + \ell_H c_{\alpha,h})\sqrt{M}\ell_{f,1}\|B_h\|$ by Assumption 5-3, and $\ell_H = \|B_h\|\sqrt{M}\ell_f$. Let $\ell_{FH,1} = \ell_{f,1}(\|A^\top\|_1 + \|B_h^\top\|_1 c_{\lambda_h})$. Adding up (G.16), (G.17), and (G.18) yields that

$$\mathbb{E}[\mathbb{V}_{f,t+1} - \mathbb{V}_{f,t} + \mathbb{V}_{\lambda_f,t+1} - \mathbb{V}_{\lambda_f,t} + \mathbb{V}_{h,t+1} - \mathbb{V}_{h,t}]$$

$$\leq \frac{1}{4}\alpha_t^2(2\ell_{FH,1} + \ell_{H^2,1})\mathbb{E}[\|d_t\|^2] - \alpha_t\mathbb{E}[\langle\lambda_t,\nabla_\lambda\varphi(\lambda_t;\theta_t)\rangle] - \alpha_t c_h\mathbb{E}[\lambda_{h,t}^\top H(\theta_t)]$$

$$-\alpha_t c_h\mathbb{E}[\|H(\theta_t)\|^2] - (\alpha_t + \gamma_t)\mathbb{E}[H(\theta_t)^\top\nabla_{\lambda_h}\varphi(\lambda_t;\theta_t)] + \frac{1}{2}\alpha_t\gamma_t\mathbb{E}[\|\tilde{\nabla}_{\lambda_f}\varphi(\lambda_t;\theta_t)\|^2]. \quad \text{(G.19)}$$

Further rearranging the above inequality, applying (F.65), invoking that $\gamma_t = c_{\gamma,\alpha}\alpha_t$, and choosing $\alpha_t \leq \min\left\{\frac{1}{2\ell_{FH,1}+\ell_{H^2,1}}, \frac{1}{4\|A_{ag}\|^2 M\ell_f^2 c_{\gamma,\alpha}}, \frac{1}{2c_h c_{\gamma,\alpha}}\right\}$, we have

$$\mathbb{E}[\mathbb{V}_{f,t+1} - \mathbb{V}_{f,t} + \mathbb{V}_{\lambda_f,t+1} - \mathbb{V}_{\lambda_f,t} + \mathbb{V}_{h,t+1} - \mathbb{V}_{h,t}]$$

$$\leq -\frac{1}{2}\alpha_t\mathbb{E}[\|\nabla F(\theta_t)A_{ag}^\top\lambda_t\|^2] - \frac{1}{4}\alpha_t c_h\mathbb{E}[\|H(\theta_t)\|^2] + \frac{(1+c_{\gamma,\alpha})^2}{c_h}\alpha_t\mathbb{E}[\|\nabla_{\lambda_h}\varphi(\lambda_t;\theta_t)\|^2]$$

$$+ \alpha_t^3 c_{\gamma,\alpha}^2\sigma^2 + \frac{1}{4}(2\ell_{FH,1} + \ell_{H^2,1})\alpha_t^2\sigma^2. \quad \text{(G.20)}$$

By applying Lemma 36, and choosing $\alpha_t \leq \min\left\{\frac{1}{\ell_{\varphi_\lambda,1}c_{\gamma,\alpha}}, \frac{1}{(\ell_{\varphi_\theta,1}+\ell_{\varphi^*,1})c_{\varphi,d}}, \frac{\mu_\varphi}{c_{\varphi,d}\ell_{\varphi_\theta,1}^2}\right\}$ and $c_{\gamma,\alpha} \geq \frac{2c_{\varphi,d}\ell_{\varphi_\theta,1}^2\mu_\varphi'}{\mu_\varphi}$, the stochastic version of (F.69) is

$$\mathbb{E}[\mathbb{V}_{\varphi,t+1} - \mathbb{V}_{\varphi,t}] \leq -\frac{1}{2}\mu_\varphi\gamma_t\mathbb{E}[\varphi(\lambda_t;\theta_t) - \varphi(\lambda^*(\theta_t);\theta_t)]$$

$$+ \frac{\alpha_t}{c_{\varphi,d}}\mathbb{E}[\|\nabla F(\theta_t)A_{ag}^\top\lambda_t\|^2] + (1 + 2c_{\gamma,\alpha}^2)\alpha_t^2\sigma^2. \quad \text{(G.21)}$$

Adding up (G.20) and (G.21) with properly chosen hyperparameters $c_h \geq (1+c_{\gamma,\alpha})^2$, $c_{\gamma,\alpha} \geq \frac{4\ell_{\varphi_\lambda,1}}{\mu_\varphi}$, and $c_{\varphi,d} = 4$, we have

$$\mathbb{E}[\mathbb{V}_{t+1} - \mathbb{V}_t] \leq -\frac{1}{4}\alpha_t\mathbb{E}[\|\nabla F(\theta_t)A_{ag}^\top\lambda_t\|^2] - \frac{1}{4}\alpha_t c_h\mathbb{E}[\|H(\theta_t)\|^2]$$

$$+ \left(1 + 2c_{\gamma,\alpha}^2 + \alpha_t c_{\gamma,\alpha}^2 + \frac{1}{4}(2\ell_{FH,1} + \ell_{H^2,1})\right)\alpha_t^2\sigma^2. \quad \text{(G.22)}$$

With the same hyperparameters and step sizes summarized in (F.73), one can choose $\alpha = \Theta(T^{-\frac{1}{2}})$, $\gamma = \Theta(T^{-\frac{1}{2}})$ to obtain

$$\frac{1}{T}\sum_{t=0}^{T-1}\mathbb{E}\left[\|\nabla F(\theta_t)A_{ag}^\top\lambda_t\|^2 + \|H(\theta_t)\|^2\right] = \mathcal{O}\left(T^{-\frac{1}{2}}\right). \quad \text{(G.23)}$$

The proof is complete. $\qquad\square$

## H Implementation Details and Additional Experiment Results

In this section, we report the additional implementation details omitted from the main text in Appendix H.1 and the additional experimental results in Appendix H.2.

### H.1 Implementation details

**Computation.** All experiments were conducted on a server with an Intel i9-7920X CPU, two NVIDIA A5000 GPUs and two NVIDIA A4500 GPUs.

For all the experiments reported in the main text except for the multi-lingual speech recognition experiment, we exactly follow the settings from [41]. The implementations of the baselines including LS, PMTL, and EPO are from the official code of the EPO paper in `https://github.com/dbmptr/EPOSearch` with their default hyperparameters. The results of XWC-MGDA are directly referenced from the paper due to lack of official implementation.

**Synthetic data.** For the results in both Figure 3 and Figure 4, the model parameter $\theta$ has dimension $q = 20$, the number of objectives is $M = 2$. The angles between the preference vectors and the horizontal axis are generated between $[\frac{1}{20}\pi, \frac{9}{20}\pi]$ with equal angular distance. This experiment does not involve stochastic optimization. For our method, we solve the subprogram using PGD with a step size $0.1$ up to an error of $10^{-5}$ or with a maximum of 250 iterations. In the experiments, we set the parameter $c_h = 1$ for the subprogram if not otherwise specified.

In Figure 3, for all preferences and all methods, the initial model parameter $\theta_0$ is randomly generated from a Gaussian distribution $\mathcal{N}(0, 1)$ for each dimension. In Table 6, we provide a summary of the hyperparameters for the baselines and our methods for the experiments in Figure 3.

Table 6: Summary of hyper-parameters for the synthetic data experiments in Figure 3.

| Hyperparameters | LS | MGDA | PMTL | EPO | Ours Figure 3e | Ours Figure 3f |
|---|---|---|---|---|---|---|
| step size $\alpha_t$ | 0.1 | 0.2 | 0.2 | 0.1 | 0.05 | 0.05 |
| max iterations | 150 | 150 | 150 | 100 | 100 | 100 |

In Figures 4a-4c, the initial model parameters are randomly generated from a uniform distribution between $[-0.3, 0.3]$ for each dimension. In Figures 4d-4f, the initial model parameters are randomly generated from a uniform distribution between $[-0.5, -0.15]$ or $[0.15, 0.5]$ for each dimension. Table 7 summarizes the hyperparameters for the experiments in Figure 4.

Table 7: Summary of hyper-parameters for the synthetic data experiments in Figure 4.

| Hyperparameters | Figures 4a-4c | | | Figures 4d-4f | | |
|---|---|---|---|---|---|---|
| | PMTL | EPO | Ours | PMTL | EPO | Ours |
| step size $\alpha_t$ | 0.25 | 0.10 | 0.60 | 0.50 | 0.20 | 0.60 |
| max iterations | 100 | 60 | 10 | 200 | 120 | 200 |
| $c_h$ | - | - | 1 | - | - | 0.01 |

**Multi-patch image classification.** For a fair comparison, we follow the same data splitting and processing procedures as [41] using their official code. In each of the three datasets, there are 120k samples for training and 20k samples for testing. There are two tasks on each dataset: 1) classifying the top-left image, and 2) classifying the bottom-right image.

For all methods, we use the SGD optimizer with batch size 256. Note that, for our stochastic method, we use batch size 128 for each batch in the double sampling. Thus the total number of samples taken at each iteration is also 256. The hyperparameters are summarized in Table 8. The results of XWC-MGDA are directly referenced from the paper.

We use the Pymoo 0.6.1 library to compute the hypervolume. The Nadir points, i.e., the worst performance on single task baselines, used for the hypervolume computation are given in Table 9. For a fair comparison, the Nadir points we use are the same with [44] inferred from Figure 4 in the paper.

Table 8: Summary of hyper-parameter choices for multi-patch image classification experiments.

| Hyperparameters | Multi-MNIST | | | | Multi-Fashion | | | | Multi-Fashion+MNIST | | | |
|---|---|---|---|---|---|---|---|---|---|---|---|---|
| | LS | PMTL | EPO | Ours | LS | PMTL | EPO | Ours | LS | PMTL | EPO | Ours |
| step size $\alpha_t$ | 1E-3 | 1E-3 | 1E-3 | 1E-3 | 1E-3 | 1E-3 | 1E-3 | 1E-3 | 1E-3 | 1E-3 | 1E-3 | 1E-3 |
| step size $\gamma_t$ | - | - | - | 1E-4 | - | - | - | 1E-4 | - | - | - | 1E-4 |
| epochs | 100 | 100 | 100 | 100 | 100 | 100 | 100 | 100 | 100 | 100 | 100 | 100 |
| $c_h$ | - | - | - | 0.5 | - | - | - | 0.5 | - | - | - | 0.5 |

Table 9: Nadir points for the hypervolume computation

| Dataset and metrics | Nadir points, metrics on objective $[1, \ldots, M]$ |
|---|---|
| Multi-MNIST loss | [0.500, 0.450] |
| Multi-Fashion loss | [0.840, 0.800] |
| Multi-F+M loss | [0.625, 0.575] |
| Multi-MNIST accuracy | [0.830, 0.848] |
| Multi-Fashion accuracy | [0.840, 0.800] |
| Multi-F+M accuracy | [0.790, 0.785] |

**Multi-lingual speech recognition.** We use two datasets, Librispeech and AISHELL v1. Librispeech is an English speech dataset that consists of 960 hours of labeled audio data. For our experiments, we use the "train-clean-100" subset of the Librispeech dataset for supervised training, which contains 100 hours of clean training data. Additionally, we use the full 960 hours of data for self-supervised training. AISHELL v1 is a 178-hour Mandarin speech corpus designed for various speech and speaker processing tasks. We use the full AISHELL v1 dataset for both self-supervised and supervised training. We combine these two datasets for our multi-lingual speech recognition experiments.

We use the conformer [26] model with 8 conformer blocks as the encoder. Each block contains 512 hidden units and 8 attention heads. Each attention head has dimension 64. The convolutional kernel size is 31. Two classification heads are used. They contain two linear layers, one with 1000 output size for English, and another with 5000 output size for Chinese.

The loss functions we use include the Contrastive Predictive Coding (CPC) loss, and the Connectionist Temporal Classification (CTC) loss. The *CPC loss* [46] is a self-supervised loss to learn robust representations from unlabeled speech data. The CPC loss is designed to maximize the probability of a future sample given a contextual representation generated from the current speech sequence. The *CTC loss* is defined as the negative log-likelihood of the model parameter given the input sequence and the label sequence.

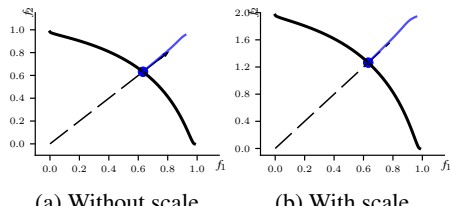

(a) Without scale      (b) With scale

Figure 7: Scale invariance verification.

For all methods including the baselines, we use the step sizes $\alpha_{t,1} = 5 \times 10^{-4}$ for training backbone conformer parameters and $\alpha_{t,2} = 5 \times 10^{-5}$ for training classification head parameters. The step size $\gamma_t = 0.1$ and the parameter $c_h = 0.5$.

## H.2 Additional experiment results

**Synthetic data.** We conduct several additional experiments on the synthetic objectives to further verify our theory. First, we conduct all the experiments on the synthetic objectives reported in the main text, using the single-loop approximate algorithm described in Algorithm 2. The results are plotted in Figure 8. The hyperparameters are the same unless otherwise specified.

From Figure 8a, we can see that Algorithm 2 with a one-step approximate update of $\lambda_t$ also leads to convergence and preference alignment. However, different from the results obtained by exactly solving for $\lambda^*(\theta_t)$ at each iteration, the models on the optimization trajectories do not align exactly with the preference. Similar observations can be found in Figure 8b. In Figure 8c, which is a difficult case due to the initialization, $A = I_M$ does not work since it does not incorporate more general relative preference to allow controlled ascent update. This is addressed in Figure 8d, where a general $A$ (the same as in prior experiments) is used. Compared with exactly solving for $\lambda^*(\theta_t)$ at each

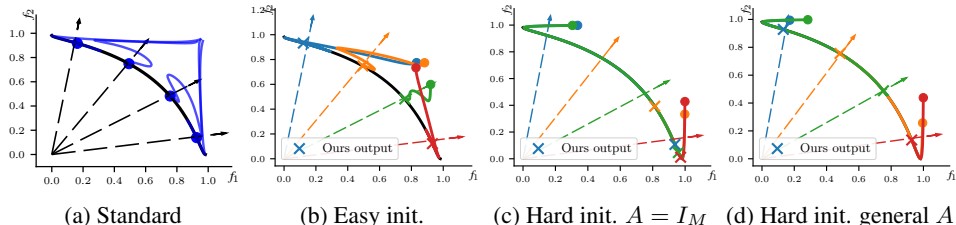

(a) Standard   (b) Easy init.   (c) Hard init. $A = I_M$ (d) Hard init. general $A$

Figure 8: Synthetic experiment results with Algorithm 2.

iteration, the approximate algorithm takes more iterations to converge, but has smaller per-iteration complexity, and smaller total time complexity.

Table 10: Summary of hyper-parameters for the synthetic data experiments in Figure 8.

| Hyperparameters | Figure 8a | Figure 8b | Figure 8c | Figure 8d |
|---|---|---|---|---|
| step size $\alpha_t$ | 0.10 | 0.06 | 0.15 | 0.15 |
| max iterations | 100 | 100 | 250 | 250 |
| $c_h$ | 6 | 6 | 0.1 | 0.1 |

We conduct another experiment to verify that the scale invariance can be preserved. We use the same objective as above, but scale the second one by 2. We use a fixed initialization $\theta_0 = 0.3 \cdot [\mathbf{1}_{q/2}; -\mathbf{1}_{q/2}]$ for this experiment. The other hyperparameters are the same as the default. We use both $F(\theta_0)$ and $F(0)$ as the reference points and choose $B_h$ such that $B_h(F(\theta_0) - F(0)) = 0$. Results in Figure 7 show that for different scales, the trajectory and the converging solution are the same.

Table 11: Summary of average run time in seconds (s) or minutes (m) and number of iterations or epochs of different methods on different datasets. We use Algorithm 1 for the synthetic experiments, and Algorithm 3 for the other two experiments.

| Datasets | Metrics | LS | PMTL | EPO | FERERO |
|---|---|---|---|---|---|
| Synthetic, Figures 3(a-c) | Iterations | 100 | 100 | 60 | 10 |
| | Per-iteration run time | 3.50E-4s | 7.67E-4s | 4.93E-3s | 7.50E-4s |
| | Total run time | 0.035s | 0.0767s | 0.296s | 0.0075s |
| Synthetic, Figures 3(d-f) | Iterations | 100 | 200 | 80 | 200 |
| | Per-iteration run time | 3.10E-4s | 7.65E-4s | 4.93E-3s | 7.30E-4s |
| | Total run time | 0.031s | 0.153s | 0.394s | 0.146s |
| Multi-MNIST/Fashion/F+M | Epochs | 100 | 100 | 100 | 100 |
| | Per-epoch run time | 3.54s | 11.88s | 9.66s | 7.02s |
| | Total run time | 5.9m | 19.8m | 16.1m | 11.7m |

