# OpenReview forum: "FERERO: A Flexible Framework for Preference-Guided Multi-Objective Learning"
_NeurIPS.cc/2024/Conference — NeurIPS 2024 poster_

### Official Review · Reviewer_PTWJ · 2024-07-01

**Soundness:** 2
**Presentation:** 2
**Contribution:** 3
**Rating:** 5
**Confidence:** 3

**Summary:**

This paper considers preference-guided multi-objective learning from the lens of constrained vector optimization. Cone captures relative preferences. Constraints capture absolute preferences. Under boundedness and smoothness assumptions on the objective function, the paper provides gradient-based methods for the identification of Pareto optimal points. The paper provides convergence guarantees for deterministic and stochastic versions of various gradient-based algorithms and illustrates their applicability in a wide range of vector optimization problems.

**Strengths:**

1)	The proposed adaptive iterative update scheme has some nice properties, such as not requiring the initial model and $\theta_t$ to be always feasible, and not requiring different subprograms or different treatment of active inequalities.
2)	Models both relative preferences and absolute preferences.
3)	Proposed iterative methods come with convergence guarantees.

**Weaknesses:**

1) This paper formalizes preference-guided multi-objective learning as a constrained vector optimization problem; however, the vector optimization aspect of the paper is not mentioned in the introduction section. It is not clear what “maximize” means, and it is not clear with respect to what cone the authors maximize. The introduction section gives the impression that the authors consider a multi-objective optimization problem, not a vector optimization problem.

2) Since there can be many solutions in the Pareto set, the set of solutions $\theta$ of PMOL may be very large. An important thing that is not mentioned explicitly in this work is how much of the true Pareto front ${\cal F}$ or the true Pareto set $\\{\theta : F(\theta) \in {\cal F} \\}$ the proposed algorithms capture when they converge. The theoretical results point to the convergence but do not point to what set of solutions the algorithms converge towards. Indeed, many algorithms whose task is to identify a Pareto set of solutions (either for multi-objective or vector optimization) come with guarantees on the returned Pareto set in the form of $(\epsilon,\delta)$-PACness results. Some success conditions that quantify the quality of the returned Pareto set or Pareto front are given in the following works.

Auer, Peter, et al. "Pareto front identification from stochastic bandit feedback." International Conference on Artificial Intelligence and Statistics. PMLR, 2016.

Ararat, Cagin, and Cem Tekin. "Vector optimization with stochastic bandit feedback." International Conference on Artificial Intelligence and Statistics. PMLR, 2023.

The contribution of this work will be much clearer if the authors can demonstrate guarantees for the solutions returned by their algorithms, and compare them with the existing $(\epsilon,\delta)$-PAC guarantees in the related works.

3) Problem setup and preliminaries: “We first introduce new optimality definitions for PMOL that go beyond standard definitions of Pareto optimality”. These are some standard textbook results, not new. See and cite some of the following works.

Boyd, Stephen P., and Lieven Vandenberghe. Convex optimization. Cambridge University Press, 2004.

Jahn, Johannes, ed. Vector optimization. Berlin: Springer, 2009.

Löhne, Andreas. Vector optimization with infimum and supremum. Springer Science & Business Media, 2011.

**Questions:**

1) Section 3.2 is confusing. As far as I understand, the goal of the paper is to optimize the objective function using gradient-based methods for a given ordering cone. First, a method that generates polyhedral cone matrix A from extreme rays of the cone is presented. This is okay, as it is just another way to express the cone. However, the second point, i.e., choosing $C_A$ for controlled ascent, is ambiguous. If the cone encodes the preferences of the decision maker, why do the authors need to change it by adding a new extreme ray?

2) Where are the green dots in Fig. 2 (a)-(e)?

3) What is the difference between the algorithms in Fig. 2 (e) and (f)?

4) What is the difference between the algorithms in Fig. 3 (c) and (f)?

**Limitations:**

Limitations are adequately discussed.

---

> ### Author Rebuttal · Authors · 2024-08-07
>
> >W1. In the introduction, the preference cones are not specified for the two examples, which causes confusion that the problem is not vector optimization.
>
> Thanks for the suggestion. In the introduction, the cone is not specified, it can be any proper cone $C$ pre-defined by the user. Here, since we only intend to give application examples to motivate the problem, we do not intend to introduce the mathematical cone concept at this point.
>
> However, to avoid confusion, we will add the cone $C$ in the revision by replacing Eq. (1.1), and (1.2) with the following equations.
>
> \begin{align}
> &\mathrm{maximize}_ C~~ (f_ {\text{acc}}(\theta), f_ {\text{fair}}(\theta))^ {\top} ~~ \text{  s.t. } f_ {\text {fair }}(\theta) \geq \epsilon; \\\\
> &\mathrm{maximize}_ C ~~ F(\theta):=(f_ 1(\theta), \ldots, f_ M(\theta))^ {\top} ~~ \text { s.t. } B F(\theta)=B v, B v=0 .
> \end{align}
>
> >W2. Comparison with PAC guarantees
>
> Thanks for raising this interesting point! The $(\epsilon, \delta)$-PACness results mentioned by the reviewer in [c,g] guarantee that with probability $1 - \delta$, the algorithm generates Pareto optimal (with some suboptimal) points within a precision $\epsilon$, and with a sample complexity depending on $\epsilon$ and $\delta$.
>
> However, **our results are not directly comparable to [c,g]** since a key difference between our work and [c,g] is that [c,g] study the **a-posteriori** methods to generate a set of Pareto optimal solutions, after which the users specify their preferences and choose from the solutions. However, we study the **a-priori** method where the users first specify their preferences, then algorithms are used to find optimal solutions which satisfy the preferences.
>
> Because of the difference between the studied problems, the PACness results in [c,g] only guarantee convergence to optimality for all the solutions generated by the algorithm, while our results guarantee convergence to **both stationarity and preferences**. When the objectives are convex / strongly convex, stationarity implies weakly Pareto optimality / Pareto optimality.
>
>
>
> We will cite the related works including [c,g], and add the comparison in the revision.
>
> [c] Cagin and Tekin, "Vector optimization with stochastic bandit feedback", AISTATS 2023.
>
> [g] Peter, et al. "Pareto front identification from stochastic bandit feedback", AISTATS 2016.
>
>
> >W3. Preliminaries are not new but standard.
>
> Thanks for the suggestion. We are aware of the standard results, and we have cited the preliminaries (Jahn, 2009) as [18] in Appendix C.1. Here we mean we consider the general cone to be a relative preference that is different from existing works [23,30,33] on preference-guided MOL. We will turn down the tone,  remove the word "new", and cite more references here.
>
>
> >Q1. Clarification of Section 3.2.
>
> The second point is to guide the decision maker to choose an appropriate cone when the cone is **not pre-specified**, or when the current extreme rays are not sufficient to encode the preferences, as discussed in **General Response-2**.
>
> Here "add" means we will specify each extreme rays one by one, and add them to the current set of extreme rays to form a set that represents the cone, when the cone is not pre-specified. We use this method for the experiments in Figures 3 and 5 to specify a cone to allow controlled ascent. We will further revise and clarify this in the paper.
>
>
> >Q2. Where are the green dots in Fig. 2 (a)-(e)?
>
> The green dots in Figure 2 are the initial objective values, which we omitted in the plots. We have regenerated the plots in Figure R1 in the attached PDF. We will change them in the revision.
>
> >Q3. What is the difference between the algorithms in Fig. 2 (e) and (f)?
>
> Both are FERERO, implemented with Algorithm 1. The difference is the choice of the **preference vectors** represented by the dashed arrows. Here, using the two results with different preference vectors, we want to show that FERERO can model flexible preferences.
>
> >Q4. What is the difference between the algorithms in Fig. 3 (c) and (f)?
>
> Both are FERERO, implemented with Algorithm 1. The difference lies in the **experiment settings**, as discussed in Section 5.1, lines 278-288. Specifically, the main difference is the initial values of the objectives. In Figures 3(a-c), all the initial objectives are in between the yellow and green preference vectors, and all three methods converge to the Pareto front, but their required iterations and convergence speeds are different. In Figures 3(d-f), the initial objective values are close to the blue or red preference vectors. In this setting, for PMTL, the green and yellow trajectories fail to converge to the Pareto front, but the other two methods can converge to the Pareto front and align with the corresponding preference vectors.

---

> > ### Comment · Reviewer_PTWJ · 2024-08-08
> >
> > Thanks for the response. The a-posteriori and a-priori comparison made in the response is unclear to me. Can you further clarify the difference of the approach from the one in [c,g]?  In these works, first, the user specifies preferences (a general polyhedral $C$ in [c] and $C=\mathbb{R}^d_{+}$ in [g]. The objective function is unknown. Then, sequential evaluation of designs is performed, taking into account $C$. Only the objective values at queried designs are observed (with noise). When the algorithms stop, they output an estimated Pareto set $P$ of designs that is ($\epsilon,\delta$)-PAC correct. It seems to me that the main difference from the prior work is the knowledge and use of gradient information.
> >
> > Can this algorithm be used to find all Pareto optimal solutions or a desired subset of them? Do you have any guarantees on coverage of all Pareto optimal solutions or the diversity of returned solutions, etc?

---

> > > ### Author Response · Authors · 2024-08-10
> > > **Detailed comparison between our work and [c,g]**
> > >
> > > Thank you very much for your prompt reply and the follow-up questions. After carefully reading the related works [c,g], we summarize the differences between our work and [c,g]. Then we answer the question about our theoretical guarantees.
> > >
> > > ### Problem difference
> > >
> > > **1. Continuous vs. discrete design space.**
> > > The design space, i.e., the domain of the optimization variables is different. [c,g] consider a discrete finite set $[K]$, see [c, Section 2, paragraph 2], while we consider a continuous space $\theta \in \mathbb{R}^q$ that might have infinitely many Pareto optimal solutions.
> > >
> > > **2. Gradient-based vs. blackbox optimization.**
> > > [c,g] focus on the best arm identification problem using blackbox optimization with unknown objectives. In contrast, we are using gradient-based optimization, where the objectives are known.
> > >
> > > **3. Preference specification.**
> > > The prior works [c,g] focus on ```relative preference only```, defined by a partial order cone, while our work considers ```both relative and absolute preferences```. The absolute preference is modeled by constraints $G(\theta), H(\theta)$ in our work.
> > >
> > > ### Method difference
> > >
> > > **A posteriori vs. a priori method [32].** Both methods need to specify the relative preference in the beginning, i.e., the partial order cone $C$ for the optimization problem to be well-defined.
> > >
> > > ***A-posteriori method [c,g]***: first returns a set of $C$-optimal solutions, then the user selects from the returned solutions based on their absolute preferences.
> > >
> > > ***A-priori method (ours)***: the user first specifies the absolute preferences before running the method, e.g., based on weights or constraints, then the method returns one solution that satisfies the absolute preferences and is $C$-optimal.
> > >
> > > Our method can find one Pareto optimal solution that satisfies the constraint, see Figures 2-5, the solutions align with the preference vectors in dashed lines. To the best of our knowledge, this cannot be achieved by [c,g].
> > >
> > > ### Guarantees to find all Pareto optimal solutions.
> > >
> > > >Can this algorithm be used to find all Pareto optimal solutions or a desired subset of them? Do you have any guarantees on coverage of all Pareto optimal solutions or the diversity of returned solutions, etc?
> > >
> > >
> > > ```No, our a-priori method returns one solution given the absolute preference, not all Pareto optimal solutions.```
> > >
> > > As there may be infinitely many Pareto optimal solutions, our method guarantees finding one solution that satisfies the absolute constraint and is Pareto optimal. We do not provide guarantees on finding all the Pareto optimal solutions. By running our algorithm multiple times, each time with a different absolute preference, we can obtain multiple different solutions, one solution at a time. We will clarify this in the paper.

---

> > > > ### Comment · Reviewer_PTWJ · 2024-08-12
> > > >
> > > > Thanks for the reply. The differences are clear now. Please include this comparison in the revised version and also mention the limitations of the current approach.

---

> > > > > ### Author Response · Authors · 2024-08-12
> > > > >
> > > > > Dear Reviewer PTWJ,
> > > > >
> > > > > Thank you for your helpful comments and suggestions! We are glad that the difference between our work and [c,g] is clarified.
> > > > >
> > > > > We will add this comparison and limitation in the paper, and make other promised revisions and clarifications to further improve our paper.

---

### Official Review · Reviewer_rRCm · 2024-07-07

**Soundness:** 3
**Presentation:** 3
**Contribution:** 2
**Rating:** 6
**Confidence:** 2

**Summary:**

This paper introduces a new algorithm called FERERO, which can handle both relative and absolute preferences in preference-guided multi-objective learning. The problem is formulated as a constrained vector optimization problem. The goal is to find the (approximated) $C_A$-optimal set that satisfies the absolute preference constraints. The cone $C_A$ can be interpreted as a relative preference that defines the objectives’ improvement directions.

In each iteration, the FERERO finds a direction $d^{\star}(\theta)=-\nabla F(\theta)A_{ag}^T\lambda^{\star}$, derived by solving a subprogram. The $\lambda^{\star}$ in $d^{\star}(\theta)$ is derived from the Lagrangian of subprogram. Moreover, the approximated update rules and a stochastic version of FERERO are proposed for large-scale learning problems.

The paper includes theoretical analysis of FERERO and its stochastic, approximated variants. The experimental comparisons with existing multi-gradient descent methods are also conducted to demonstrate the effectiveness of this approach.

**Strengths:**

1.	FERERO deals with both preference constraints and objective values in an adaptive way, eliminating the need to solve different subprograms at different stages.

2.	The finite-time convergence rates of FERERO and its variants are established and the proof is solid.

3.	The approximated rules, practical choice of preferences, and stochastic variants of FERERO-SA can be efficient for real-world problems.

**Weaknesses:**

1.	The step sizes in some of the convergence analyses, e.g., Theorem 2 and Theorem 3, depend on the $T$, which can be unknown in real-world problems.

2.	The computational cost per-iteration can be higher than the traditional scalarization methods.

3.	The studied problem is not well motivated. Compared to the traditional multi-objective optimization, the authors consider a more complicated case, where each objective function is a linear combination of the original multiple objectives. More real-world applications should be given to validate the significance of the problem.

4.	C_A is not used in the experiments. That is, traditional multi-objective optimization is considered. This seems to be insufficient.

**Questions:**

1.	The preference constraints are expressed as linear functions of $F(\theta)$. Is this kind of linear structure assumption for preference constraints common in real world problems?

2.	Could you explain your choice of step sizes in the experiment and how can it be related to your theoretical results.

**Limitations:**

Yes

---

> ### Author Rebuttal · Authors · 2024-08-07
>
> >W1. The step sizes in Theorems 2 and 3 depend on iterations $T$, which can be unknown in real-world problems.
>
> Choosing step size depending on the number of iterations $T$ is common in optimization convergence analysis. For example, in [4,10,42], for the convergence analysis of unconstrained MOO, the step sizes are chosen based on the pre-defined value $T$.
>
> In all our experiments, including the real-world emotion recognition and multi-lingual speech recognition tasks, a proper number of iterations $T$ is set before optimization, and then we can choose $\alpha_t$ accordingly. Detailed iterations and step sizes are provided in Appendix F.
>
> We also have general convergence results where the step size $\alpha_t$ is not explicitly chosen to be dependent on $T$. See the derivation in Appendix E.2, Eq. (E.18). They can be summarized as follows
>
> \begin{align}
> \sum_{t=0}^{T-1} \alpha_t ||d_t||^2 + \alpha_t ||H(\theta_t)||^2 = O\Big(\sum_{t=0}^{T-1}\alpha_t^2 + 1 \Big).
> \end{align}
>
> In such cases, we choose the step sizes to satisfy $\sum_{t=0}^{\infty} \alpha_t^2 < \infty$, and $\sum_{t=0}^{\infty} \alpha_t = \infty$. For example, $\alpha_t = \frac{1}{(t+1)^{\frac{1}{2}+\epsilon}}$ with $0<\epsilon\leq \frac{1}{2}$ is also a proper choice to guarantee convergence. Similar choices exist in stochastic single-objective optimization literature, see [f].
>
> [f] Optimization Methods for Large-Scale Machine Learning, Bottou et al.
>
>
> >W2. Computational cost per iteration is higher than the scalarization method.
>
> Yes, we have discussed this limitation in the Broader Impacts and Limitations section. Though traditional scalarization methods have better per-iteration complexity, the benefit of the proposed method with flexible preference modeling cannot be achieved by scalarization methods. For example, see the experiments in Figure 2. The linear scalarization (LS) method cannot find certain points on the Pareto front.
>
> Despite this, our computational cost per iteration is **similar to or lower than** existing non-scalarization-based methods [23,30]. See **General Response-1**.
>
> In addition, since we proposed single-loop algorithms (Algorithms 2, 3), existing techniques for fast approximation of Gramian evaluation in FAMO [h, Section 3.2] can be applied to our method to largely reduce the gradient evaluation per iteration.
>
> [h] Liu et al., "FAMO: Fast Adaptive Multitask Optimization," NeurIPS, 2023.
>
> >W3 & Q1. Motivation & examples of the studied problem.
>
> - In the introduction (Section 1), we provide two examples when the constraints are linear functions of the objectives, with further explanation in Appendix B.1. Furthermore, in the experiments of multi-lingual speech recognition, we have another example when the constraints are linear functions of the objectives.
> - In the experiments in Figures 3 and 5, we provide examples of using a general polyhedral cone $C_A$, where $A$ is not an identity matrix, to allow controlled ascent for the update of the objectives.
> - We provide more examples in the **General Response-3**.
>
>
> Also note that, our framework and algorithm with subprogram Eq. (2.1) can be directly applied to handle more general constraints that are not necessarily linear w.r.t. the objectives. However, it may require other constraint qualification (CQ) assumptions, or the CQs can be more challenging to prove. We leave this for future work.
>
>
> >W4. $C_A$ is not used in experiments.
>
> This might be a **misunderstanding**. The use of $C_A$ is reflected in that we use a **more general matrix $A$ that is not necessarily the identity matrix**. Please see our discussions in Section 5.
> - For synthetic data, in Figures 3(c) and 3(f), we use a more general $C_A$ as discussed in lines 278-288.
> - For multi-patch image classification, to avoid the fact that some preference vectors are not attainable, we choose different preference vectors and different corresponding $C_A$ to obtain the results in Figure 5.
>
> Specifically, in these cases, the extreme rays of $C_A$ are determined by the given preference vector, then $A$ is computed by solving (3.5).
>
>
> >Q2. How are the experimental choices of step sizes related to the theory?
>
>
> In our experiments, a proper $T$ is set before optimization, and we can choose $\alpha_t$ accordingly.
>
> - In the deterministic case, and for Algorithm 2, in theory, we choose $\alpha_t = \Theta(\frac{1}{T})$, i.e., $\frac{c_1}{T} \leq \alpha_t \leq \frac{c_2}{T}$ for positive constants $c_1,c_2$. This theoretical result is reflected in that under the same experiment settings, when we choose smaller $T$, we can choose larger step size $\alpha_t$; see e.g., in Table 10, Appendix F.2.
> - In theory, we require $\alpha_t c_h \leq 1$. This is reflected in the experiments in Tables 7,8,10 in Appendix F.

---

> > ### Comment · Reviewer_rRCm · 2024-08-11
> >
> > Thanks for the response. Most of my concerns have been addressed. I raise my score by 1.

---

> > > ### Author Response · Authors · 2024-08-12
> > >
> > > Dear Reviewer rRCm,
> > >
> > > Thank you for your helpful comments and acknowledgement!
> > >
> > > We will make the promised revisions and clarifications to further improve our paper.

---

### Official Review · Reviewer_aduv · 2024-07-09

**Soundness:** 4
**Presentation:** 4
**Contribution:** 3
**Rating:** 7
**Confidence:** 5

**Summary:**

This paper introduces FERERO, a novel framework for preference-guided multi-objective learning by depicting the task as a constrained single-objective optimization problem. It incorporates relative preferences, defined by a polyhedral cone, and absolute preferences, defined by linear constraints, into the above optimization problem. Both deterministic and stochastic algorithms with finite-time convergence guarantees are proposed. FERERO is validated through experiments on multiple benchmarks, demonstrating its effectiveness in finding competitive preference-guided Pareto optimal solutions.

**Strengths:**

1.  This paper is very well-written and its results are solid.
2.  A novel concept called relative preference is proposed, which interests me.
3.  A stochastic version is also proposed for fast gradient estimation.
4.  Only one optimization problem needs to be solved for each preference, while some methods require solving multiple subproblems.
5.  The experimental results are competitive and promising.

**Weaknesses:**

I do not notice any obvious weakness. Some comments are given below:

1. What is the evidence of ``flexible``? Is flexibility important?
2. What is the meaning of ``cross`` in Table 1? Is it not mentioned in the literature or is it inherently impossible to do this?
3. What are the advantages of a method based on general partial order? Is it related to escaping weak Pareto optimality?
4. As far as I am concerned, Linear Scalarization and (Smooth) Tchebycheff can escape weak Pareto optimality.
5. The per-iteration may need more descriptions.
6. What if some absolute preferences are not available? e.g., $B_h$ or $B_g$ are not provided by the decision maker. Besides, the settings of preferences in the experiments are hard to find.
7. Are there some typos in the subtitles of Figure 2(e) and Figure 3(c)?

**Questions:**

Refer to the above comments.

**Limitations:**

The discussion is comprehensive in my view.

---

> ### Author Rebuttal · Authors · 2024-08-07
>
> >Q1. Evidence and importance of "flexible".
>
> Indeed, this is a critical point of our paper!
>
> The "flexibility" is evidenced by that we model **both absolute and relative preferences**. Furthermore, absolute preference considers both inequality and equality constraints. The relative preference considers more general partial order compared to existing works. "Flexible" means we can flexibly choose the preferences such as absolute or relative preferences or both, without limiting to preference vectors or certain partial order cones.
>
> This is in contrast to scalarization methods [22] or other preference-guided MOL methods such as [23,30], which have relatively restrictive formulations. For example, this is further evidenced by the multi-lingual speech recognition experiments in Section 5.2, where both inequality and equality constraints are considered, which cannot be directly solved by existing methods [22,23,30] that focus on either inequality constraints or preference vectors. Also, as evidenced by Figures 3 and 5, we use a more general partial order cone to allow controlled ascent of objectives, which cannot be achieved by e.g., [23],  without such flexibility.
>
> This flexibility is important because it makes our method ```applicable to many other real-world problems``` as listed in **General Response-3**. Existing methods [22,23,30,33] without such flexibility may not be able to solve these problems.
>
>
> >Q2. Meaning of "cross" in Table 1. Is it not mentioned in the literature or inherently impossible to do?
>
> The "cross" means it is not mentioned or provided in the corresponding literature.
>
> >Q3. Advantages of general partial order? Is it related to escaping weak Pareto optimality?
>
> It is not related to escaping weak Pareto optimality. One key advantage of using general partial order is that it allows flexible modeling of more general relative preferences. And therefore, it extends the ability of the FERERO method to solve broader problems. We further explain with two detailed advantages below.
>
> - It allows us to use a primal algorithm with controlled ascent, as discussed in lines 98-99. Therefore, it is possible for us to find certain Pareto optimal solutions that would be otherwise impossible to find with a primal algorithm using $A=I$, e.g., see Figures 3 (d,f).
> - It makes the method applicable to many other real-world problems that cannot be solved using $A=I$, as discussed in **General Response-3**.
>
> The ability to escape weak Pareto optimality is because our formulation in Eq. (2.1) considers finding update direction $d$ such that the descent amount is normalized by function values. If certain objective function values already achieve zero, then the algorithm finds directions that further decrease other objectives rather than stopping at weak Pareto optimality. The ability to escape weak Pareto optimality is proved in Appendix D.3.
>
> >Q4. Scalarization methods can escape weak Pareto optimality.
>
> We are not aware that scalarization methods can directly escape weak Pareto optimality in general. For example, for linear scalarization on general nonconvex objectives, the gradient descent algorithm converges at a stationary solution of the linearly scalarized objective, which may be weakly Pareto optimal but not Pareto optimal.
>
> However, we are aware that under certain conditions, the global solutions of scalarization methods with positive weights are Pareto optimal [32, Theorem 3.1.2]. We will make revisions and clarifications on this point in Table 5. And we are very willing to cite proper references if the reviewer could kindly provide some.
>
> >Q5. More description of per-iteration complexity.
>
> We have provided a detailed description of the per-iteration complexity. See **General Response-1**.
>
> >Q6-1. What if preferences are unavailable, e.g., $B_h,B_g$ not provided?
>
> When the absolute preferences are not available, the MOO problem reduces to an unconstrained one, and the proposed algorithm reduces to an adaptive cone descent algorithm with the following subprogram replacing Eq. (2.1).
> \begin{align}
> \psi(\theta):=\min _{(d, c) \in \mathbb{R}^q \times \mathbb{R}} c+\frac{1}{2}||d||^2 \quad \text { s.t. } A \nabla F(\theta)^{\top} d \leq c\left(\mathbf{1}^{\top} A F(\theta)\right)^{-1} A F(\theta) .
> \end{align}
>
> If $B_h,B_g$ are not directly specified, they can be computed given the information of the preference. For example, if a preference vector $v\in \mathbb{R}^M$ is given, we can convert it to an equality constraint with $B_h$ satisfying $B_h v = 0$, where $B_h$ is formed by linearly independent row vectors that belong to $\mathrm{ker}(v)$.
>
> >Q6-2. The settings of preferences in experiments are hard to find.
>
> In synthetic, multi-patch image classification, and emotion recognition experiments, the preferences are specified by the preference vectors, following the experiment settings in [30]. The details of the preference vectors can be found in Appendix F. In these experiments, they are generated uniformly between certain angles, and plotted as dashed lines or arrows in Figures 2-5. As discussed in the answers to Q6-1, we can compute $B_h$ from the given preference vector $v$ by finding the orthonormal row vectors that belong to $\mathrm{ker}(v)$. When $M=2$, $v = [v_1;v_2]$, $B_h = [-v_2, v_1] \in \mathbb{R}^{1\times 2}$.
>
> In the multi-lingual speech recognition experiment, the problem formulation is provided in Eq. (5.2). In this case, $B_g = [1,0,0], b_g = -\epsilon_1$, and $B_h = [0,1,-1], b_h = -\epsilon_2$.
>
> We will further clarify the preference settings in the revision.
>
> For more practical examples of the preferences that fit in the framework we study in this paper, please see **General Response-3**.
>
> >Q7. Clarification of subtitles in Figures 2,3.
>
> The subtitle of Figure 2(e) is correct. It shows the result of our method with the same preference vectors as the previous methods. We will change the subtitle of Figure 3(c) from "Ours" to "FERERO" to be consistent.

---

> > ### Comment · Reviewer_aduv · 2024-08-10
> >
> > Thanks for your kind response. I am pleased to see that you have provided many examples to illustrate their considered preferences.
> >
> > Your method is general/flexible for various tasks, as it accommodates multiple types of preferences and still works when some preferences are unavailable. My concerns regarding weak Pareto optimality have been addressed, and the relationship between general partial order and weak Pareto optimality has been clarified. My other concerns have been addressed as well.

---

> > > ### Author Response · Authors · 2024-08-12
> > >
> > > Dear Reviewer aduv,
> > >
> > > Thank you very much for your helpful suggestions! We are glad that we have addressed your concerns.
> > >
> > > We will make the promised revisions and clarifications to further improve our paper.

---

### Official Review · Reviewer_dD8J · 2024-07-10

**Soundness:** 3
**Presentation:** 3
**Contribution:** 3
**Rating:** 6
**Confidence:** 3

**Summary:**

This paper tackled constrained multi-objective optimization problems with Pareto optimality generalized to partial order cone. The authors proposed an algorithm framework to iteratively solving a subproblem to get a update direction and aim at search for Pareto optimal solution in a certain Pareto front region associated pre-defined preferences.

The paper presented rich convergence analysis for deterministic algorithm with exact subproblem solutions, deterministic algorithm with inexact subproblem solutions, as well as stochastic version where gradient is estimated through double samplings.

The stochastic algorithm variant is tested on image classification and speech recognition tasks and compared against state-of-art benchmark algorithms.

**Strengths:**

* This paper is well-structured, well-written and easy to follow.
* The proposed algorithm framework can handle relative preferences where decision makers don’t need to specify explicit weights or threshold numbers. Also, the proposed algorithm has more flexible control on the search region on the Pareto front which is specified by extreme rays and/or constraints. This makes the algorithm more practical.
* This paper provides comprehensive theoretical convergence analysis on the proposed three algorithm variants in general non-convex cases. The ideal of normalizing search direction by function values is novel to me.
* The experiment on image and speech recognition tasks demonstrates its applicability to large scale neural network learning objectives. The improvement over recognition error rate is significant.

**Weaknesses:**

* The relative preference seems being handled by both partial order cone associated with the objective function and inequality constraints. It is kind of confusing and redundant to have both preference cone and constraints defined in the objective space in the optimization problem. See the questions part for more details. The experiment on multi-task learning seems only tackling constraints.
* In real decision making application, computing the ordering cone by solving another LP makes the decision making process more complex and less doable.
* The authors selected number of algorithm iterations to compare computational cost, which leads to apple to orange comparison. Better to adopt number of function evaluations, number of gradient evaluations, etc. as the metrics.

**Questions:**

* As both the relative preference and constraints are defined in objective space and both equality and inequality constraints are defined as linear functions of objectives, would it possible to convert at least inequality constraints to preference defined by a cone?
* Can we reduce Pareto dominance defined in a polyhedral cone to Pareto dominance defined in Real space $R^m_+$ by taking $AF(\theta)$ as objective vector?
* From Theorem 1 to Theorem 2, the second part on the left of Equation (2.5) $G(\theta_t)$ no longer present in equation (3.2). How is the inequality violation converging to zero in the single loop approximate algorithm variant?
* Equation (5.2): what is the rationale to have the representation learning loss for Chinese and English differing by an exact number. Would it be more reasonable to have the loss gap smaller than a threshold?
* It is not clear in Table 2 what is Emotion loss without referring to Appendix.
* Figure 2(a): “LS only finds extreme points on the PF with one objective minimized." How are the secularization weights selected? Is the weight selection related to extreme rays?
* Figure 4 clarification
    * 4(c) LS accuracy is missing. 4(a) also missing red square.
    * As mentioned in the paper, orange preference vector is not attainable. In (d)-(f), the proposed algorithm tends to search for Pareto front region closer to the ideal Pareto minimizer. Is this constraint violation controllable?
    * Figure (e) green cross given by XWC-MGDA seems dominating most of other solutions.

* Minor issues:
    * Notation consistency: Theorem 2 equation (3.2) $d_t$ → $d_t(\theta_t)$.
    * Line 279: figure 3d-3f should be 3a-3c.
    * Figure 3: caption (c) and (f) should be consistent.

---

> ### Author Rebuttal · Authors · 2024-08-06
>
> >W1-1 & Q1. Relative preferences are handled by both partial order and inequality constraints. It is confusing and redundant. Can inequality constraints be converted to the preference cone?
>
> We respectfully disagree. As mentioned in the abstract, relative preference is handled by the partial order, and absolute preference is handled by the constraints. The two preferences are generally **not redundant but complementary**. Essentially, the cone captures *relative improvement* of the objectives, and the constraints capture the *absolute feasible regions* of the objectives. Some detailed differences are explained below.
> - **Equality constraints cannot be captured** by the preference cone with nonempty interior.
> - **The constants $b_g$ in the inequality constraints cannot be captured** by the preference cone, as the cone can only guarantee relative improvement at each iteration.
> - **Our method deals with the two preferences differently.** For preference cone, our Algorithm 1 with proper step sizes guarantees the next iterate always improves over the previous one in terms of the partial order induced by the cone, i.e., $AF(\theta_{t+1}) \leq AF(\theta_t) \leq AF(\theta_0)$. In contrast, for preference constraints, our method does not guarantee feasibility at every iteration, i.e., $G(\theta_t)\leq 0, H(\theta_t)=0$, but only when the algorithm converges.
>
> Nevertheless, there are special cases where the constants $b_g$ in the inequality constraints align with the initial objective values $F_0$, with $B_g (F(\theta) - F_0) = B_g F(\theta) + b_g$. In this case, it is possible to convert the inequality constraints to preferences defined by a cone to ensure feasibility.
>
> >W1-2. The experiment on multi-task learning seems to only tackle constraints.
>
> This might be a misunderstanding. See Figures 3, 5. We use a more general cone $C_A$ to allow controlled ascent of objectives.
>
> >W2. Computing the ordering cone by solving another LP is complex.
>
> We do not always have to compute $A$. See **General Response-2**.
>
> When $A$ needs to be computed from the given extreme rays, the linear programming (LP) in Eq. (3.5) can be solved in polynomial time w.r.t. $M$. In practice and our experiments, $M\ll d$, therefore, solving this LP requires much less computation/time than solving the main program (PMOL) to obtain the optimal $\theta$.
>
> >W3. Comparison metrics using the number of function/gradient evaluations.
>
> Following your suggestion, we adopt these metrics. Results are summarized in **General Response-1**.
>
> >Q2. Take $AF(\theta)$ instead of $F(\theta)$ as objectives, and use $\mathbb{R}_+^M$ as the preference cone.
>
> Interesting point! This is possible **only under the special cases** with more restrictions on the constraints and preference cone.
>
> - This is possible when there are no constraints. However, for constrained problems with constraints defined as functions of $F(\theta)$, if we use $AF(\theta)$ as objectives, we need to compute $B_h A^{-1}(AF(\theta))$, which may lead to higher computational complexity, and thus not practical. Moreover, $A$ may not always be invertible in general.
> - This is only possible when the preference cone is polyhedral. For more general preference cones, they may not necessarily be written as in Definition 1. Therefore, one cannot use $AF(\theta)$ as objectives in such cases. However, in such cases, our FERERO method can be extended to cover the more general preference cones $C$ by replacing Eq. (2.1), $A\nabla F(\theta)^\top d \leq c (\mathbf{1}^\top AF(\theta))^{-1}AF(\theta)$ with $\nabla F(\theta)^\top d \leq_C c (\mathbf{1}^\top F(\theta))^{-1} F(\theta)$.
>
> >Q3. Inequality constraints in Theorem 2 are not present in Eq. (3.2).
>
> Sorry for the confusion. Indeed, we focus on equality constraints ($M_g = 0$) for Theorem 2. We will further clarify this in the paper. The results can be extended to include inequality constraints in future work.
>
> >Q4. Clarification of multilingual constraint in Eq. (5.2). Would it be more reasonable to have the loss gap smaller than a threshold?
>
> The multilingual constraint in Eq. (5.2) is a heuristic choice based on the experience. We found that in this experiment setting, if $\epsilon_2$ is close to zero, the performance in Chinese will be much better than the performance in English. However, we want the performance in both languages to be similar, hence we impose this equality constraint.
>
> >Q5. Clarification of emotion loss.
>
> The emotion loss refers to the emotion classification loss. The task is to predict 6 types of emotions from 593 songs on the Emotions and Music dataset [38]. See Appendix F.2. We will add explanations of the emotion loss in the main text in the revision if space allows.
>
> >Q6. How are the scalarization weights in Figure 2(a) selected? Is the weight selection related to extreme rays?
>
> The scalarization weights are selected uniformly from a simplex following the experiment settings in [30]. They are not related to extreme rays. We will clarify this in the revision.
>
> >Q7. Clarification of Figure 4.
>
> - *Missing square plots.* Sorry for the confusion. This is because we limit the plot scales to a certain range to make the majority of the results clear. We add plots with larger scales in Figure R2 in the attached PDF. We will include the plots in the revision.
> - *Controlling constraint violation.* Yes, constraint violation can be controlled by the hyperparameter $c_h$. Larger $c_h$ will make the constraint violation smaller.
> - *Figure (e) green plus is dominating.* Indeed, in this experiment, under the green preference vector, XWC-MGDA gives the best solution compared to other methods. However, under other preference vectors, the solutions of XWC-MGDA are not dominating, and the overall hypervolume of XWC-MGDA calculated from all the solutions is worse than the proposed method.
>
> >Q8. Minor issues.
>
> Thank you for your detailed comments! We will revise the paper accordingly.

---

> > ### Comment · Reviewer_dD8J · 2024-08-12
> >
> > Thanks for adding the computational time comparison. For Synthetic, Figures 3(d-f), Per-iteration run time of PMTL should be 7.65E-4s and  Per-iteration run time of FERERO should be 7.30E-4s. Otherwise, it looks good to me.
> >
> > A minor comment: regarding the multilingual constraint in Eq. (5.2), the author claimed that it is a heuristic choice based on experience. This will require domain knowledge for decision-makers to specify the exact gap number. As the proposed framework is capable of handling inequality constraints, a more practical choice is to use inequality constraints.
> >
> > The authors' response has addressed my concerns. I will keep my rating. Thanks!

---

> > > ### Author Response · Authors · 2024-08-12
> > >
> > > Dear Reviewer dD8J,
> > >
> > > Thanks for spotting this typo in Table R1. You are correct, we will make the changes.
> > >
> > > Thank you again for your detailed comments and helpful suggestions! We will make the promised revisions to further improve our paper.

---

### Author Rebuttal · Authors · 2024-08-07

## General Response

We thank all reviewers for their support and constructive comments. Below we address 3 common questions from the initial reviews.

>1. Comparison of computational cost with baselines.

Following the suggestion, we summarize the per-iteration complexity of different algorithms in terms of **gradient/function evaluation** in the table below.

|Algorithm|Per-iteration gradient evaluation|Per-iteration function evaluation|
|---|---|---|
|Linear scalarization|$1$|-|
|PMTL [23]|$M$|$M_g$|
|EPO [30]|$M$|$M$|
|(X)WC-MGDA [33]|$M$|$M$|
|**FERERO (Ours)**|$M$|$M$|

To summarize, our method has **similar per-iteration complexity with PMTL and EPO**, but it is **higher than linear scalarization** methods.

Among them, PMTL only considers inequality constraints, i.e., $M_h=0$. At each iteration, it computes $M$ gradients, and evaluates $M_g$ inequality constraints to determine whether the inequalities are active or not. EPO and (X)WC-MGDA also compute $M$ gradients per iteration. They both evaluate $M$ objective values to obtain the index set $J^*$ for EPO, and to obtain the subprogram constraints for (X)WC-MGDA. FERERO computes $M$ gradients, and evaluates the $M$ function values per iteration.

We also summarize the **iterations and run time** for all experiments in Table R1 in the attached PDF. In practice, our Algorithm 1 has similar per-iteration run time with PMTL, but higher than EPO. Our Algorithm 3 has lower per-iteration run time than both PMTL and EPO.

>2. How to choose/compute the preference cone?

We do not always have to compute the preference cone or $A$.

Usually, the preference cone is pre-defined by the user, represented by matrix $A$ or the extreme rays. Only when $A$ is not pre-defined, but the extreme rays of the cones are defined, we need to compute $A$ following Eq. (3.5) in order to use the FERERO method. Otherwise, we can simply use a pre-defined $A$.

>3. More motivating examples of the studied problem.

Thanks for raising this question! We provide examples from the following two cases.

- **Examples of general $C_A$.**
    - **Portfolio selection in security markets** requires finding weakly minimal points w.r.t. feasible portfolio cones, which are nonlattice, i.e., cones with more extreme rays than the ambiance space dimension. See [a].
    - **Multi-asset markets with transaction costs** use "solvency cones", where $A\neq I$ is defined in terms of the bid-ask prices of the assets. See [b].
    - **Optimal arm identification under noisy feedback** uses multiple objectives based on the rewards. The practitioners want to narrow down the set of solutions by using their domain knowledge about the relative importance of each objective. If the user wants to give at least 100$\alpha$% relative importance to each objective for $\alpha \in (0, 0.5)$, then this can be achieved by defining a new cone-induced partial order in [c, Eq. (2)].
    - **Small molecule drug discovery** considers the optimization of properties such as solubility, metabolic stability, and toxicity. By using a smaller cone $C_A \subset \mathbb{R}_+^M$ for in silico experiments, the practitioner can ensure that more designs are passed to the wet lab stage. See [c, Remark 2.3].

- **Examples of linear constraints of objectives.**
Linear structure in the constraints can be very common. Below we provide several examples with more details.
    - In **drug discovery**, constrained property optimization is often considered, where molecules are generated such that molecular properties are optimized, while satisfying certain constraints [d, 48]. See more discussion in Appendix B.1.
    - In **fairness-aware learning**, one needs to consider the tradeoff between fairness and utility. For example, one may want to maximize the classification accuracy of multiple tasks/population groups (objective function), while ensuring fairness. The fairness constraint can be introduced such that the accuracy discrepancy between two groups (e.g., female, male) is bounded by a certain threshold, or the utility of a certain group should reach the prescribed threshold. See [e].
    - In **self-supervised learning**, typically, the self-supervised loss without labeled data is combined with supervised losses for downstream tasks. This problem naturally fits in preference-guided MOL. The preference can be specified that the self-supervised loss is bounded by a certain threshold.
    - In our paper, a **multi-lingual speech recognition** example is given in Section 5.2, Eq. (5.2). We consider a combination of self-supervised learning and fairness-aware learning. We use the self-supervised loss $f_p(\theta)$ besides the supervised language prediction losses as objectives, and the difference of language performances as constraints.

[a] Aliprantis et al., "Equilibrium analysis in financial markets with countably many securities", J. Math. Econom. 40: 683–699, 2004.

[b] Kabanov, "Hedging and liquidation under transaction costs in currency markets", Finance and Stochastics, 3: 237–248, 1999.

[c] Cagin and Tekin, "Vector optimization with stochastic bandit feedback", AISTATS 2023.

[d] You et al., "Graph Convolutional Policy Network for Goal-Directed Molecular Graph Generation", NeurIPS 2018.

[e] Liu and Vicente, "Accuracy and fairness trade-offs in machine learning: A stochastic multi-objective approach", arXiv:2008.01132, 2020.

---

### Decision · Program_Chairs · 2024-09-25

**Decision:**

Accept (poster)

**Comment:**

All reviewers agree that this submission makes significant progress in the area of multi-objective learning. The authors propose a new framework based on constrained vector optimization and present new iterative (gradient descent type) algorithms. They obtain both theoretical guarantees and provide also convincing experimental results. There has been a discussion between the authors and the reviewers in which all concerns and questions of the reviewers could be resolved.